# Variational Neural Stochastic Differential Equations with Change Points

**Yousef El-Laham**                                                    *yousef.el-laham@jpmchase.com*
*J.P. Morgan AI Research*

**Zhongchang Sun**                                                        *zhongcha@buffalo.edu*
*University at Buffalo*

**Haibei Zhu**                                                          *haibei.zhu@jpmchase.com*
*J.P. Morgan AI Research*

**Tucker Balch**                                                        *tucker.balch@jpmchase.com*
*J.P. Morgan AI Research*

**Svitlana Vyetrenko**                                              *svitlana.vyetrenko@jpmchase.com*
*J.P. Morgan AI Research*

**Reviewed on OpenReview:** `https://openreview.net/forum?id=GEilvtsFNV`

## Abstract

In this work, we explore modeling change points in time-series data using neural stochastic differential equations (neural SDEs). We propose a novel model formulation and training procedure based on the variational autoencoder (VAE) framework for modeling time-series as a neural SDE. Unlike existing algorithms training neural SDEs as VAEs, our proposed algorithm only necessitates a Gaussian prior of the initial state of the latent stochastic process, rather than a Wiener process prior on the entire latent stochastic process. We develop two methodologies for modeling and estimating change points in time-series data with distribution shifts. Our iterative algorithm alternates between updating neural SDE parameters and updating the change points based on either a maximum likelihood-based approach or a change point detection algorithm using the sequential likelihood ratio test. We provide a theoretical analysis of this proposed change point detection scheme. Finally, we present an empirical evaluation that demonstrates the expressive power of our proposed model, showing that it can effectively model both classical parametric SDEs and some real datasets with distribution shifts.

## 1 Introduction

Stochastic differential equations (SDEs) are a class of probabilistic models frequently used to model continuous-time stochastic processes (Lelièvre & Stoltz, 2016; Soboleva & Pleasants, 2003; Huillet, 2007). They have a broad range of applications in fields such as quantitative finance, physics, biology, and engineering (Sauer, 2011; Browning et al., 2020). SDEs comprise two main components: a *drift* function, which models the deterministic evolution of the stochastic process over time, and a *diffusion* function, which captures the stochastic component of the process. In traditional SDE modeling, domain experts design simple parametric models for the drift and diffusion functions to encapsulate the key properties of the system of interest. Model parameters are then learned using statistical estimation approaches, such as the method of moments estimation or maximum likelihood estimation (Casella & Berger, 2024; Kay, 1993). While this SDE learning

process is feasible for a variety of applications, such as population ecology or mathematical finance, it can be challenging to apply in more complex systems. Recently, the concept of neural SDEs was introduced by integrating neural networks with SDEs (Li et al., 2020; Tzen & Raginsky, 2019; Hodgkinson et al., 2020). This offers a more adaptable approach to modeling real-world time-series, eliminating the need to define the structure of the drift and diffusion functions a prior.

Following the introduction of neural ordinary differential equations (neural ODEs), a wealth of research has emerged on neural SDEs to model the dynamics of a stochastic process $\{\boldsymbol{X}_t\}_{t\in[0,T]}$. In (Kidger et al., 2021), a connection was established between neural SDEs and Wasserstein generative adversarial networks (W-GANs), demonstrating that certain types of neural SDEs can be interpreted and trained within an infinite-dimensional GAN framework. An alternative approach to training neural SDEs involves the use of the variational autoencoder (VAE) framework, which has been adopted in various studies (Hasan et al., 2021; Li et al., 2020). The VAE framework was introduced in (Hasan et al., 2021) to learn latent SDEs from noisy observations, assuming a prior distribution for the latent variable at each time step. In (Li et al., 2020), the training of SDEs as VAEs was also explored, assuming a prior over a latent stochastic process characterized by an SDE with a diffusion term for tractability of the evidence lower bound (ELBO). However, both approaches assume a prior over the entire latent stochastic process $\{\boldsymbol{Z}_t\}_{t\in[0,T]}$, which may be too strong an assumption, as the training data may not always conform to this prior. Therefore, in this paper, we propose a new framework for training SDEs as VAEs that does not require such a strong prior in the latent space.

While much of the existing research on neural SDEs has primarily focused on time-series modeled by a single SDE, the underlying dynamics of real-world time-series data often surpass the complexity that a single model can capture. Scenarios where the dynamics of time-series abruptly change over time, such as the distributional shifts in stock prices during the COVID period, present significant challenges for existing approaches. In training neural SDEs, it's often assumed that the drift and diffusion terms exhibit Lipschitz continuity, a requirement necessary to ensure the convergence of SDE solvers (Kidger et al., 2021). However, this assumption can be restrictive, as a single SDE with Lipschitz continuous drift and diffusion terms may struggle to accurately model time-series with sharp distributional shifts. This limitation motivates our investigation into the problem of change point detection for neural SDEs. With the detected change point, the time-series can be further modeled using multiple SDEs conditioned on the occurrence of a change point. Similar work in this line of research includes the previously proposed neural jump SDE (Jia & Benson, 2019), which augments the neural ODE model with a temporal point process to model sharp changes in the ODE dynamics, without considering the stochastic nature (i.e., diffusion) of the time-series. In (Sun et al., 2024), a neural SDE model with change points is proposed based on the W-GAN framework; however, since the W-GAN framework is based on an implicit generative model, it is difficult to derive theoretical results regarding the convergence of the training algorithm.

In this paper, we introduce a framework for training SDEs as VAEs and develop an algorithm for change point detection in neural SDEs based on this VAE framework. Specifically, we propose an iterative algorithm for change point detection under unknown SDE dynamics, which alternately updates the change point estimate and the neural SDE model parameters. The algorithm is summarized in two steps: (1) *Update model parameters*: Given the current change point estimate, we train different SDE models based on our proposed VAE framework; and (2) *Update the change points*: Given the current model parameters, we run a likelihood ratio test sequentially to refine the change point estimates. Our specific contributions are as follows:

1. We propose a novel framework to train SDEs as VAEs. Unlike existing approaches, which require a prior over the latent stochastic process $\{\boldsymbol{Z}_t\}_{t\in[0,T]}$, our formulation only necessitates specifying a prior over the initial state $\boldsymbol{z}_0$;

2. Leveraging our proposed VAE framework, we develop two approaches for learning change points in time-series model as latent neural SDEs: a method based on the idea of maximum likelihood estimation and a change detection algorithm based on the sequential likelihood ratio test. We utilize the Euler-Maruyama approximation to SDE solutions and apply suitable stochastic filtering methodologies to obtain an unbiased estimator of both the marginal likelihood of the change point and the test statistic in the sequential likelihood ratio test;

3. We develop an iterative algorithm to jointly learn the SDE model parameters and the unknown change points. Under certain conditions, we demonstrate that our iterative algorithm achieves performance guarantees regarding the estimation accuracy;

4. Lastly, we demonstrate the generative power of the neural SDE model on our proposed distributional shift generation benchmark datasets, showing that our model outperforms state-of-the-art deep generative models across a variety of metrics.

## 2 Problem Formulation

Let $\boldsymbol{W} = \{\boldsymbol{W}_t\}_{t \in [0,T]}$ denote a $d_w$-dimensional Brownian motion with admissible filtration $\mathbb{F} = (\mathcal{F}_t)_{t \in [0,T]}$ on the interval $[0, T]$. This work is concerned with modeling the distribution of an $\mathbb{R}^{d_x}$-valued continuous-time stochastic process $\boldsymbol{X} = \{\boldsymbol{X}_t\}_{t \in [0,T]}$ defined on the filtered probability space $(\Omega, \mathcal{F}, \mathbb{F}, \mathbb{P})$, which is assumed to be the solution of an SDE of the following form:

$$d\boldsymbol{X}_t = f(t, \boldsymbol{X}_t)dt + g(t, \boldsymbol{X}_t)d\boldsymbol{W}_t, \quad t \in (0, T] \tag{1}$$

where $\boldsymbol{X}_0 \sim \mu_0$ is the initial state following the initial distribution $\mu_0$, $f : [0, T] \times \mathbb{R}^{d_x} \to \mathbb{R}^{d_x}$ is called the drift function, and $g : [0, T] \times \mathbb{R}^{d_x} \to \mathbb{R}^{d_x \times d_w}$ is called the diffusion function. The drift and diffusion functions are typically assumed to satisfy some regularity conditions. Mainly, the drift and diffusion functions are bounded above by a linear function:

$$\|f(t, \boldsymbol{x})\| + \|g(t, \boldsymbol{x})\| \le \gamma_1(1 + \|\boldsymbol{x}\|), \quad \boldsymbol{x} \in \mathbb{R}^{d_x}, \quad t \in [0, T], \tag{2}$$

and also satisfy a Lipschitz smoothness condition:

$$\|f(t, \boldsymbol{x}_1) - f(t, \boldsymbol{x}_2)\| + \|g(t, \boldsymbol{x}_1) - g(t, \boldsymbol{x}_2)\| \le \gamma_2\|\boldsymbol{x}_1 - \boldsymbol{x}_2\|, \quad \boldsymbol{x}_1, \boldsymbol{x}_2 \in \mathbb{R}^{d_x}, \quad t \in [0, T], \tag{3}$$

for some constants $\gamma_1, \gamma_2 > 0$. Under these assumptions, the stochastic process $\boldsymbol{X}$ is said to be a strong solution of the SDE in (1) if it satisfies (1) for each sample path of the Wiener process $\{\boldsymbol{W}_t\}_{t \in [0,T]}$ and for all $t$ in the defined time interval almost surely (Oksendal, 2013). These conditions are also sufficient for proving strong order convergence of the Euler-Maruyama method (Kloeden et al., 1992).

Our goal in this work is to learn the drift and diffusion of the SDE defined in (1) given an irregularly sampled time-series $\boldsymbol{x}_{\text{obs}} = (\boldsymbol{x}_{t_1}, \boldsymbol{x}_{t_2}, \dots, \boldsymbol{x}_{t_K})$, where $t_k \in (0, T]$ for all $k$. An ideal methodology would be robust to potential distribution shifts and could potentially model change points in the time-series (see Figure 1).

## 3 Related Work

Due to the large capacity of neural networks for function approximation, neural SDEs have been proposed to allow for data-driven learning of SDEs. In neural SDEs, the drift and diffusion are modeled via neural networks, rather than "simple" pre-defined parametric functions. Neural SDEs can be trained using the VAE framework (Li et al., 2020; Hasan et al., 2021) where it is assumed that there is an underlying latent stochastic process $\{\boldsymbol{Z}_t\}_{t \in [0,T]}$ with some prior distribution. In the following, we review an existing approach for training neural SDEs using the VAE framework.

### 3.1 Neural SDEs under the Variational Autoencoder Framework

Training SDEs as VAEs has been studied in (Li et al., 2020), where the prior is defined over the $d_z$-dimensional latent stochastic process $\boldsymbol{Z} = \{\boldsymbol{Z}_t\}_{t \in [0,T]}$, which is characterized by an SDE:

$$d\boldsymbol{Z}_t = f_{\boldsymbol{\alpha}}(\boldsymbol{Z}_t, t)dt + g_{\boldsymbol{\alpha}}(\boldsymbol{Z}_t, t)d\boldsymbol{W}_t, \quad t \in (0, T] \tag{4}$$

where $\boldsymbol{z}_0 \sim \mu_0$ denotes the initial state of $\boldsymbol{Z}$ with corresponding initial distribution $\mu_0$ and $\boldsymbol{\alpha}$ denotes a set of hyperparameters. The posterior of $\{\boldsymbol{Z}_t\}_{t \in [0,T]}$ is approximated as the solution of another SDE, which is of

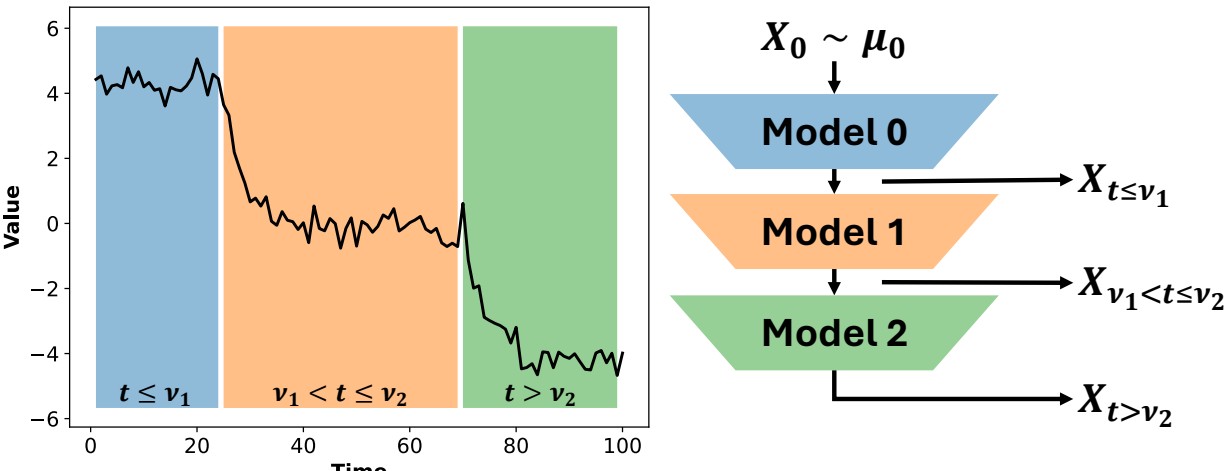

Figure 1: An example of a non-stationary time-series of length $T = 100$. Within the highlighted segments, the time-series is stationary and can be easily modeled with generative models, such as neural SDEs. The sharp distributional shifts occurring at the change points complicate the modeling of such a time-series with a single out-of-the-box generative model.

the form:

$$z_0 \sim q_\phi(z_0|x_{\text{obs}}) \tag{5}$$

$$dZ_t = f_\phi(Z_t, t)dt + g_\phi(Z_t, t)dW_t, \quad t \in (0, T], \tag{6}$$

where $\phi$ denotes the parameters of the variational approximation. Given the latent variable $Z_t$, we assume that the observation $X_t$ has a distribution characterized by:

$$X_t = h_\theta(Z_t) + \varepsilon_t, \tag{7}$$

where $\theta$ denotes a set of parameters, $\varepsilon_t$ are i.i.d. noise terms usually assumed to be Gaussian distributed and independent of $Z_t$. Here, the function $h_\theta$ can be thought of as a decoder that decodes each sampled $Z_t$ to the mean of the original stochastic process $X$ sampled at the same time point $X_t$.

In (Li et al., 2020), it is assumed that the diffusion terms for the prior SDE and posterior SDE are the same, i.e., $g_\alpha(z_t, t) = g_\phi(z_t, t)$. Let $q_\phi(z_t|x_{\text{obs}})$ denote the marginal posterior of $z_t$ for all $t \in [0, T]$. Then, the lower-bound to the marginal likelihood called the ELBO, denoted by $\tilde{\mathcal{E}}(\phi, \theta; x_{\text{obs}})$, can be established as follows:

$$\log p_\alpha(x_{\text{obs}}) \geq \tilde{\mathcal{E}}(\phi, \theta; x_{\text{obs}}) \tag{8}$$

$$\triangleq \mathbb{E}_{q_\phi}\left[\sum_{k=1}^{K} \log p_\theta(x_{t_k}|z_{t_k}) - \int_0^T \frac{1}{2}\|u_\phi(z_t, t)\|_2^2 dt\right], \tag{9}$$

where $u_\phi(z_t, t) = g_\phi^{-1}(z_t, t)(f_\phi(z_t, t) - f_\alpha(z_t, t))$. Therefore, the parameters of the neural SDE model can be optimized by maximizing the ELBO. Exact evaluation of $\tilde{\mathcal{E}}(\phi, \theta; x_{\text{obs}})$ is intractable, but a Monte Carlo approximation can be obtained by sampling from the variational approximation:

$$\tilde{\mathcal{E}}(\phi, \theta; x_{\text{obs}}) \approx \frac{1}{J}\sum_{j=1}^{J}\sum_{k=1}^{K} \log p_\theta(x_{t_k}|z_{t_k}^{(j)}) - \frac{1}{J}\sum_{j=1}^{J}\int_0^T \frac{1}{2}\|u_\phi(z_{t_k}^{(j)}, t_k)\|_2^2 dt,$$

where $z^{(j)} = \{z_{t_1}^{(j)}, \ldots, z_{t_K}^{(j)}\}$ denotes a sampled trajectory of the stochastic process $Z$ from the variational approximation and $J$ is the total number of sampled trajectories. In (Li et al., 2020), stochastic gradients are

obtained by using the adjoint sensitivity method; although other solvers, such as the Euler-Maruyama solver, can trivially be applied. At inference time, samples from the generative model are obtained by sampling from the latent trajectory from the Wiener process prior, which are then processed by the decoder. Hereafter, we refer to the approach proposed by (Li et al., 2020) as the `LatentSDE` model.

### 3.2 Identifying Change Points with Latent SDEs

A change point detection scheme based on the aforementioned variational framework was proposed in (Ryzhikov et al., 2022). The authors propose to utilize a sequential likelihood ratio test (SLRT) to detect changes in a given time-series using a trained SDE model based on the VAE framework. To elaborate, in their work, a single `LatentSDE` model is trained to model a given time-series dataset. Once the model is trained, it is used as a means for online change detection using the following derived test statistic approximated with a Monte Carlo average based on samples drawn from the variational posterior:

$$\text{CCPD}(\boldsymbol{x}_t) = \sum_{l=1}^{L} \log \left( \frac{p(\boldsymbol{x}_t|t)}{p(\boldsymbol{x}_t|t-l)} \right), \tag{10}$$

$$= \sum_{l=1}^{L} \log \left( \frac{\int p(\boldsymbol{x}_t|\boldsymbol{z}_t) q_\phi(\boldsymbol{z}_{0:t}|\boldsymbol{x}_{\text{obs}}) d\boldsymbol{z}_{0:t}}{\int p(\boldsymbol{x}_t|\boldsymbol{z}_{t-l}) q_\phi(\boldsymbol{z}_{0:t-l}|\boldsymbol{x}_{\text{obs}}) d\boldsymbol{z}_{0:t-l}} \right) \tag{11}$$

$$\approx \sum_{l=1}^{L} \log \left( \frac{\frac{1}{M} \sum_{m=1}^{M} \mathcal{N}\left(\boldsymbol{x}_t|f(\boldsymbol{z}_t^{(m)}), \boldsymbol{C}\right)}{\frac{1}{M} \sum_{m=1}^{M} \mathcal{N}\left(\boldsymbol{x}_t|f(\boldsymbol{z}_{t-l}^{(m)}), \boldsymbol{C}\right)} \right) \tag{12}$$

where $p(\boldsymbol{x}_t|t)$ denotes the likelihood of observing $\boldsymbol{x}_t$ at time $t$, $\boldsymbol{z}_{0:t}^{(m)} \sim q_\phi(\boldsymbol{z}_{0:t}|\boldsymbol{x}_{\text{obs}})$ denotes a sampled trajectory from the variational posterior of the trained `LatentSDE` model, $f : \mathbb{R}^{d_z} \to \mathbb{R}^{d_x}$ denotes the decoder function, $\boldsymbol{C}$ denotes a prior covariance matrix, and $L$ denotes a lag. An important distinction of this work from ours is that their proposed method focused on the *online* detection task and does not explicitly include change points in the modeling of the latent SDE. This implies that their model cannot be used for generation of time-series with distributional shifts, but only as a means to detecting shifts in the data (or future data). Moreover, theoretical insights shown in their work focused only on the analytical form of the test statistic, rather than the theoretical properties of their algorithm (e.g., convergence guarantees and rates). We want to re-emphasize that the goal of our work is to design a neural SDE model to accurately capture the dynamics of time-series data exhibiting distributional shifts, which requires capturing the change points in an *offline* manner. This is in contrast to the goal of the work in (Ryzhikov et al., 2022), which purely focuses on the detection task.

### 3.3 Neural SDEs Trained as GANs

As an alternative to the VAE formulation to modeling neural SDEs, an approach based on a W-GAN formulation was also proposed in (Kidger et al., 2020), which we refer to `SDEGAN` hereafter. An important distinction between this approach and its VAE counterpart is the the input noise distribution in `SDEGAN` only defines the initial state of the latent SDE. An approach for modeling change points in neural SDEs has already been proposed based on the W-GAN framework (Sun et al., 2024), which extends the original work on training neural SDEs as infinite-dimensional GANs by (Kidger et al., 2021). In this work, change points were directly modeled in the latent SDE dynamics (via the W-GAN generator network). The training of the model alternated between two phases: (1) updating the W-GAN parameters with fixed change points; and (2) updating the change points using a CUSUM-type algorithm (Page, 1954) with test statistic based on the difference in discriminator scores between two consecutive windows of a time-series dataset. The proposed test statistic turns out to be connected to the Wasserstein-1 distance, and so the algorithm can be viewed as performing an approximate Wasserstein two-sample test (see (Ramdas et al., 2017)) for making change point updates. While the approach proposed in (Sun et al., 2024) demonstrated good empirical performance for generation of time-series with distributional shift, the theoretical validity of the method remains an open question. Furthermore, recent works have shown that W-GANs provide inaccurate measures

to the Wasserstein distance (Mallasto et al., 2019; Stanczuk et al., 2021) and therefore, the justification of the approach based on Wasserstein two-sample testing becomes questionable.

### 3.4 Limitations of Existing Approaches

Most works on VAE-based neural SDEs are structured in a manner similar to the aforementioned approach, where the prior is assumed over the entire latent process (e.g., one can assume that *a priori* $\{\boldsymbol{Z}_t\}_{t \in [0,T]}$ is a Wiener process). This modeling assumption, however, may be too restrictive in practice since the training data might not always conform to this prior latent SDE, which may degrade the generative performance of the model.To elaborate, the ELBO objective function in (9) is comprised of two-terms: a reconstruction term (expected log-likelihood); and a penalty term (KLD between prior and posterior stochastic processes). The penalty term will ensure that solution of the posterior SDE will not be too far away from the Wiener process prior. The adverse effect of this regularization is that the decoder function needs to be powerful enough to transform trajectories from a Wiener process to trajectories from the underlying data distribution. Another important point is that, in the training of neural SDEs, it's common to assume that the drift function $f$, and the diffusion function $g$, have Lipschitz continuity which ensures the existence of a unique and strong solution to the SDE (1). Assuming smooth drift and diffusion, however, may limit the model's capability to accurately model time-series with sudden distributional shift (e.g., sharp changes in the mean or volatility). A simple way to account for distributional shift is to incorporate change points into the `LatentSDE` model by utilizing different decoders before and after the change point. Unfortunately, this type of modeling does not explicitly capture how the dynamics of the underlying SDE change before and after the change point - a single Wiener process prior would be utilized at inference time to generate trajectories and the decoder function is responsible for capturing the changes.

## 4 Proposed Methodology

In this work, we design a novel algorithm for training neural SDEs that does not require a strong prior in the latent space to train the SDEs as VAEs. Furthermore, within the VAE framework, we propose an algorithm to incorporate change points to identify distribution shifts in the times series. Given the change points, we model the time-series as multiple SDEs based on the change points. Specifically, we propose an optimization procedure that alternately updates the change point estimate and the SDE model parameters. To simplify the presentation, in the following, we consider the case where there is one change point. Our algorithm can be generalized to the case with multiple change points. A high-level overview of our modeling approach is summarized in Figure 2. In Fig 2, a time-series sample is first passed into an encoder (e.g., LSTM or neural CDE), which outputs the variational posterior parameters of the initial state of the latent SDE $\{\boldsymbol{Z}_t\}_{t \geq 0}$. An SDE solver (with SDE dynamics based on $\boldsymbol{\theta}_0$) is employed to sample the stochastic process $\{\boldsymbol{Z}_t\}_{t \geq 0}$ at times $t \leq \nu$ (before change point). The terminal latent state of this sample $\boldsymbol{Z}_\nu$ is then passed to a second SDE solver (with SDE dynamics based on $\boldsymbol{\theta}_1$), which samples the latent SDE until time $T$. To obtain the corresponding samples in the original time-series space, a probabilistic decoder (e.g., fully connected network) is used to decode each sampled latent SDE code $\boldsymbol{Z}_t$ into its corresponding value in the original data space $\boldsymbol{X}_t$.

### 4.1 System Model

To incorporate change points in our model, we assume that a change occurs at an unknown time $\nu \in (0, T]$. That is, the latent process $\{\boldsymbol{Z}_t\}_{t \in [0,T]}$ in our model is characterized by two different SDEs before and after the change point:

$$\boldsymbol{z}_0 \sim p(\boldsymbol{z}_0), \tag{13}$$

$$d\boldsymbol{Z}_t = f_{\boldsymbol{\theta}_0}(\boldsymbol{Z}_t, t)dt + g_{\boldsymbol{\theta}_0}(\boldsymbol{Z}_t, t)d\boldsymbol{W}_t, \quad t \in (0, \nu], \tag{14}$$

$$d\boldsymbol{Z}_t = f_{\boldsymbol{\theta}_1}(\boldsymbol{Z}_t, t)dt + g_{\boldsymbol{\theta}_1}(\boldsymbol{Z}_t, t)d\boldsymbol{W}_t, \quad t \in (\nu, T], \tag{15}$$

where $f_{\boldsymbol{\theta}_0}$ and $g_{\boldsymbol{\theta}_0}$ are the latent drift and diffusion neural networks (parameterized by $\boldsymbol{\theta}_0$) before the change point $\nu$, and $f_{\boldsymbol{\theta}_1}$ and $g_{\boldsymbol{\theta}_1}$ are the latent drift and diffusion neural networks (parameterized by $\boldsymbol{\theta}_1$) after the

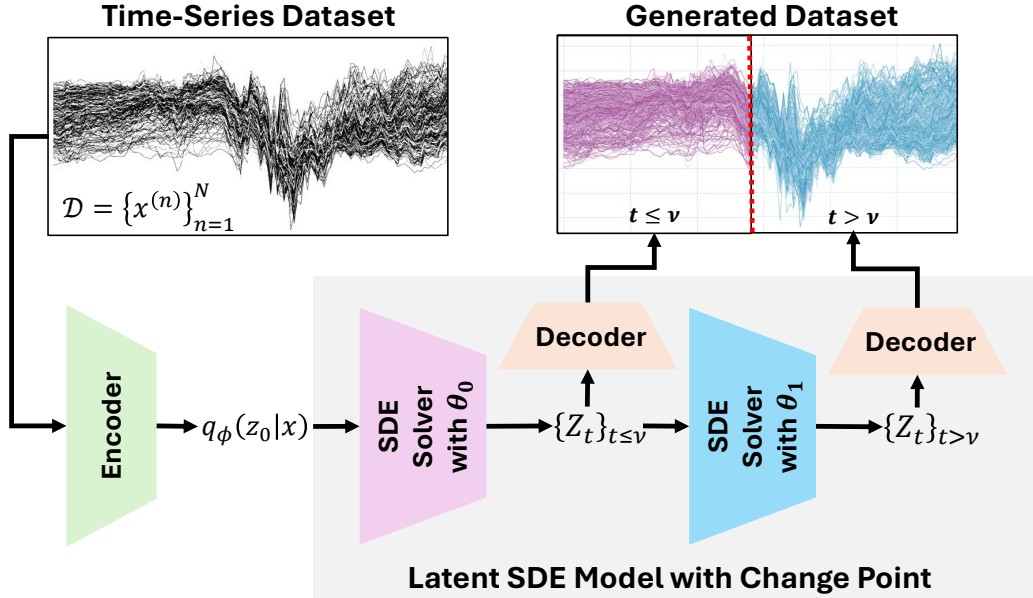

Figure 2: A simplified flow diagram of the latent SDE model considered in this work that accounts for potential change points in the time-series dataset.

change point. The observation process is modeled as:

$$\boldsymbol{X}_t = h_{\boldsymbol{\theta}_h}(\boldsymbol{Z}_t) + \boldsymbol{\varepsilon}_t, \tag{16}$$

where $h_{\boldsymbol{\theta}_h} : \mathbb{R}^{d_z} \to \mathbb{R}^{d_x}$ is assumed to be a fully connected neural network with standard activations and $\boldsymbol{\varepsilon}_t \sim \mathcal{N}(\boldsymbol{0}, \sigma_h^2 \boldsymbol{I}_{d_x})$. We highlight that the decoder is homogeneous across time and is thus not impacted by the change point.

## 4.2 Algorithm Summary

Let $\boldsymbol{\theta} = \{\boldsymbol{\theta}_0, \boldsymbol{\theta}_1, \boldsymbol{\theta}_h\}$ denote the "decoder" parameters. Consider a variational approximation $q_{\boldsymbol{\phi}}(\boldsymbol{z}_0|\boldsymbol{x}_{\mathrm{obs}})$ of the posterior of the initial state, where $\boldsymbol{\phi}$ denote the parameters of the variational approximation. We train the neural SDE model with change points using an iterative algorithm, where each iteration of the algorithm has two steps. In the first step, given the current change point estimate $\nu^{(i-1)}$, we update the model parameters $\boldsymbol{\theta}^{(i)} \leftarrow \boldsymbol{\theta}^{(i-1)}$ and the variational parameters $\boldsymbol{\phi}^{(i)} \leftarrow \boldsymbol{\phi}^{(i-1)}$ by maximizing the ELBO. In the second step, given the current value of the model parameters $\boldsymbol{\theta}^{(i)}$, we update the change point $\nu^{(i)} \leftarrow \nu^{(i-1)}$ by maximizing the marginal likelihood of the observed data. We present pseudocode for the training algorithm in Algorithm 1 and discuss each of the two steps in more details in the following.

## 4.3 Model Parameter Updates

We update the model parameters $\boldsymbol{\theta}$ (given $\nu$) using variational inference, by introducing a variational approximation over the posterior of the initial state of the latent stochastic process $\boldsymbol{z}_0$ given the observed data $\boldsymbol{x}_{\mathrm{obs}}$. Let $\nu^{(i)}$ denote our current guess of the change point at iteration $i$ of our algorithm. In our work, the parameters $\boldsymbol{\theta}$ are updated by maximizing the following lower bound on the log-evidence in the case of fixed change point $\nu = \nu^{(i)}$:

$$\log p_{\boldsymbol{\theta}}(\boldsymbol{x}_{\mathrm{obs}}|\nu = \nu^{(i)}) \geq \mathcal{E}_{\boldsymbol{\theta}, \boldsymbol{\phi}, \nu^{(i)}}(\boldsymbol{x}_{\mathrm{obs}}),$$

where $\mathcal{E}_{\boldsymbol{\theta}, \boldsymbol{\phi}, \nu}(\boldsymbol{x}_{\mathrm{obs}})$ is defined as

$$\mathcal{E}_{\boldsymbol{\theta}, \boldsymbol{\phi}, \nu}(\boldsymbol{x}_{\mathrm{obs}}) \triangleq -\mathcal{D}_{\mathrm{KL}}(q_{\boldsymbol{\phi}}(\boldsymbol{z}_0|\boldsymbol{x}_{\mathrm{obs}})\|p(\boldsymbol{z}_0)) + \mathbb{E}_{q_{\boldsymbol{\phi}}}\left[\log p_{\boldsymbol{\theta}}(\boldsymbol{x}_{\mathrm{obs}}|\boldsymbol{z}_0)\right].$$

---

**Algorithm 1** Variational Neural SDEs with Change Points (`CP-SDEVAE`)

---

Initialize model parameters $\boldsymbol{\theta}^{(0)} = \{\boldsymbol{\theta}_0^{(0)}, \boldsymbol{\theta}_1^{(0)}, \boldsymbol{\theta}_h^{(0)}\}$, variational parameters $\boldsymbol{\phi}^{(0)}$ and change point estimate $\nu^{(0)}$.

**for** $i = 1$ to $E$ **do**                                                  ▷ Number of training epochs

    **Update model parameters:**

        Fixing $\nu = \nu^{(i-1)}$, update $\boldsymbol{\theta}^{(i)} \leftarrow \boldsymbol{\theta}^{(i-1)}$ and $\boldsymbol{\phi}^{(i)} \leftarrow \boldsymbol{\phi}^{(i-1)}$ by minimizing the loss function in (21).

    **Update change point:**

        Fixing $\boldsymbol{\theta} = \boldsymbol{\theta}^{(i)}$, update the change point $\nu^{(i)} \leftarrow \nu^{(i-1)}$ by maximizing the marginal likelihood given $\boldsymbol{\theta}$. This can be done exactly using the greedy maximum likelihood-based update or approximately with the fast detection-based update.

**Return:** $\boldsymbol{\theta}^{(E)}, \boldsymbol{\phi}^{(E)}, \nu^{(E)}$.

---

A key distinction between this ELBO and the one utilized in (Li et al., 2020) is that the variational posterior is defined only over the initial state. This pushes the influence of the latent SDE dynamics into the expected log-likelihood term, rather than the KLD penalty. This choice gives us mainly two advantages:

1. If $p(\boldsymbol{z}_0)$ is Gaussian and the choice of the variational approximation $q_{\boldsymbol{\phi}}(\boldsymbol{z}_0|\boldsymbol{z}_{\text{obs}})$ is Gaussian, the KLD penalty can be analytically computed. In (Li et al., 2020), tractability of the KLD penalty is achieved by making the more restrictive choice that the prior and posterior diffusion are the same.

2. After training, the learned latent neural SDE dynamics are utilized to generate samples. Our sampling procedure is a direct analog to the GAN-based approach presented in (Kidger et al., 2020), which has been shown to work practically well on a variety of datasets, where the initial state of the latent SDE is generated from random noise and then propagated through the GAN generator (VAE decoder in our case).

The challenge of utilizing our variational formulation is now the tractability of the expected log-likelihood $\mathbb{E}_{q_{\boldsymbol{\phi}}}[\log p_{\boldsymbol{\theta}}(\boldsymbol{x}_{\text{obs}}|\boldsymbol{z}_0)]$, which we discuss in the following.

### 4.3.1 Expected Log-Likelihood

Let $\boldsymbol{z}_{\text{obs}}$ denote the latent stochastic process $\{\boldsymbol{Z}_t\}_{t\in[0,T]}$ sampled at same time steps as $\boldsymbol{x}_{\text{obs}}$. By the law of total probability, we can write:

$$p_{\boldsymbol{\theta}}(\boldsymbol{x}_{\text{obs}}|\boldsymbol{z}_0) = \int p_{\boldsymbol{\theta}}(\boldsymbol{x}_{\text{obs}}|\boldsymbol{z}_{\text{obs}})p_{\boldsymbol{\theta}}(\boldsymbol{z}_{\text{obs}}|\boldsymbol{z}_0)d\boldsymbol{z}_{\text{obs}}$$

$$= \int p_{\boldsymbol{\theta}}(\boldsymbol{z}_{\text{obs}}|\boldsymbol{z}_0)\left(\prod_{k=1}^{K} p_{\boldsymbol{\theta}}(\boldsymbol{x}_{t_k}|\boldsymbol{z}_{t_k})\right)d\boldsymbol{z}_{\text{obs}}.$$

Thus, the expected log-likelihood term can be written as a nested expectation:

$$\mathcal{L}_{\boldsymbol{\theta},\boldsymbol{\phi}}(\boldsymbol{x}_{\text{obs}}) \triangleq \mathbb{E}_{q_{\boldsymbol{\phi}}}[\log p_{\boldsymbol{\theta}}(\boldsymbol{x}_{\text{obs}}|\boldsymbol{z}_0)]$$

$$= \mathbb{E}_{q_{\boldsymbol{\phi}}}\left[\log \mathbb{E}\left[\prod_{k=1}^{K} p_{\boldsymbol{\theta}}(\boldsymbol{x}_{t_k}|\boldsymbol{z}_{t_k})\middle|\boldsymbol{z}_0\right]\right], \tag{17}$$

where the inner expectation is taken with respect to $p_{\boldsymbol{\theta}}(\boldsymbol{z}_{\text{obs}}|\boldsymbol{z}_0)$. For almost all choices of latent drift and diffusion of the neural SDE, this expression is intractable, but can be approximated using a nested Monte Carlo estimator:

$$\widehat{\mathcal{L}}_{\boldsymbol{\theta},\boldsymbol{\phi}}(\boldsymbol{x}_{\text{obs}}) = \frac{1}{J}\sum_{j=1}^{J}\log\left(\frac{1}{M}\sum_{m=1}^{M}\prod_{k=1}^{K} p_{\boldsymbol{\theta}}(\boldsymbol{x}_{t_k}|\boldsymbol{z}_{t_k}^{(j,m)})\right),$$

where $\boldsymbol{z}_{\text{obs}}^{(j,m)} \sim p_{\boldsymbol{\theta}}(\boldsymbol{z}_{\text{obs}}|\boldsymbol{z}_0^{(m)})$ is sampled via an SDE solver and $\boldsymbol{z}_0^{(m)} \sim q_{\boldsymbol{\phi}}(\boldsymbol{z}_0|\boldsymbol{x}_{\text{obs}})$ for $j = 1, \dots, J$ and $m = 1, \dots, M$. The mean-squared error (MSE) of this estimator converges to 0 at a rate of $\mathcal{O}(\frac{1}{J} + \frac{1}{M})$ (Rainforth et al., 2018), implying that the estimator is consistent (i.e., converges in probability to the true expected log-likelihood). By standard results in stochastic optimization, this should guarantee that the optimization of the ELBO will converge (in expectation) to a local optimum of the model parameters, since one component of the ELBO can be approximated via a consistent estimator (expected log-likelihood) and the other component can be computed analytically (KLD). In practice, stochastic gradients of $\widehat{\mathcal{E}}_{\boldsymbol{\theta},\boldsymbol{\phi}}$ can be obtained "backpropagating through the SDE solver", for which there are a variety of different approaches including the Euler-Maruyama method, the pathwise method (Yang & Kushner, 1991), and the adjoint sensitivity method (Li et al., 2020). The Euler-Maruyama method solves the SDE using finite differences, generating a sample path using a sequence of Gaussian state transitions. The reparameterization trick typically employed in VAE training can easily be applied since each transition is Gaussian, enabling low variance stochastic gradients. The pathwise method is considered to be the continuous-time analog to the reparameterization trick, while the adjoint sensitivty method works by solving a SDE whose solution is the desired gradient. In this work, we consider the Euler-Maruyama solver, but note that both the pathwise method and the adjoint sensitivity method can easily be utilized as well. For more information on the advantages and disadvantages of each solver, we refer to (Li et al., 2020), which provides a nice summary.

### 4.3.2 Expected Predictive Log-Likelihood

In recent works, the problem of noise underestimation in `LatentSDE` models has been investigated (Heck et al., 2024), a phenomenon that is analogous to a recurring observation made in underestimation of posterior variance in variational inference. Since our work also employs variational inference, a weakness of the training loss for the model parameters of the `SDEVAE` model is it will also lead to an underestimation of noise variance, since it is also balancing an expected log-likelihood term (reconstruction error) and a KLD term.

To understand specifically why this is problematic for our model, let us consider a simple example. Consider that the decoder of our model is linear, that is let $h_{\boldsymbol{\theta}_h}(\boldsymbol{z}_t) = \boldsymbol{\alpha}_h \boldsymbol{z}_t + \boldsymbol{\beta}_h$, where $\boldsymbol{\alpha}_h \in \mathbb{R}^{d_x \times d_x}$ and $\boldsymbol{\beta}_h \in \mathbb{R}^{d_x}$. When utilizing a single trajectory a single trajectory $\boldsymbol{z}_{\text{obs}}$ for the nested Monte Carlo estimator ($J = 1$, $M = 1$), the expected log-likelihood when using a linear decoder is approximated as:

$$\widehat{\mathcal{L}}_{\boldsymbol{\theta},\boldsymbol{\phi}}(\boldsymbol{x}_{\text{obs}}) = \sum_{k=1}^{K} \log p_{\boldsymbol{\theta}}(\boldsymbol{x}_{t_k}|\boldsymbol{z}_{t_k}), \tag{18}$$

$$= \sum_{k=1}^{K} \log \mathcal{N}(\boldsymbol{x}_{t_k}|\boldsymbol{\alpha}_h \boldsymbol{z}_{t_k} + \boldsymbol{\beta}_h, \sigma_h^2 \boldsymbol{I}_{d_x}), \tag{19}$$

$$= -\frac{K d_x}{2} \log(2\pi\sigma_h^2) - \sum_{k=1}^{K} \frac{(\boldsymbol{x}_{t_k} - (\boldsymbol{\alpha}_h \boldsymbol{z}_{t_k} + \boldsymbol{\beta}_h))^2}{2\sigma_h^2}. \tag{20}$$

As can be seen from (20), the expected log-likelihood term focuses on optimizing the model parameters, such that the marginals distributions of each $\boldsymbol{x}_{t_k}$ are well-calibrated (the underlying dynamics of the latent SDE are not influencing the expected log-likelihood). We note that hyperparameter tuning $\sigma_h^2$ does not solve the noise estimation issue, as changing $\sigma_h^2$ also only effects the marginal distribution of $\boldsymbol{x}_{t_k}$ and thus places emphasis on calibrating the marginal distribution of $\boldsymbol{x}_{t_k}$ given $\boldsymbol{z}_{t_k}$ for all $k$.

To improve the generative quality of our model, we propose to improve noise estimation by regularizing the ELBO with the following additional term:

$$\mathcal{L}_{\boldsymbol{\theta},\boldsymbol{\phi}}^{\text{pred}}(\boldsymbol{x}_{\text{obs}}) \triangleq \mathbb{E}_{q_{\boldsymbol{\phi}}}\left[\sum_{k=1}^{K} \log\left(\mathbb{E}\left[p_{\boldsymbol{\theta}}(\boldsymbol{x}_{t_k}|\boldsymbol{z}_{t_{k-1}})\Big|\boldsymbol{z}_0\right]\right)\right],$$

where the inner expectation is taken with respect to $p(\boldsymbol{z}_{t_{k-1}}|\boldsymbol{z}_0)$. We refer to $\mathcal{L}_{\boldsymbol{\theta},\boldsymbol{\phi}}^{\text{pred}}(\boldsymbol{x}_{\text{obs}})$ as the *expected predictive log-likelihood*. Just like the standard expected log-likelihood, $\mathcal{L}_{\boldsymbol{\theta},\boldsymbol{\phi}}^{\text{pred}}(\boldsymbol{x}_{\text{obs}})$ can be approximated with a nested MC estimator. For our estimator, we use a first-order Taylor approximation to obtain a Gaussian

approximation for the distribution $p_{\boldsymbol{\theta}}(\boldsymbol{x}_{t_k}|\boldsymbol{z}_{t_{k-1}})$, an approximation typically used in extended Kalman filtering, which is designed for non-linear state-space models with additive Gaussian (Kalman, 1960; Smith et al., 1962). A key difference between $\mathcal{L}_{\boldsymbol{\theta},\boldsymbol{\phi}}$ and $\mathcal{L}_{\boldsymbol{\theta},\boldsymbol{\phi}}^{\text{pred}}$ is that maximizing $\mathcal{L}_{\boldsymbol{\theta},\boldsymbol{\phi}}^{\text{pred}}$ encourages well-calibrated conditional distributions $p_{\boldsymbol{\theta}}(\boldsymbol{z}_{t'}|\boldsymbol{z}_t)$ rather than well-calibrated marginal distributions $p_{\boldsymbol{\theta}}(\boldsymbol{z}_t)$. We have found that empirically, this improves the generative performance of our model in terms of capturing noise properties in the time-series. To summarize, when updating the model parameters, for a fixed change point $\nu$ and observed time-series $\boldsymbol{x}_{\text{obs}}$ we minimize the following loss function:

$$\ell(\boldsymbol{\theta}, \boldsymbol{\phi}; \boldsymbol{x}_{\text{obs}}, \nu) = \lambda_{\text{kl}} \mathcal{D}_{\text{KL}}(q_{\boldsymbol{\phi}}(\boldsymbol{z}_0|\boldsymbol{x}_{\text{obs}}) \| p(\boldsymbol{z}_0)) - \lambda_{\text{nll}} \mathcal{L}_{\boldsymbol{\theta},\boldsymbol{\phi}}(\boldsymbol{x}_{\text{obs}}) - \lambda_{\text{pred}} \mathcal{L}_{\boldsymbol{\theta},\boldsymbol{\phi}}^{\text{pred}}(\boldsymbol{x}_{\text{obs}}), \quad (21)$$

where $\lambda_{\text{kl}} > 0$, $\lambda_{\text{nll}} > 0$, and $\lambda_{pred} > 0$ are regularization constants. In practice, we often train generative models on a collection of time-series $\{\boldsymbol{x}_{\text{obs}}^{(n)}\}_{n=1}^N$, in which case we minimize the average loss across all time-series (amortized inference) to make model parameter updates:

$$\boldsymbol{\theta}^{(i)}, \boldsymbol{\phi}^{(i)} = \operatorname*{arg\,min}_{\boldsymbol{\theta} \in \Theta, \boldsymbol{\phi} \in \Phi} \frac{1}{N} \sum_{n=1}^N \ell(\boldsymbol{\theta}, \boldsymbol{\phi}; \boldsymbol{x}_{\text{obs}}^{(n)}, \nu^{(i-1)}). \quad (22)$$

where $\Theta$ and $\Phi$ denote the feasibile sets of the model parameters and the variational approximation parameters, respectively.

## 4.4 Change Point Updates

We present two approaches for updating the change points: a greedy approach based on exact maximum likelihood estimate; and an online approach based on the sequential likelihood ratio test. For simplicity, we assume that the change point $\nu$ belongs to the set of sampled time points $\mathcal{T} = (t_1, \ldots, t_K)$. We refer the reader to the Appendix for an extension to the case where the change point can occur at any time index in $(0, T)$. Before delving into each approach, we provide an overview of particle filtering methods and how they can be used for obtaining an estimator of the change point likelihood $p_{\boldsymbol{\theta}}(\boldsymbol{x}_{\text{obs}}|\nu = t)$, which is a critical quantity for the change point update.

### 4.4.1 Particle Filtering for Change Point Likelihood Estimation

Particle filtering is a stochastic filtering methodology for approximating the posterior distribution of a latent process given sampled observations from another stochastic process. Consider the system model in Section 4.1 under the assumption that the change point is fixed to $\nu = \tau$. The system model can approximately be expressed in terms of a system of probability distributions:

$$\text{State Equation}: \qquad \boldsymbol{z}_{t_k} \sim p_{\boldsymbol{\theta}}(\boldsymbol{z}_{t_k}|\boldsymbol{z}_{t_{k-1}}, \nu = \tau) = \begin{cases} p_{\boldsymbol{\theta}_0}(\boldsymbol{z}_{t_k}|\boldsymbol{z}_{t_{k-1}}), & t_k \le \tau \quad \text{(before change)} \\ p_{\boldsymbol{\theta}_1}(\boldsymbol{z}_{t_k}|\boldsymbol{z}_{t_{k-1}}), & t_k > \tau \quad \text{(after change)} \end{cases}$$

$$\text{Observation Equation}: \qquad \boldsymbol{x}_{t_k} \sim p_{\boldsymbol{\theta}_h}(\boldsymbol{x}_{t_k}|\boldsymbol{z}_{t_k})$$

The goal of a particle filtering method is to obtain a sample-based (discrete random measure) approximation to the filtering distribution $p_{\boldsymbol{\theta}}(\boldsymbol{z}_{t_k}|\boldsymbol{x}_{t_{1:k}}, \nu = \tau)$ or the smoothing distribution $p_{\boldsymbol{\theta}}(\boldsymbol{z}_{t_{0:k}}|\boldsymbol{x}_{t_{1:k}}, \nu = \tau)$ by using importance sampling. For example, in this system model, the smoothing distribution $p_{\boldsymbol{\theta}}(\boldsymbol{z}_{t_{0:k}}|\boldsymbol{x}_{t_{1:k}})$ can be expressed in terms of the joint distribution $p_{\boldsymbol{\theta}}(\boldsymbol{z}_{t_{0:k}}, \boldsymbol{x}_{t_{1:k}}|\nu = \tau)$ and the normalizing constant $p_{\boldsymbol{\theta}}(\boldsymbol{x}_{t_{1:k}}|\nu = \tau)$:

$$p_{\boldsymbol{\theta}}(\boldsymbol{z}_{t_{0:k}}|\boldsymbol{x}_{t_{1:k}}) = \frac{p_{\boldsymbol{\theta}}(\boldsymbol{z}_{t_{0:k}}, \boldsymbol{x}_{t_{1:k}}|\nu = \tau)}{p_{\boldsymbol{\theta}}(\boldsymbol{x}_{t_{1:k}}|\nu = \tau)}$$

$$\propto p(\boldsymbol{z}_0) \left( \prod_{s=1}^k p_{\boldsymbol{\theta}_h}(\boldsymbol{x}_{t_s}|\boldsymbol{z}_{t_s}) \right) \underbrace{\left( \prod_{s:t_s \le \nu} p_{\boldsymbol{\theta}_0}(\boldsymbol{z}_{t_s}|\boldsymbol{z}_{t_{s-1}}) \right) \left( \prod_{s:t_s > \nu} p_{\boldsymbol{\theta}_1}(\boldsymbol{z}_{t_s}|\boldsymbol{z}_{t_{s-1}}) \right)}_{p_{\boldsymbol{\theta}}(\boldsymbol{z}_{t_{1:t_k}}|\boldsymbol{z}_0, \nu = \tau) = \prod_{s=1}^k p_{\boldsymbol{\theta}}(\boldsymbol{z}_{t_s}|\boldsymbol{z}_{t_{s-1}}, \nu = \tau)}$$

The fundamental idea behind the particle filtering approach is sequential importance sampling, which utilizes a proposal distribution at time $t_k$ that is factorized in a manner similar to the Markov process defining the state equation:

$$q(\boldsymbol{z}_{0:t_k}|\boldsymbol{x}_{t_{1:k-1}}) = q(\boldsymbol{z}_0)\prod_{s=1}^{k} q(\boldsymbol{z}_{t_s}|\boldsymbol{z}_{t_{s-1}},\boldsymbol{x}_{t_s})$$

At time instant $t_k$, the (unnormalized) importance weight of a trajectory sampled from $\boldsymbol{z}_{0:t_k}^{(j)} \sim q(\boldsymbol{z}_{0:t_k}|\boldsymbol{x}_{t_{1:k-1}})$, denoted by $\tilde{w}_{t_k}^{(j)}$, is weighted according to the smoothing distribution $p_{\boldsymbol{\theta}}(\boldsymbol{z}_{t_{0:k}}|\boldsymbol{x}_{t_{1:k}})$ can be recursively computed as follows:

$$\tilde{w}_{t_k}^{(j)} \propto \tilde{w}_{t_{k-1}}^{(j)} \frac{p_{\boldsymbol{\theta}_h}(\boldsymbol{x}_{t_k}|\boldsymbol{z}_{t_k}^{(j)})p_{\boldsymbol{\theta}}(\boldsymbol{z}_{t_k}^{(j)}|\boldsymbol{z}_{t_{k-1}}^{(j)},\nu=\tau)}{q(\boldsymbol{z}_{t_k}^{(j)}|\boldsymbol{z}_{t_{k-1}}^{(j)},\boldsymbol{x}_{t_k})}, \quad j=1,\ldots,J.$$

The pairs of sampled trajectories and their weights in particle filtering provides a means for obtaining estimators of quantities related to the smoothing distribution. An variation of particle filtering is bootstrap particle filtering (BPF), which samples trajectories according to the assumed state model, i.e., $q(\boldsymbol{z}_{t_s}|\boldsymbol{z}_{t_{s-1}},\boldsymbol{x}_{t_s}) = p_{\boldsymbol{\theta}}(\boldsymbol{z}_{t_s}|\boldsymbol{z}_{t_{s-1}},\nu=\tau)$ and includes an additional resampling step to avoid the path degeneracy problem. In this case, the importance weights are proportional to the likelihood function:

$$\tilde{w}_{t_k}^{(j)} \propto p_{\boldsymbol{\theta}_h}(\boldsymbol{x}_{t_k}|\boldsymbol{z}_{t_k}^{(j)}),$$

due to the fact that if the particle streams are resampled at each time instant, then $\tilde{w}_{t_{k-1}}^{(j)} \propto \frac{1}{J}$ for all $j$. Finally, we discuss the utility of particle filtering in the context of this work, which is that it can be used to evaluate the marginal likelihood of a particular change point (which is used in our maximum likelihood update of the change point) and it can be used to compute likelihood ratios (which is used to in our detector based update of the change point).

**Marginal likelihood of a change point:** An important quantity in this work is the marginal likelihood of the change point $\nu$ being equal to a particular value $\tau$ (over a time horizon $T = t_K$), which can be approximated as a product of the average importance weight:

$$p_{\boldsymbol{\theta}}(\boldsymbol{x}_{\text{obs}}|\nu=\tau) \approx \widehat{Z}_{t_k}^{\nu=\tau} = \left(\prod_{k=1}^{K} \frac{1}{J}\sum_{j=1}^{J} \tilde{w}_{t_k}^{(j)}\right) \tag{23}$$

Under weak assumptions, this estimator is unbiased and converges almost surely to the true marginal likelihood (Crisan & Doucet, 2002).

**Approximation of the likelihood ratio for change point detection:** The likelihood ratio is a fundamental quantity in statistics, typically used to construct a test statistic for a hypothesis test. For instance, for change point detection, being able to compute the log-likelihood ratio $\Lambda(\boldsymbol{x}_{t_{1:k}})$, which we define as:

$$\Lambda(\boldsymbol{x}_{t_{1:k}}) \triangleq \log\left(\frac{p_{\boldsymbol{\theta}}(\boldsymbol{x}_{t_{1:k}}|\nu=\tau)}{p_{\boldsymbol{\theta}}(\boldsymbol{x}_{t_{1:k}}|\nu>\tau)}\right), \tag{24}$$

where the numerator in (24) corresponds to the likelihood the change point occurs at time $\tau$ and the denominator corresponds to the likelihood the change point does not occur at time $\tau$, but at a later time. Under both models, the latent trajectories generated up until time $t_k$ are the same - they are both generated by latent SDE with parameter $\boldsymbol{\theta}_0$. The difference in these likelihoods comes from the fact that in the case of the numerator, $\boldsymbol{z}_{t_{k+1}}$ is sampled by propagating the previous latent state $\boldsymbol{z}_{t_k}$ with post-change SDE (with parameters $\boldsymbol{\theta}_1$) rather than the pre-change SDE (with parameters $\boldsymbol{\theta}_0$). It turns out this quantity can be approximated with BPF by taking the ratio of their average importance weights. To be precise, we can form the following approximation of the marginal likelihoods for each hypothesis at time step $\tau = t_k$ for each

model:

$$\widehat{p}^J(\boldsymbol{x}_{t_{1:k}}|\nu = t_k) = \left(\frac{1}{J}\sum_{j=1}^{J}\tilde{w}_{t_k}^{(j,1)}\right)\prod_{s=1}^{k-1}\frac{1}{J}\sum_{j=1}^{J}\tilde{w}_{t_s}^{(j,0)}, \tag{25}$$

$$\widehat{p}^J(\boldsymbol{x}_{t_{1:k}}|\nu > t_k) = \prod_{s=1}^{k}\frac{1}{J}\sum_{j=1}^{J}\tilde{w}_{t_s}^{(j,0)}, \tag{26}$$

where $\tilde{w}_{t_s}^{(j,0)}$ and $\tilde{w}_{t_s}^{(j,1)}$ denote the importance weights of the $j$th particle stream when propagated by the pre-change SDE and post-change SDE at the instant $t_s$, respectively. Given these two approximations, we can estimate the log-likelihood ratio $\Lambda(\boldsymbol{x}_{t_{1:k}})$ using an estimator $\widehat{\Lambda}(\boldsymbol{x}_{t_{1:k}})$ at time instant $t_k$ as[1]:

$$\widehat{\Lambda}(\boldsymbol{x}_{t_{1:k}}) = \log\left(\frac{\widehat{p}^J(\boldsymbol{x}_{t_{1:k}}|\nu = t_k)}{\widehat{p}^J(\boldsymbol{x}_{t_{1:k}}|\nu > t_k)}\right) \tag{27}$$

$$= \log\left(\frac{\frac{1}{J}\sum_{j=1}^{J}\tilde{w}_{t_k}^{(j,1)}}{\frac{1}{J}\sum_{j=1}^{J}\tilde{w}_{t_k}^{(j,0)}} \times \prod_{s=1}^{k-1}\frac{\frac{1}{J}\sum_{j=1}^{J}\tilde{w}_{t_s}^{(j,0)}}{\frac{1}{J}\sum_{j=1}^{J}\tilde{w}_{t_s}^{(j,0)}}\right) \tag{28}$$

$$= \log\left(\frac{1}{J}\sum_{j=1}^{J}\tilde{w}_{t_k}^{(j,1)}\right) - \log\left(\frac{1}{J}\sum_{j=1}^{J}\tilde{w}_{t_k}^{(j,0)}\right), \tag{29}$$

### 4.4.2 Greedy Update: Maximum Likelihood

Now that we have discussed particle filtering methods, we can now elaborate how change points can be updated in our algorithm. Change point updates are made by finding the optimal value of the change points given the most recently updated model parameter. We define the optimal change point update $\nu^{(i)}$ as the one that maximizes the marginal likelihood of the data:

$$\nu^{(i)} = \arg\max_{\tau \in \mathcal{T}} p(\boldsymbol{x}_{\text{obs}}|\nu = \tau). \tag{30}$$

By the chain rule of probability, we can write:

$$p(\boldsymbol{x}_{\text{obs}}|\nu = t) = \prod_{k=1}^{K} p(\boldsymbol{x}_{t_k}|\boldsymbol{x}_{t<t_k}, \nu = t) \tag{31}$$

where $\boldsymbol{x}_{t<t_k}$ denotes the observed data such before time $t_k$. While for general models $p(\boldsymbol{x}_{\text{obs}}|\nu = t)$ is an intractable integral, it can be recursively estimated using Bayesian filtering techniques. In this work, we use particle filtering (Djuric et al., 2003), which provides a straightforward way to obtain a consistent estimator $\widehat{p}(\boldsymbol{x}_{\text{obs}}|\nu = t)$ for $p(\boldsymbol{x}_{\text{obs}}|\nu = t)$ (please see (23)). We call the maximum likelihood update for $\nu$ the *greedy update* because it requires $\mathcal{O}(|\mathcal{T}|^2)$ runs of the BPF to estimate the marginal likelihood for all candidate values $\nu \in \mathcal{T}$ (see Algorithm 2). This may not be practical for long sequences - and so we propose an alternative approach for a faster update of $\nu$ based on the sequential likelihood ratio test.

---

[1]Note that in the approximation of log-likelihood ratio in (27), the number of particles generated for both pre-/post- SDE are assumed to be the same (i.e., $J$ trajectories); however, one can generalize the estimator to consider different numbers of generated trajectories for the pre-/post- change (i.e., $J_0$ for the pre-change SDE and $J_1$ for the post change SDE).

---

**Algorithm 2** Maximum Likelihood CP Update

---

Initialize particle filtering particles. Initialize $\log \widehat{Z}_{0:0}^{\nu=0} = 0$.

**for** $k = 1$ to $K$ **do**     ▷ Number of sampled times

    **Run particle filter with model parameters fixed to $\widehat{\theta}$ and obtain marginal likelihood estimator:**

        Run PF from time $t_k$ to time $t_K$ and approximate of the logarithm of the marginal likelihood $\log p(\boldsymbol{x}_{\mathrm{obs}}|\nu = t_k)$:

$$\log p(\boldsymbol{x}_{\mathrm{obs}}|\nu = t_k) \approx \log \widehat{Z}^{\nu=t_k}$$
$$= \log \widehat{Z}_{0:t_{k-1}}^{\nu=t_k} + \log \widehat{Z}_{t_k:T}^{\nu=t_k}$$

        Note: Our estimator is composed of two components: $\log \widehat{Z}_{0:t_{k-1}}^{\nu=t_k}$ and $\log \widehat{Z}_{t_k:T}^{\nu=t_k}$. The component $\log \widehat{Z}_{0:t_{k-1}}^{\nu=t_k}$ can be obtained from particles recycled from the previous PF run.

    **Change point greedy approximation:**

$$\widehat{\nu} = \arg \max_{t \in \mathcal{T}} \log Z^{\nu=t}$$

**Return:** $\widehat{\nu}$.

---

**Algorithm 3** Detection-based CP Update

---

Initialize particle filtering particles. Initialize $\log \widehat{Z}_0 = 0$.

**for** $k = 1$ to $K$ **do**     ▷ Number of sampled times

    **Propagate particle using assuming no change point and assuming a change point:**

        Run PF to approximate log marginal likelihood under $\mathcal{H}_{0,k} : \nu > t_k$:

$$\log p(\boldsymbol{x}_{1:t_k}|\nu > t_k) \approx \log \widehat{Z}_{t_k}^{\nu>t_k}$$
$$= \log \widehat{Z}_{k-1} + \log \widehat{Z}_{t_k}^{\nu>t_k}$$

        Run PF to approximate log marginal likelihood under $\mathcal{H}_{1,k} : \nu = t_k$:

$$\log p(\boldsymbol{x}_{1:t_k}|\nu = t_k) \approx \log \widehat{Z}_{t_k}^{\nu=t_k}$$
$$= \log \widehat{Z}_{k-1} + \log \widehat{Z}_{t_k}^{\nu=t_k}$$

    **Approximate log-likelihood ratio:**

$$\log \widehat{\Lambda}(\boldsymbol{x}_{t_1:t_k}) = \log \widehat{Z}_{t_k}^{\nu=t_k} - \log \widehat{Z}_{t_k}^{\nu>t_k}$$

    **If** $\log \widehat{\Lambda}(\boldsymbol{x}_{t_1:t_k}) > \gamma$:
        **Return:** $\widehat{\nu} = t_k$
    **Else:**
        **Set:** $\log \widehat{Z}_k = \log \widehat{Z}_{t_k}^{\nu>t_k}$
**Return:** $\widehat{\nu} = \arg \max_{t \in \mathcal{T}} \log \Lambda(\boldsymbol{x}_{t_1:t})$.

---

### 4.4.3 Fast Update: Sequential Likelihood Ratio Detector

A fast and online method for updating the change points at each training iteration is to use a sequential change point detection scheme (Polunchenko & Tartakovsky, 2012). Notably, the CUSUM algorithm has been applied for detecting change points in neural SDEs trained as W-GANs, where an approximated Wasserstein distance based on the learned W-GAN critic is used to detect the change point in a single forward pass of $\mathcal{O}(|T|)$ segments of the time-series (obtained via a sliding window). Practically speaking, it is only useful for neural SDEs trained under the W-GAN framework, since a proxy for computing the Wasserstein distance is required. Furthermore, the learned change point does not have any theoretical guarantees. Unlike W-GANs, which are *implicit* generative models, VAEs are *explicit* generative models and provide easy access to the probability measures of the latent and observed processes. This allows us to utilize the sequential likelihood ratio test for detecting the change point, a test for which theoretical implications have been well-studied.

Specifically, our change point updates are inspired by the classical sequential testing framework, where we consider the candidate change points $\tau$ to belong to the set of sampled time points $\mathcal{T}$. At each time $t_k \in \mathcal{T}$ we decide between two hypotheses:

$$\mathcal{H}_0 : \boldsymbol{x}_{t_{1:k}} \sim p(\boldsymbol{x}_{t_{1:k}}|\nu > t_k),$$
$$\mathcal{H}_1 : \boldsymbol{x}_{t_{1:k}} \sim p(\boldsymbol{x}_{t_{1:k}}|\nu = t_k),$$

where $\boldsymbol{x}_{t_{1:k}} = (\boldsymbol{x}_{t_1}, \ldots, \boldsymbol{x}_{t_k})$. The null hypothesis $\mathcal{H}_0$ is that the change occurs after time $t_k$ (and thus, the detection algorithm continues to run) and the alternative hypothesis $\mathcal{H}_1$ is that the change occurs precisely at $\nu = t_k$ (and thus, we stop the detection algorithm and adopt $\nu = t_k$ as the change point). We adopt the

change point update as the value of $t_k$ that rejects the null hypothesis, i.e., when

$$\log \Lambda(\boldsymbol{x}_{t_{1:k}}) \triangleq \log p(\boldsymbol{x}_{t_{1:k}}|\nu = t_k) - \log p(\boldsymbol{x}_{t_{1:k}}|\nu > t_k) \geq \gamma, \tag{32}$$

where $\Lambda(\boldsymbol{x}_{t_{1:k}})$ denotes the likelihood ratio of the test at time $t_k$ and $\gamma$ is a threshold determined by the pre-specified false alarm probability of the test $\alpha$. In practice, the log-likelihood ratio is typically monitored as the test statistic. Importantly, evaluation of the likelihood ratio involves the integration over $\boldsymbol{Z}_{t_{0:k}}$ (in both the numerator and denominator) and thus, is generally an intractable quantity. Similar to the greedy approach for updating the change points, we use a BPF to sequentially obtain an estimator of $\Lambda(\boldsymbol{x}_{t_{1:k}})$ based on (27)-(29):

$$\widehat{\Lambda}^J(\boldsymbol{x}_{t_{1:k}}) = \frac{\widehat{p}^J(\boldsymbol{x}_{t_{1:k}}|\nu = t_k)}{\widehat{p}^J(\boldsymbol{x}_{t_{1:k}}|\nu > t_k)}, \tag{33}$$

where $J$ denotes the number of trajectories sampled in the BPF. The advantage of the sequential testing approach is that a maximum of $|\mathcal{T}|$ BPF steps are needed to detect the change, which can all be done using a single run of the BPF, reducing the change point update complexity to $\mathcal{O}(|\mathcal{T}|)$ BPF steps.

## 4.5 Theoretical Insights

In this section, we provide some theoretical insights of our proposed work. Mainly, we show that under certain assumptions, the training algorithm converges to a stationary point w.r.t. the ELBO. We also show that our detection scheme, under certain assumptions, also achieves optimal error probability, further justifying it as a method for estimating the change point in our algorithm. We note that all theoretical results presented in this work consider the ELBO defined in (17) without the predictive expected log-likelihood regularizer term. A theoretical analysis of this regularizer is left for future work.

### 4.5.1 Convergence of Training Algorithm to a Stationary Point

To prove that our algorithm converges to a stationary point, we need to make a few assumptions about the efficiency of the updates at each iteration of the algorithm. Mainly, we assume that both model updates and change point updates lead to an improvement based on their respective criterion. Mainly, model parameter updates improve the ELBO and change point updates improve the marginal likelihood. We also make the assumption that the inference gap as a result of the variational approximation does not widen after change points are updated. A detailed description of the assumptions can be found in Appendix A.1.1.

In the following theorem, we show that our training algorithm converges to a stationary point of the ELBO – mainly that after each update in the algorithm the ELBO either stays the same or increases in value. We provide a visualization of the result in Fig 3.

**Theorem 1.** *As $E \to \infty$, our algorithm (under maximum likelihood updates for the change points) reaches a stationary point w.r.t. a lower bound on the marginal likelihood, i.e.,*

$$\mathcal{E}_{\boldsymbol{\theta}^{(i)},\boldsymbol{\phi}^{(i)},\nu^{(i)}}(\boldsymbol{x}_{\text{obs}}) \geq \mathcal{E}_{\boldsymbol{\theta}^{(i-1)},\boldsymbol{\phi}^{(i-1)},\nu^{(i-1)}}(\boldsymbol{x}_{\text{obs}})$$

*for all $i \in \mathbb{N}$, where $\mathbb{N}$ denotes the natural numbers.*

*Proof Sketch.* To prove this result, we needed to show that change point updates (which we assume yield an improvement in marginal likelihood) imply an improvement w.r.t. the ELBO as well. The difference between the logarithm of the marginal likelihood can be shown to be a sum of two components: the improvement in the ELBO and the change in accuracy in the variational approximation (based on the KLD between the variational approximation and the true posterior distribution) after change point updates are made. Under the assumption that change point updates do not vastly impact the accuracy of the variational approximation, we directly arrive at the desired result. The detailed proof can be found in Appendix A.2. □

### 4.5.2 Optimality of the Detector

In the following theorem, we provide a theoretical insight into the performance of our online change point update. Specifically, we demonstrate that at each time $t_k$, our update asymptotically achieves the optimal

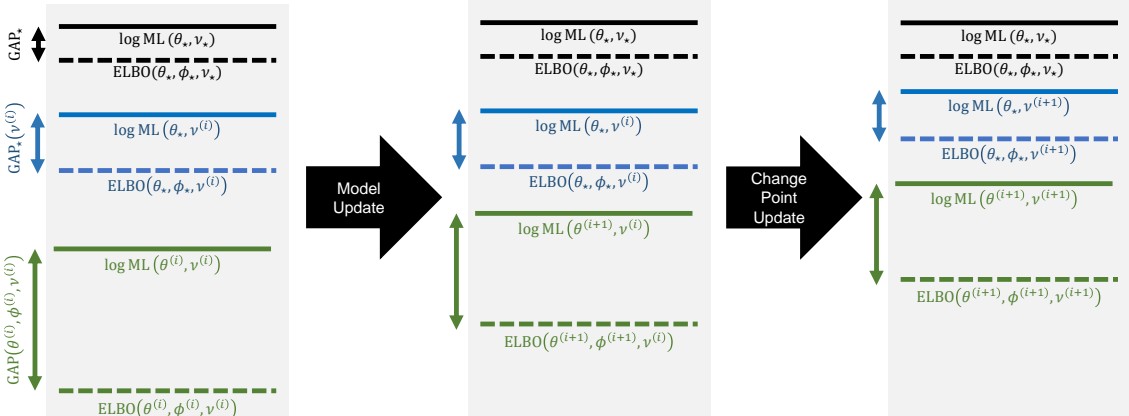

Figure 3: Stationary point convergence based on training algorithm. Under the assumption that change point updates do not widen the inference gap, the result is evident and demonstrated in this diagram.

error probability as the number of sampled trajectories $J$ tends to infinity. This result is significant as it provides a theoretical guarantee for the performance of our proposed method.

**Theorem 2.** *As $J \to \infty$, we have that $\mathbb{P}(\widehat{\Lambda}(\boldsymbol{x}_{t_{1:k}}) \geq \gamma | \mathcal{H}_0) \to \mathbb{P}(\Lambda(\boldsymbol{x}_{t_{1:k}}) \geq \gamma | \mathcal{H}_0)$ and $\mathbb{P}(\widehat{\Lambda}(\boldsymbol{x}_{t_{1:k}}) < \gamma | \mathcal{H}_1) \to \mathbb{P}(\Lambda(\boldsymbol{x}_{t_{1:k}}) < \gamma | \mathcal{H}_1)$.*

*Proof Sketch.* We begin by showing that as $J \to \infty$, the likelihoods $\widehat{p}^J(\boldsymbol{x}_{t_{1:k}} | \nu = t)$ and $\widehat{p}^J(\boldsymbol{x}_{t_{1:k}} | \nu > t)$ converge almost surely to $p(\boldsymbol{x}_{t_{1:k}} | \nu = t)$ and $p(\boldsymbol{x}_{t_{1:k}} | \nu > t)$, respectively. This is achieved by applying standard convergence results of bootstrap particle filters (BPFs). The continuous mapping theorem then implies that the likelihood ratio $\widehat{\Lambda}(\boldsymbol{x}_{t_{1:k}})$ converges to $\Lambda(\boldsymbol{x}_{t_{1:k}})$ almost surely. We then demonstrate that our test achieves the optimal error probability. The detailed proof can be found in the Appendix A.3. □

**Remark on rate of convergence**: With a little abuse of notation, let $\boldsymbol{\mu}(\boldsymbol{x}_{t_{1:k}})$ and $\boldsymbol{\Sigma}(\boldsymbol{x}_{t_{1:k}})$ the mean and covariance matrix of $\boldsymbol{x}_{t_{1:k}}$, respectively. By standard results in Monte Carlo, we have that $\sqrt{J}(\boldsymbol{x}_{t_{1:k}} - \boldsymbol{\mu}(\boldsymbol{x}_{t_{1:k}}))$ converges in distribution to $\mathcal{N}(0, \boldsymbol{\Sigma}(\boldsymbol{x}_{t_{1:k}}))$. Since we sample $J$ independent trajectories of $\boldsymbol{x}_{t_{1:k}}$ in the BPF, from the convergence of the delta method (Fisher, 1925), we have that the estimate $\widehat{\Lambda}^J(\boldsymbol{x}_{t_{1:k}})$ converges to the ground truth $\Lambda(\boldsymbol{x}_{t_{1:k}})$ at the rate of $\frac{1}{\sqrt{J}}$.

## 4.6 Practical Considerations

In this section, we highlight several important aspects to consider in order to ensure success training of the `CP-SDEVAE` algorithm.

**Model architecture:** Our model architecture comprises several key components designed to effectively capture and process time-series data. The encoder utilizes an LSTM network, which is well-suited for sequential data processing. For scenarios involving irregularly sampled time-series, an alternative approach such as a neural CDE could be considered. The decoder is implemented as a fully connected network, providing flexibility in output generation. The core of the model lies in the latent SDE components. Both the drift and diffusion networks of the latent SDE are implemented as fully connected networks with LipSwish activation functions. This design choice introduces an important tradeoff: while more complex drift and diffusion networks can potentially capture more intricate dynamics, they tend to reduce the meaningfulness of detected change points. This phenomenon was observed in our ablation study conducted on both real and synthetic data, as detailed in Section C of the Appendix. For the SDE solver, we employ the Euler-Maruyama. Although we experimented with alternative approaches based on the adjoint sensitivity method, we found no

significant performance differences, leading us to favor the simpler Euler method for its efficiency and ease of implementation.

**Optimizer and stochastic weight averaging:** Our optimization strategy is carefully crafted to ensure robust model training. We utilize the Adam optimizer with a learning rate of $1 \times 10^{-4}$ and a weight decay of $1 \times 10^{-4}$. The training process continues for a maximum of $E = 10000$ epochs or until convergence is reached, as determined by the ELBO loss. To enhance training stability, we incorporate stochastic weight averaging, a technique that has shown promise in previous work on training neural SDEs, such as the SDEGAN approach.

**Initialization of change points:** The initialization of change points plays a crucial role in model performance. We explored two methods: random initialization and initialization based on mean shift using the ruptures library in Python (Truong et al., 2020). Our findings strongly favor the latter approach, as the model exhibits sensitivity to poorly initialized change points. The ruptures library provides a more informed starting point, leading to improved overall performance.

To further enhance the robustness of our change point detection, we implement a warm-start period of $E = 50$ epochs before making any change point updates in the training process. This warm-start period is essential because the accuracy of change point detection is intrinsically linked to the overall model performance. Mismatches in model parameters can lead to degradation in both the maximum likelihood estimation and detection-based approaches for estimating change points. By allowing the model to stabilize initially, we mitigate these potential issues and improve the reliability of our change point estimates.

**Detection threshold:** For a given threshold $\gamma$, the fast detection-based update corresponds to a certain level of tolerance for false alarms. In online settings, it's crucial to set this threshold before deploying the detection algorithm. Much of the literature on sequential testing frameworks focuses on calibrating this threshold for various statistical models to meet specific tolerances for false alarm probabilities. However, the focus of this work is on using the detector to estimate the change point in an offline manner. In this context, the threshold can be seen as a hyperparameter of the `CP-SDEVAE` model, which can be tuned to enhance the quality of generative performance. It's important to note that the detection threshold introduces a trade-off. A larger value of $\gamma$ means that a change point will only be detected in the event of a more extreme distributional shift. Conversely, a smaller value of $\gamma$ increases the likelihood of detecting a change point in response to minor and possibly insignificant changes.

**Extension to multiple change points:** Our proposed mathematical formulation provides a method to incorporate a single change point in modeling neural SDEs. To extend to $D$ change points, $\nu_1, \ldots, \nu_D$, a variety of approaches can be used. For the greedy approach based on maximum likelihood, if there are multiple change points, one can update each change point $\nu_d$ by maximizing the marginal likelihood, holding all other change points and the model parameters fixed to their most recently updated values:

$$\widehat{\nu}_d = \underset{\widehat{\nu}_{d-1} \leq t \leq \widehat{\nu}_{d+1}}{\arg\max} \; p(X_{\text{obs}} | \nu_d = t, \nu_{-d} = \widehat{\nu}_{-d}),$$

where $\widehat{\nu}_{-d}$ denotes the current estimate of all other change points and we define $\widehat{\nu}_0 = 0$ and $\widehat{\nu}_{d+1} = T$. For the detection-based update, one continues running the detector until the desired number of change points are detected. If the number of detected change points is less than the number of change points assumed in the model, an adaptive threshold can be utilized. With regards to choosing the number of change points, standard model selection approaches, such as cross-validation or marginal likelihood maximization, can be considered. With regards to marginal likelihood maximization, we recall that particle filtering methods can be utilized to obtain an unbiased estimator of the marginal likelihood (i.e., using the estimator in (23)). This particular estimator can be used to compare generalization performance between difference classes of models (i.e., different number of change points). This strategy has been repeatedly employed in Bayesian inference settings (see (Llorente et al., 2023) for a review on the use of the marginal likelihood for model selection).

**Greedy vs. fast update:** In our work, we compare two primary approaches for estimating change points: the greedy approach and the fast (detection-based) approach. The greedy approach proves superior in terms

of accuracy, as it precisely determines the change points that maximize the marginal likelihood. For a single change point, this method tests $K$ hypotheses, each requiring $\mathcal{O}(T)$ propagation steps in the particle filter. When dealing with multiple change points $(1 < D < K)$, the computational complexity increases significantly, with the number of hypotheses to be tested growing to $\mathcal{O}\left(\binom{K}{D}\right)$. As previously discussed, the number of hypotheses needed to be tested at each epoch can be reduced using coordinate-ascent style updates; however, this does not solve the long time horizon issue.

In contrast, the fast approach, while potentially less accurate, offers significant computational advantages. It requires only a single $\mathcal{O}(T)$ propagation step through the particle filter, regardless of the number of change points. This makes it particularly suitable for scenarios involving long time-series with multiple change points, where the greedy approach may become computationally infeasible.

The choice between these approaches ultimately depends on the specific characteristics of the data being analyzed, including the length of the time-series and the number of time-series samples. For shorter time-series or when computational resources allow, the greedy approach provides the most accurate results. However, for longer time-series or when dealing with large datasets, the fast approach offers a practical alternative that balances accuracy with computational efficiency.

## 5    Experiments

Here, we present numerical experiments to verify the validity of the proposed `CP-SDEVAE` model. To that end, we conduct two different types of experiments. First, we conduct experiments on synthetic data generated from an Ornstein-Uhlenbeck (OU) process. We use the OU process experiment to compare different variants of the `CP-SDEVAE` method (e.g., with/without change points, MLE-based change point updates vs. detection-based change point updates). We also use this dataset as a means to conduct basic ablations to understand the effect of different hyperparameters and the impact of the proposed predictive negative log-likelihood regularizer. The description and results of the ablation studies can be found in the Appendix C. For all methods, we use a detection threshold of $\gamma = 0$ for the log-likelihood ratio as a means to detect the change point.

### 5.1    Toy Data

We consider a synthetic univariate time-series dataset generated from an OU process. In the first example, we compare the generative performance of our proposed `SDEVAE` model with the `LatentSDE` model described in Section 3.1. In this example, we assume the same switching OU process that is utilized for our ablation studies in Appendix C.In second example, we compare different variants of our proposed approach for an OU process with a single change point. In the third example, we test the robustness of the proposed method by introducing multiple change points.

#### 5.1.1    OU Process with Single Change Point (Slow Change)

We consider time-series generated from a switching OU process that is the solution to the following SDE

$$dX_t = \theta_0(\mu_0 - X_t)dt + \sigma_0 dW_t, \qquad t \in (0, \nu]$$
$$dX_t = \theta_1(\mu_1 - X_t)dt + \sigma_1 dW_t, \qquad t \in (\nu, T],$$

where we consider the parameter settings $\theta_0 = 0.05$, $\mu_0 = 4$, $\sigma_0 = 0.15$, $\theta_1 = 0.03$, $\mu_1 = -2$, $\sigma_1 = 15$, a change point of $\nu = 25$, and a time-horizon of $T = 50$. We assume that for initial state $X_0$ is Gaussian distributed with mean 1 and variance 0.01. Using an Euler solver with step-size $\Delta_t = 1$ for all $t$, we simulate $N = 500$ trajectories to construct a time-series dataset. Our goal is to compare the performance of the `LatentSDE` model and `SDEVAE` under different settings of the observation noise.[2] Similar to the `LatentSDE` implementation, we utilize a GRU-based encoder and 2 layer MLP with 100 hidden units for the latent drift, latent diffusion, and decoder networks.

---

[2]The implementation of the `LatentSDE` model utilized is based on an implementation found in the `torchsde` library at `https://github.com/google-research/torchsde` for modeling a Lorenz attractor. We adopted the implementation into our codebase and utilized an analogous architecture for fair comparison.

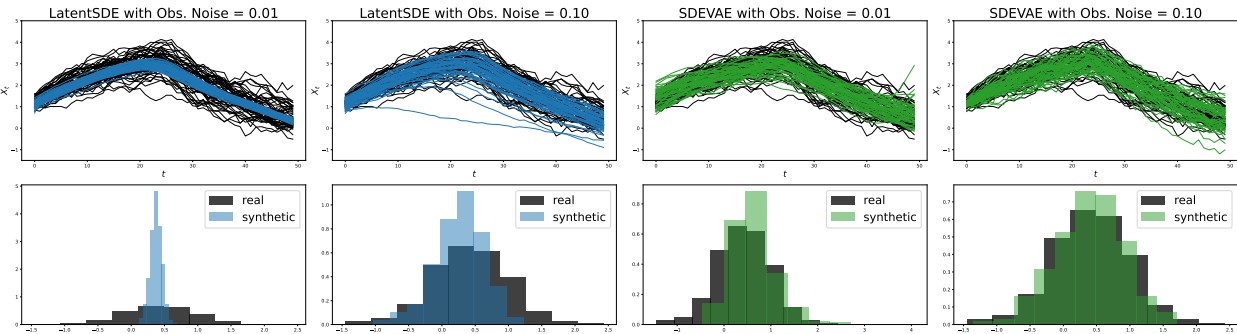

Figure 4: Comparison of generative performance of `LatentSDE` and `SDEVAE` (*ours*) models for different observation noise settings.

We summarize the results in Figure 4. As evidenced by the figure, in the case of low observation noise of 0.01, the `LatentSDE` model underestimates both the noise of the time-series, with smooth trajectories that approximate that capture the mean of the time-series dataset well. When increasing the observation noise to 0.10, although the marginal distribution is better captured, the trajectories are still too smooth to accurately capture the distribution of the observed dataset. In contrast, `SDEVAE` is able to capture the trajectories well under both noise assumptions, with the observation noise of 0.10 providing more realistic samples.

### 5.1.2  OU Process with Single Change Point (Sharp Change)

We consider the same SDE as the previous experiments, except with parameter settings $\theta_0 = 0.2$, $\mu_0 = 4$, $\sigma_0 = 1$, $\theta_1 = 0.5$, $\mu_1 = -4$, $\sigma_1 = 1$, a change point of $\nu = 25$, and a time-horizon of $T = 50$. Under this parameterization of the switching OU process, the distribution shift is more sharp (since the reversion parameters $\theta_0$ and $\theta_1$ are larger). We assume that for initial state $X_0$ is Gaussian distributed with mean 3 and variance 1. Using an Euler solver with step-size $\Delta_t = 1$ for all $t$, we simulate $N = 100$ trajectories to construct a time-series dataset. For each baseline model, we standardize the dataset using the global mean and variance taken across all time-series. For the baselines in this experiment, we consider four different variants of our method: (1) `CP-SDEVAE` assuming no change points (which we also refer to as `SDEVAE`); (2) `CP-SDEVAE` assuming a single change point with maximum likelihood-based change point updates; (3) `CP-SDEVAE` assuming a single change point with detection-based change point updates; and (4) `CP-SDEVAE` assuming two change points with detection-based ML updates. For each of the methods, we assume the following hyperparameter settings: for the encoder architecture with a 2-layer fully-connected neural network with standard ReLU activation functions; the latent dimension of the SDE is assumed to be 32; for all latent drift/diffusion functions, we use 2-layer fully-connected neural network with LipSwish activations; for the decoder network, we use a 1-layer fully-connected network with ReLU activations; we use the Adam optimizer with a weight decay of $1 \times 10^{-4}$ and $J = 5$ trajectories for each MC estimator of the ELBO. As previously mentioned, we utilize this example as a means to conduct an ablation study to test the effectiveness of different components of our model. For more information about the parameter settings of the ablation study and the key finds, please see Appendix C.

**Results:**  A summary figure showing the results of the generated time-series from each model (along with the detected change point) is shown in Figure 5. As can be demonstrated from Figure 5, the change point variants of the proposed approach outperform the variant without any change points assumed. This is evident by looking at the ELBO metric shown in the title of each subfigure, where the change point based approaches achieve superior value, which indicates that the generated dataset with the change point variants demonstrate a higher degree of realism. Moreover, an interesting point to add is that overparameterization in terms of the number of change points does not impact the model's ability to capture the dataset; however, it leads to over representation in terms of the number of change points. We can see that when a second change point is assumed for the `CP-SDEVAE`, the second detected change point does not actually reflect a shift in distribution. Lastly, we point out that both the MLE-based change point update and the detection-based change point

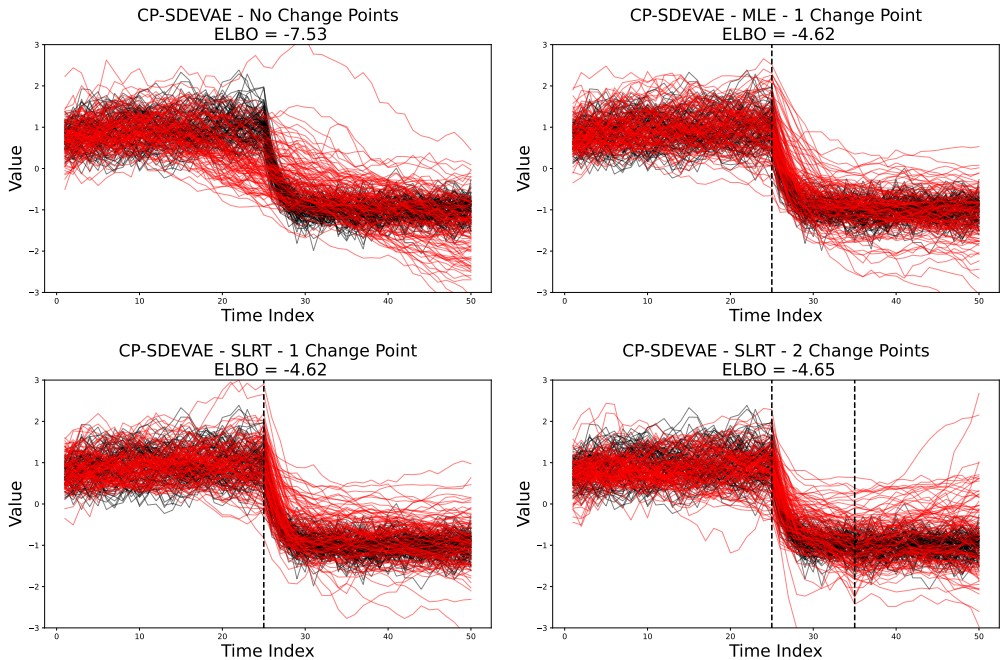

Figure 5: Time-series plots for each of the baseline methodologies.

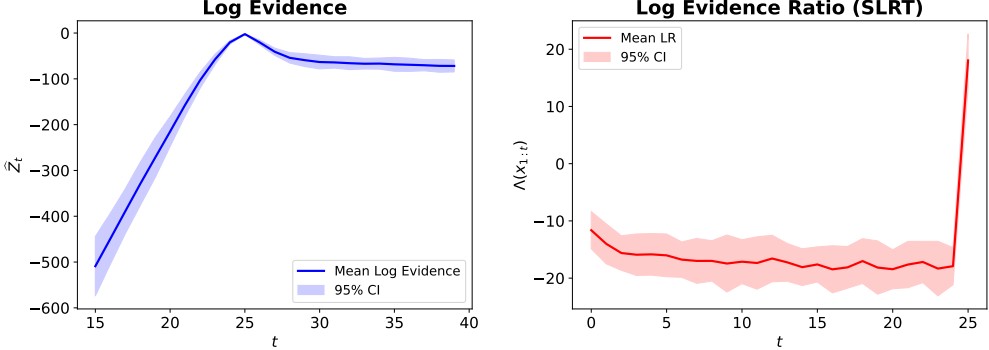

Figure 6: Comparison of log-likelihood ratio for detection-based change point updates and log evidence for MLE-based change point updates.

update lead to the same overall detected change point in the algorithm. As can be seen in Figure 6, the log marginal likelihood achieves its maximum value at the true change point value. For the detection-based update, we track the log-evidence ratio remains relatively stable until we approach the change point value. It is clear that for this example of distribution shift, both approaches are easily able to identify the change point.

### 5.1.3 OU Process with Multiple Change Points

Consider a time-series generated from a switching OU process that is the solution to the following SDE:

$$
\begin{aligned}
dX_t &= \theta_0(\mu_0 - X_t)dt + \sigma_0 dW_t, & t \in (0, \nu_1] \\
dX_t &= \theta_1(\mu_1 - X_t)dt + \sigma_1 dW_t, & t \in (\nu_1, \nu_2], \\
dX_t &= \theta_2(\mu_2 - X_t)dt + \sigma_2 dW_t, & t \in (\nu_2, T],
\end{aligned}
$$

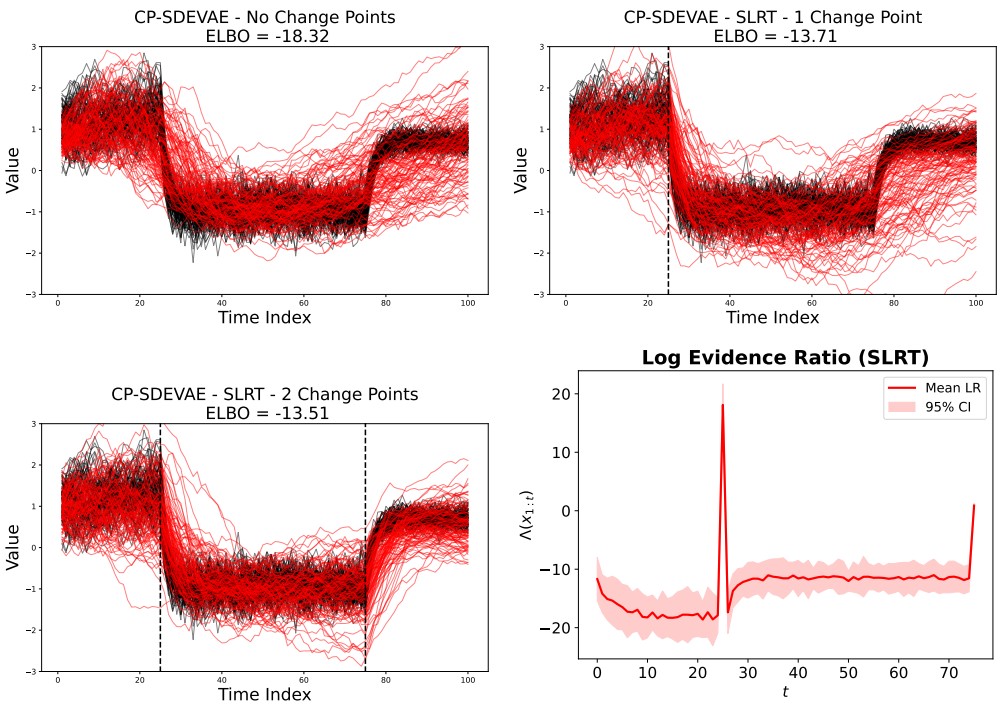

Figure 7: Time-series plots for each of the baseline methodologies for the multi-change example. The log-likelihood ratio statistic is also shown as a function of time.

where we consider the parameter settings $\theta_0 = 0.2$, $\mu_0 = 4$, $\sigma_0 = 1$, $\theta_1 = 0.5$, $\mu_1 = -4$, $\sigma_1 = 1$, $\theta_2 = 0.5$, $\mu_2 = 2$, $\sigma_2 = 0.5$, change point values of $\nu_1 = 25$ and $\nu_2 = 75$, and a time-horizon of $T = 100$. We utilize the same hyperparameter settings as the previous example. Using an Euler solver with step-size $\Delta_t = 1$ for all $t$, we simulate $N = 100$ trajectories to construct a time-series dataset. Here, we also test three different variants of our method: (1) `CP-SDEVAE` assuming no change points; (2) `CP-SDEVAE` assuming one change point with with detection-based change point updates; (3) `CP-SDEVAE` assuming two change points with detection-based change point updates.

**Results:** A summary figure showing the results of the generated time-series from each model (along with the detected change point) and the the log-likelihood ratio over time is shown in Figure 7. As we can see from Figure7, the `CP-SDEVAE` variants which assume a change point are able to better capture the shift in the distribution, where assuming a single change point shows better performance than assuming no change points at all and assuming two change points shows the best performance overall. We can also see that the log-likelihood ratio flips signs exactly at the location of the change points, demonstrating the methodologies' ability to reflect both small and large shifts in the distribution.

## 5.2   Real Data Experiments

We conducted experiments utilizing five baseline models on four datasets as benchmarks to evaluate the proposed models in the time-series data generation task. Then, we evaluated the generated data based on three metrics, including marginal distribution, indistinguishability, and predictiveness. The evaluation results are shown in Table 1.

### 5.2.1   Baseline Models and Datasets

We included five representative time-series models as comparison baselines: TimeVAE (Desai et al., 2021), TimeGAN (Yoon et al., 2019), QuantGAN (Wiese et al., 2020), LS4 (Zhou et al., 2023), and SDEGAN

(Li et al., 2020; Kidger et al., 2021). Each model brings a distinct approach to the generation of synthetic time-series data.

Specifically, TimeVAE leverages a variational autoencoder structure with convolutional layers to capture the temporal dynamics and dependencies inherent in time-series. TimeGAN, QuantGAN, and SDEGAN are built on the generative adversarial network framework to maintain temporal correlations within the data. The LS4 model addresses the challenge of capturing long-term dependencies within time-series by introducing a state space ordinary differential equations (ODE) framework for latent variables. We note that our model is denoted by `CP-SDEVAE`$_L$, where $L$ denotes the number of change points assumed.

We ran experiments on four datasets, including the S&P500 prices, S&P500 intraday prices, cryptocurrency prices, and air quality measurements. Supplementary Material provides a detailed description of these datasets. Time-series samples in all datasets have a fixed length of 120, while different datasets contain different numbers of samples. To evaluate the performance of the models, each dataset was split into two subsets, a training and a testing set. This split was conducted following the "80-20" rule that 80% of the samples were randomly selected to form the training set, and the remaining 20% was the testing set.

### 5.2.2 Evaluation Metrics

Our quantitative evaluation framework in this work encompasses three distinct metrics, each targeting a specific aspect of data quality and utility. The metrics include marginal similarity, data indistinguishability via classification, and predictive quality:

- **Marginal Distribution Similarity**: First, we assess the similarity between the marginal distributions of the generated and real time-series data. This evaluation is conducted based on a histogram-based difference. It calculates the density histogram of the real data, which serves as a reference for synthetic data. Here, we fixed the number of histogram bins for density calculation. Then, the comparison calculates the absolute difference in densities across corresponding distributions. Specifically, the final marginal score is obtained by averaging the discrepancies across all bins and data dimensions. The minimum value of the marginal score would be zero, indicating a perfect match in marginal distributions between real and synthetic data, while the upper bound of the marginal score cannot be directly determined without any constraint. Thus, the smaller the marginal score, the better the capability of the proposed model to replicate the distributional properties of the real data. We denote this score as "Marginal ($\downarrow$)" in our experiments.

- **Synthetic Data Indistinguishability**: The second metric evaluates the indistinguishability of synthetic data from real data through a classification approach (Yoon et al., 2019). A downstream classifier, built upon the structured state space model, or the S4 model mentioned in (Zhou et al., 2023), is trained to differentiate between synthetic and real data. Specifically, the S4 model maps the input time-series data to a higher dimensional space via a linear encoder to capture the temporal dynamics within the data and produces classification scores via a linear decoder. Real and generated synthetic data with the same sample size are concatenated to form a unified dataset, which is then split into training and testing sets. Binary labels are assigned to indicate the source of each time-series. The classifier is trained on the labeled dataset using binary cross-entropy (BCE) loss to distinguish real and synthetic samples. The outcome of this evaluation represents the model's accuracy in classifying synthetic versus real data in the testing set. We took the absolute value of the accuracy after subtracting 0.5, which is the value that indicates that the model cannot distinguish real and synthetic samples. Thus, lower classification scores indicate a higher degree of indistinguishability, suggesting that the synthetic data closely mimics the real data. This metric directly addresses the proposed model's capability of generating data that is qualitatively indistinguishable from real data. We denote this score as "Classification ($\downarrow$)" in our experiments.

- **Synthetic Data Predictive Quality**: The third aspect of evaluation is the predictive quality of synthetic data. This metric provides insights into how well synthetic data can be a proxy for real data in prediction tasks. We utilized the S4 model with the same model structure as the predictor but enabled it to predict corresponding future values given time-series data. Unlike training on a

combined dataset to evaluate the classification quality, the predictor was trained only on the synthetic data and tested on the real data. The prediction accuracy is calculated using the mean squared error (MSE) between the prediction results and the actual values. Thus, the lower the score, the better the proposed model's capability to generate high-fidelity synthetic data. We denote this score as "Prediction ($\downarrow$)" in our experiments.

### 5.2.3 Results and Discussion

The experimental results, presented in Table 1, demonstrate that `CP-SDEVAE` outperforms most baseline models, even without assuming any change points ($L = 0$) in real datasets. Among the competitors, the LS4 model emerges as the closest rival, with `CP-SDEVAE` achieving comparable performance across nearly all datasets. Notably, `CP-SDEVAE` significantly outshines its GAN-based counterpart, SDE-GAN, even in scenarios without change points. This superior performance is attributed to the greater stability and efficacy of VAE-based generative models compared to GAN-based models, particularly with smaller datasets. For the S&P 500 datasets and the cryptocurrency datasets, the baselines approaches such as a TimeVAE, TimeGAN, QuantGAN and SDEGAN tend to perform relatively worse than LS4 and `CP-SDEVAE`. This is due to the fact that those datasets contain stronger distributional shifts than the Air Quality dataset, which has the most comparable results across all methods. As a note, the Air Quality dataset demonstrates repeated fluctuations; making it a poor candidate for SDE modeling in the first place.

Further discussion is warranted on the necessity of incorporating change points in our model. The architecture used in these experiments features an underlying latent SDE represented by an MLP with 3 hidden layers and 64 hidden units, which is adept at capturing nonlinear shifts in drift and diffusion. In the datasets tested, shifts occur more gradually rather than abruptly, reducing the importance of change points for some datasets. However, the utility of change points becomes evident with the S&P 500 data, where employing $L = 2$ change points significantly improved the classification score. To explore this hypothesis further, we conducted ablation studies on the S&P 500 sectors dataset, varying the number of layers and hidden units, as shown in Table 3 in Section C of the Appendix. These studies reveal that as the complexity of the underlying neural SDE model increases, the need for change points to enhance performance diminishes.

Finally, we compare the performance of `CP-SDEVAE` and LS4, as LS4 remained competitive across all datasets. Unlike `SDEVAE` and `CP-SDEVAE`, which are based on latent SDEs, LS4 employs a latent ODE with a structured state-space model. By leveraging a convolutional representation, LS4 bypasses explicit hidden state evaluations, significantly improving computational efficiency. Notably, LS4 has been shown in (Zhou et al., 2023) to perform well on stochastic data with sharp transitions. However, `CP-SDEVAE` offers two key advantages over LS4. First, if the application requires segmenting time-series into different regimes, `CP-SDEVAE` provides this capability, whereas LS4 assumes a single underlying model. This segmentation allows for learning distinct temporal dynamics and generating diverse time-series patterns during inference. Second, for interpretability, `CP-SDEVAE` explicitly models the latent diffusion term, whereas LS4 does not separately approximate it. This feature can be beneficial for analyzing the role of stochasticity in time-series evolution.

## 6 Conclusions

In this work, we introduced a novel formulation of neural SDEs within the VAE framework, enabling seamless integration of change points using principles from maximum likelihood estimation and classical change point detection theory. We presented theoretical results demonstrating the convergence of change points and VAE model parameters to a stationary point, as well as the optimality of the Bayesian detector used in our method, which minimizes the probability of error in the test. We evaluated our algorithm on various real-world datasets, finding that our generative model achieves competitive performance compared to other deep generative models for time-series data and effectively captures distributional shifts.

### Future Work

There are various research directions that can be considered as future work. Importantly, one limitation of the `CP-SDEVAE` method is that it requires that the number of change points be specified a priori. While selecting

Table 1: Time Series Experiments Summary Table

| Data (dimension) | Metric | Baseline Models | | | | | Proposed Models | | |
|---|---|---|---|---|---|---|---|---|---|
| | | TimeVAE | TimeGAN | QuantGAN | LS4 | SDEGAN | CP-SDEVAE$_0$ | CP-SDEVAE$_1$ | CP-SDEVAE$_2$ |
| S&P 500 (506, 120) | Marginal ↓ | $0.089 \pm 0.002$ | $0.080 \pm 0.015$ | $0.122 \pm 0.065$ | $\mathbf{0.013 \pm 0.001}$ | $0.076 \pm 0.019$ | $0.030 \pm 0.004$ | $0.027 \pm 0.006$ | $0.026 \pm 0.006$ |
| | Classification ↓ | $0.300 \pm 0.107$ | $0.261 \pm 0.130$ | $0.500 \pm 0.000$ | $0.105 \pm 0.071$ | $0.285 \pm 0.071$ | $0.149 \pm 0.048$ | $0.168 \pm 0.059$ | $\mathbf{0.061 \pm 0.035}$ |
| | Prediction ↓ | $0.617 \pm 0.273$ | $0.365 \pm 0.475$ | $0.582 \pm 0.422$ | $\mathbf{0.045 \pm 0.004}$ | $0.046 \pm 0.003$ | $0.055 \pm 0.009$ | $0.060 \pm 0.011$ | $0.057 \pm 0.016$ |
| S&P 500 intraday (500, 120) | Marginal ↓ | $0.086 \pm 0.002$ | $0.036 \pm 0.006$ | $0.100 \pm 0.032$ | $0.016 \pm 0.004$ | $0.091 \pm 0.023$ | $0.012 \pm 0.002$ | $0.012 \pm 0.004$ | $\mathbf{0.010 \pm 0.003}$ |
| | Classification ↓ | $0.475 \pm 0.032$ | $0.485 \pm 0.012$ | $0.500 \pm 0.000$ | $0.070 \pm 0.033$ | $0.435 \pm 0.070$ | $\mathbf{0.045 \pm 0.037}$ | $0.070 \pm 0.040$ | $0.060 \pm 0.060$ |
| | Prediction ↓ | $1.634 \pm 0.438$ | $1.448 \pm 1.022$ | $3.257 \pm 2.330$ | $\mathbf{0.144 \pm 0.010}$ | $0.886 \pm 1.422$ | $0.199 \pm 0.030$ | $0.195 \pm 0.027$ | $0.166 \pm 0.022$ |
| Crypto currency (12, 120) | Marginal ↓ | $0.120 \pm 0.012$ | $\mathbf{0.102 \pm 0.012}$ | $0.151 \pm 0.022$ | $0.122 \pm 0.033$ | $0.178 \pm 0.033$ | $0.107 \pm 0.015$ | $0.104 \pm 0.016$ | $0.107 \pm 0.011$ |
| | Classification ↓ | $0.300 \pm 0.245$ | $0.300 \pm 0.245$ | $0.200 \pm 0.245$ | $0.200 \pm 0.245$ | $0.300 \pm 0.245$ | $0.200 \pm 0.245$ | $\mathbf{0.100 \pm 0.200}$ | $\mathbf{0.100 \pm 0.200}$ |
| | Prediction ↓ | $\mathbf{0.073 \pm 0.022}$ | $1.180 \pm 0.962$ | $1.344 \pm 1.222$ | $0.105 \pm 0.053$ | $0.441 \pm 0.446$ | $0.559 \pm 0.404$ | $0.507 \pm 0.414$ | $0.492 \pm 0.268$ |
| Air quality (60, 120) | Marginal ↓ | $0.063 \pm 0.003$ | $0.040 \pm 0.004$ | $0.133 \pm 0.049$ | $\mathbf{0.034 \pm 0.006}$ | $0.109 \pm 0.015$ | $0.057 \pm 0.024$ | $0.044 \pm 0.005$ | $0.056 \pm 0.012$ |
| | Classification ↓ | $\mathbf{0.220 \pm 0.098}$ | $0.260 \pm 0.150$ | $0.500 \pm 0.000$ | $\mathbf{0.220 \pm 0.098}$ | $0.380 \pm 0.098$ | $\mathbf{0.220 \pm 0.160}$ | $0.260 \pm 0.150$ | $\mathbf{0.220 \pm 0.160}$ |
| | Prediction ↓ | $0.723 \pm 0.308$ | $1.189 \pm 0.722$ | $4.696 \pm 2.175$ | $\mathbf{0.657 \pm 0.335}$ | $3.845 \pm 4.274$ | $1.272 \pm 0.582$ | $0.894 \pm 0.232$ | $0.938 \pm 0.230$ |

among multiple candidate models is straightforward (using model selection techniques like cross-validation or marginal likelihood maximization), it can be computationally expensive. An interesting extension of this work would consider that the number of change points need not be specified, and furthermore, would not require that all observed time-series share the same change point (i.e., heterogeneous change points). Another interesting direction for future work is to theoretically understand why the predictive log-likelihood regularizer improves the performance of the generative model (i.e., the model's noise estimation). Lastly, the current version of the `CP-SDEVAE` model uses a numerical algorithm that alternately optimizes the model parameters and the change points, due to the fact that the joint optimization of the model parameters (which are real valued) and the change points (which are discrete valued). While this work showed that theoretically the numerical algorithm converges to a stationary point of the ELBO, it introduces the additional computational burden of having to use particle filtering to resolve the change point updates, which can be expensive in practice. A practically useful extension of this work would allow for differentiable change point modeling, possibly using temporal point processes (see Koley et al. (2023)). This would enable joint optimization of the model parameters and the change points, which comes with standard convergence guarantees of stochastic optimization algorithms.

## Broader Impact Statement

The presented work proposes an algorithm for modeling change points in time series data, which can potentially be applied in a variety of disciplines including ecology, finance, and robotics, among others. The proposed work can have a positive impact on society if used responsibly, as it offers the capability to model distribution shifts in time series data. We highlight that the proposed work can also be incorporated into risk forecasting systems and potentially provide more robustness in forecasting uncertainty when distribution shifts may occur. Importantly, the scenario considered in this work addressed change points modeled agnostic of exogenous

drivers (i.e., covariate information). In applications where distribution shifts are known to be driven by certain covariates, responsible usage of this work would entail incorporating this covariate information into the generative model in order to detect and model change points.

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

# A  Appendix

## A.1  Theoretical Proofs

In this part of the appendix, we provide proofs for the propositions presented in the paper. For convenience, we restate our assumptions for each proposition and provide a short justification of the assumption.

### A.1.1  Assumptions for Convergence to a Stationary Point

To prove that the training algorithm for `CP-SDEVAE` converges to a stationary point, we made a few assumptions (which we find reasonable). We state these assumptions in the following and provide a short justification:

**Assumption 1.** *[Model update phase always leads to an ELBO improvement] Let $\boldsymbol{\theta}' \leftarrow \boldsymbol{\theta}$ and $\boldsymbol{\phi}' \leftarrow \boldsymbol{\phi}$ denote the model and variational parameter updates obtained from numerically maximizing $\mathcal{E}_{\boldsymbol{\theta},\boldsymbol{\phi},\nu}(\boldsymbol{x}_{\mathrm{obs}})$. We assume that for fixed change point $\nu$, our model parameter updates lead to an improvement in the ELBO:*

$$\Delta\mathcal{E}(\boldsymbol{\theta}',\boldsymbol{\theta},\boldsymbol{\phi}',\boldsymbol{\phi}) \triangleq \mathcal{E}_{\boldsymbol{\theta}',\boldsymbol{\phi}',\nu}(\boldsymbol{x}_{\mathrm{obs}}) - \mathcal{E}_{\boldsymbol{\theta},\boldsymbol{\phi},\nu}(\boldsymbol{x}_{\mathrm{obs}}) \geq \delta_{\mathrm{m}}, \quad \delta_{\mathrm{m}} \geq 0$$

**Justification:**  The model update phase aims to update the model parameters $\boldsymbol{\theta}$ and the variational approximation parameters $\boldsymbol{\phi}$ by maximizing the ELBO for a fixed change point. Standard stochastic optimization approaches guarantee that these updates improve the objective function (in expectation). Therefore, it is a reasonable assumption.

**Assumption 2.** *[Change point update phase always leads to a marginal likelihood improvement] Let $\nu' \leftarrow \nu$ denote the change point update obtained from numerically maximizing the marginal likelihood: $p_{\boldsymbol{\theta}}(\boldsymbol{x}_{\mathrm{obs}}|\nu = t)$. We assume that for fixed model parameters $\boldsymbol{\theta}$, our change point updates lead to an improvement in the marginal likelihood:*

$$\Delta\mathcal{L}(\nu',\nu) \triangleq \log p_{\boldsymbol{\theta}}(\boldsymbol{x}_{\mathrm{obs}}|\nu = \nu') - \log p_{\boldsymbol{\theta}}(\boldsymbol{x}_{\mathrm{obs}}|\nu = \nu) \geq \delta_{\mathrm{cp}}, \quad \delta_{\mathrm{cp}} \geq 0$$

**Justification:**  With the maximum likelihood approach, the change point updates are made to maximize $\log p_{\boldsymbol{\theta}}(\boldsymbol{x}_{\mathrm{obs}}|\nu = t)$ w.r.t. the change point $\nu$. Since we assume that the change point occurs at a sample time $t \in \mathcal{T}$ this is a discrete optimization problem that can numerically be solved with Bayesian filtering approaches. In this work, we utilized the bootstrap particle filter (BPF), which provides an estimator of the marginal likelihood that converge almost surely to the true marginal likelihood. Therefore, the greedy change point update guarantees the assumption in the limit infinite particles used in the BPF to form the estimator of the marginal likelihood (justified by the strong law of large numbers).

**Assumption 3.** *[Change point updates do not widen the inference gap] Let $\nu' \leftarrow \nu$ denote the change point update obtained from numerically maximizing the marginal likelihood: $p_{\boldsymbol{\theta}}(\boldsymbol{x}_{\mathrm{obs}}|\nu = t)$. We assume for fixed variational parameter $\boldsymbol{\phi}$, the KLD between the variational approximation and the posterior distribution of the initial state $\boldsymbol{z}_0$ is smaller than the improvement in the marignal likelihood:*

$$\mathcal{D}_{\mathrm{KL}}(q_{\boldsymbol{\phi}}(\boldsymbol{z}_0|\boldsymbol{x}_{\mathrm{obs}})\|p_{\boldsymbol{\theta},\nu'}(\boldsymbol{z}_0|\boldsymbol{x}_{\mathrm{obs}})) - \mathcal{D}_{\mathrm{KL}}(q_{\boldsymbol{\phi}}(\boldsymbol{z}_0|\boldsymbol{x}_{\mathrm{obs}})\|p_{\boldsymbol{\theta},\nu}(\boldsymbol{z}_0|\boldsymbol{x}_{\mathrm{obs}})) \leq \delta_{\mathrm{KL}}, \quad \delta_{\mathrm{cp}} \geq \delta_{\mathrm{KL}},$$

*where $\delta_{\mathrm{cp}}$ denotes the maximum improvement in the marginal likelihood during the change point update phase.*

**Justification:** Unlike the aforementioned assumptions, this assumption is non-standard and requires proper mathematical justification based on this specific problem setting. Let $\Delta\mathcal{D}_{\mathrm{KL}}(\nu', \nu)$ denote the change in the KLD between the variational approximation and the true posterior. We can manipulate this expression as follows:

$$
\begin{aligned}
\Delta\mathcal{D}_{\mathrm{KL}}(\nu', \nu) &= \mathcal{D}_{\mathrm{KL}}(q_{\boldsymbol{\phi}}(\boldsymbol{z}_0|\boldsymbol{x}_{\mathrm{obs}})\|p_{\boldsymbol{\theta},\nu'}(\boldsymbol{z}_0|\boldsymbol{x}_{\mathrm{obs}})) - \mathcal{D}_{\mathrm{KL}}(q_{\boldsymbol{\phi}}(\boldsymbol{z}_0|\boldsymbol{x}_{\mathrm{obs}})\|p_{\boldsymbol{\theta},\nu}(\boldsymbol{z}_0|\boldsymbol{x}_{\mathrm{obs}})) \\
&= \mathbb{E}_{q_{\boldsymbol{\phi}}}\left[\log\left(\frac{q_{\boldsymbol{\phi}}(\boldsymbol{z}_0|\boldsymbol{x}_{\mathrm{obs}})}{p_{\boldsymbol{\theta},\nu'}(\boldsymbol{z}_0|\boldsymbol{x}_{\mathrm{obs}})}\right)\right] - \mathbb{E}_{q_{\boldsymbol{\phi}}}\left[\log\left(\frac{q_{\boldsymbol{\phi}}(\boldsymbol{z}_0|\boldsymbol{x}_{\mathrm{obs}})}{p_{\boldsymbol{\theta},\nu}(\boldsymbol{z}_0|\boldsymbol{x}_{\mathrm{obs}})}\right)\right] \\
&= \underbrace{\mathbb{E}_{q_{\boldsymbol{\phi}}}\left[\log\left(\frac{q_{\boldsymbol{\phi}}(\boldsymbol{z}_0|\boldsymbol{x}_{\mathrm{obs}})}{q_{\boldsymbol{\phi}}(\boldsymbol{z}_0|\boldsymbol{x}_{\mathrm{obs}})}\right)\right]}_{0} + \mathbb{E}_{q_{\boldsymbol{\phi}}}\left[\log\left(\frac{p_{\boldsymbol{\theta},\nu}(\boldsymbol{z}_0|\boldsymbol{x}_{\mathrm{obs}})}{p_{\boldsymbol{\theta},\nu'}(\boldsymbol{z}_0|\boldsymbol{x}_{\mathrm{obs}})}\right)\right] && \text{(Manipulate logarithm)} \\
&= \mathbb{E}_{q_{\boldsymbol{\phi}}}\left[\log\left(\frac{\frac{p_{\boldsymbol{\theta},\nu}(\boldsymbol{x}_{\mathrm{obs}}|\boldsymbol{z}_0)p(\boldsymbol{z}_0)}{p_{\boldsymbol{\theta},\nu}(\boldsymbol{x}_{\mathrm{obs}})}}{\frac{p_{\boldsymbol{\theta},\nu'}(\boldsymbol{x}_{\mathrm{obs}}|\boldsymbol{z}_0)p(\boldsymbol{z}_0)}{p_{\boldsymbol{\theta},\nu'}(\boldsymbol{x}_{\mathrm{obs}})}}\right)\right] && \text{(Bayes' theorem)} \\
&= \mathbb{E}_{q_{\boldsymbol{\phi}}}\left[\log\left(\frac{p_{\boldsymbol{\theta},\nu'}(\boldsymbol{x}_{\mathrm{obs}})}{p_{\boldsymbol{\theta},\nu}(\boldsymbol{x}_{\mathrm{obs}})}\right)\right] + \mathbb{E}_{q_{\boldsymbol{\phi}}}\left[\log\left(\frac{p_{\boldsymbol{\theta},\nu}(\boldsymbol{x}_{\mathrm{obs}}|\boldsymbol{z}_0)}{p_{\boldsymbol{\theta},\nu'}(\boldsymbol{x}_{\mathrm{obs}}|\boldsymbol{z}_0)}\right)\right] && \text{(Manipulate logarithm)} \\
&= \underbrace{\log\left(\frac{p_{\boldsymbol{\theta},\nu'}(\boldsymbol{x}_{\mathrm{obs}})}{p_{\boldsymbol{\theta},\nu}(\boldsymbol{x}_{\mathrm{obs}})}\right)}_{\delta_{\mathrm{cp}}} + \underbrace{\mathbb{E}_{q_{\boldsymbol{\phi}}}\left[\log\left(\frac{p_{\boldsymbol{\theta},\nu}(\boldsymbol{x}_{\mathrm{obs}}|\boldsymbol{z}_0)}{p_{\boldsymbol{\theta},\nu'}(\boldsymbol{x}_{\mathrm{obs}}|\boldsymbol{z}_0)}\right)\right]}_{\delta_{\mathrm{lr}}}
\end{aligned}
$$

Therefore, to justify our assumption that $\Delta\mathcal{D}_{\mathrm{KL}}(\nu', \nu) \leq \delta_{\mathrm{KL}}$, where $\delta_{\mathrm{cp}} \geq \delta_{\mathrm{KL}}$, we need to justify the inequality $\delta_{\mathrm{lr}} < 0$, which is equivalent to showing that:

$$
\log p_{\boldsymbol{\theta},\nu'}(\boldsymbol{x}_{\mathrm{obs}}|\boldsymbol{z}_0) \geq \log p_{\boldsymbol{\theta},\nu}(\boldsymbol{x}_{\mathrm{obs}}|\boldsymbol{z}_0), \quad \boldsymbol{z}_0 \in \mathbb{R}^{d_z}.
$$

The key difference between this inequality and Assumption 2 is the additional conditioning on the initial state $\boldsymbol{z}_0$. We can write each of these conditional likelihood as integral measures with respect to $\boldsymbol{z}_{\mathrm{obs}}$:

$$
p_{\boldsymbol{\theta},\nu}(\boldsymbol{x}_{\mathrm{obs}}|\boldsymbol{z}_0) = \int \left(\prod_{k=1}^{K} p_{\boldsymbol{\theta}}(\boldsymbol{x}_{t_k}|\boldsymbol{z}_{t_k})p_{\boldsymbol{\theta},\nu}(\boldsymbol{z}_{t_k}|\boldsymbol{z}_{t_{k-1}})\right) d\boldsymbol{z}_{t_1} \cdots d\boldsymbol{z}_{t_K}
$$

We can see that as $K$ grows, the dependence of the latent state $\boldsymbol{z}_0$ gets smaller. This is something we should expect from SDE models, since the dependence of the initial state $\boldsymbol{z}_0$ on the observed trajectory $\boldsymbol{x}_{\mathrm{obs}}$ should be minimal for long sequences. Since the change point updates are made to maximize marginal likelihood, on average we expect that $\log p_{\boldsymbol{\theta},\nu'}(\boldsymbol{x}_{\mathrm{obs}}|\boldsymbol{z}_0)$ be larger than $\log p_{\boldsymbol{\theta},\nu}(\boldsymbol{x}_{\mathrm{obs}}|\boldsymbol{z}_0)$, and therefore it is reasonable to assume $\delta_{\mathrm{lr}} < 0$. This implies that $\delta_{\mathrm{KL}} \leq \delta_{\mathrm{cp}}$ and the assumption is verified. We emphasize that the validity of this assumption is conditional on the fact that the time-series being considered are of sufficient length.

### A.2 Proof for Convergence to a Stationary Point

In this subsection, we prove Theorem 1.

*Proof.* We would like to show that for any $i \in \mathbb{N}$:

$$
\Delta\mathcal{E}(\boldsymbol{\theta}', \boldsymbol{\theta}, \boldsymbol{\phi}', \boldsymbol{\phi}, \nu', \nu) \triangleq \mathcal{E}_{\boldsymbol{\theta}',\boldsymbol{\phi}',\nu'}(\boldsymbol{x}_{\mathrm{obs}}) - \mathcal{E}_{\boldsymbol{\theta},\boldsymbol{\phi},\nu}(\boldsymbol{x}_{\mathrm{obs}}) \geq 0
$$

We can show this in two steps by showing that:

$$
\mathcal{E}_{\boldsymbol{\theta}',\boldsymbol{\phi}',\nu'}(\boldsymbol{x}_{\mathrm{obs}}) \geq \mathcal{E}_{\boldsymbol{\theta}',\boldsymbol{\phi}',\nu}(\boldsymbol{x}_{\mathrm{obs}}) \geq \mathcal{E}_{\boldsymbol{\theta},\boldsymbol{\phi},\nu}(\boldsymbol{x}_{\mathrm{obs}})
$$

Change Point Update   Model Update

By Assumption 1, if the updates to the model parameters are efficient (for fixed change point $\nu$), then we the first inequality is trivial, i.e.,

$$
\Delta\mathcal{E}(\boldsymbol{\theta}', \boldsymbol{\theta}, \boldsymbol{\phi}', \boldsymbol{\phi}) \geq \delta_m \geq 0 \implies \mathcal{E}_{\boldsymbol{\theta}',\boldsymbol{\phi}',\nu}(\boldsymbol{x}_{\mathrm{obs}}) \geq \mathcal{E}_{\boldsymbol{\theta},\boldsymbol{\phi},\nu}(\boldsymbol{x}_{\mathrm{obs}})
$$

To show the second inequality, we capitalize on Assumption 3 which claims the variational inference gap does not widen after change point updates. The change in the logarithm of the marginal likelihood (pre/post-change point updates) is given by:

$$\Delta\mathcal{L}(\nu',\nu) = \underbrace{(\mathcal{E}_{\boldsymbol{\theta}',\boldsymbol{\phi}',\nu'}(\boldsymbol{x}_{\text{obs}}) - \mathcal{E}_{\boldsymbol{\theta}',\boldsymbol{\phi}',\nu}(\boldsymbol{x}_{\text{obs}}))}_{\Delta\mathcal{E}(\nu',\nu)} +$$

$$\underbrace{(\mathcal{D}_{\text{KL}}(q_{\boldsymbol{\phi}'}(\boldsymbol{z}_0|\boldsymbol{x}_{\text{obs}})\|p_{\boldsymbol{\theta}',\nu'}(\boldsymbol{z}_0|\boldsymbol{x}_{\text{obs}})) - \mathcal{D}_{\text{KL}}(q_{\boldsymbol{\phi}'}(\boldsymbol{z}_0|\boldsymbol{x}_{\text{obs}})\|p_{\boldsymbol{\theta}',\nu}(\boldsymbol{z}_0|\boldsymbol{x}_{\text{obs}})))}_{\Delta\mathcal{D}_{\text{KL}}(\nu',\nu)}$$

It follows from Assumption 2 that $\Delta\mathcal{L}(\nu',\nu) \geq \Delta\mathcal{D}_{\text{KL}}(\nu',\nu)$ and therefore,

$$\Delta\mathcal{E}(\nu',\nu) \geq 0$$

Thus, we conclude that $\Delta\mathcal{E}(\boldsymbol{\theta}',\boldsymbol{\theta},\boldsymbol{\phi}',\boldsymbol{\phi},\nu',\nu) \geq 0$. $\qquad\square$

### A.3 Proof of Asymptotic Optimality of Test Statistic

*Proof.* In the following, we show a simple proof for the case of using a Monte Carlo estimator of the marginal likelihood of the change point. We note; however that this proof can be trivially extended to any estimator that converges almost surely (e.g., particle filtering-based estimators of the marginal likelihood). For brevity in the notation, we also remove the dependency in the discussed distributions on $\boldsymbol{\theta}$ since we assume that it is fixed and the locally optimal value.

Consider a set of i.i.d. samples $\boldsymbol{z}_{0:t}^{(m)} \sim p(\boldsymbol{z}_{0:t}|\nu = \tau)$ for $m = 1, \ldots, M$. By the strong law of large numbers, we have that

$$\lim_{M\to\infty} \frac{1}{M} \sum_{m=1}^{M} p(\boldsymbol{x}_t|\boldsymbol{z}_t^{(m)}, \nu = \tau) = p(\boldsymbol{x}_t|\nu = \tau), \tag{34}$$

almost surely. Consider the function $f(x) = \frac{1}{x}$. The set of discontinuity points $D_g$ of $f(x)$ satisfies $P(x \in D_g) = 0$. By the continuous mapping theorem, we have that

$$\lim_{M\to\infty} \frac{1}{\frac{1}{M} \sum_{m=1}^{M} p(\boldsymbol{x}_t|\boldsymbol{z}_t^{(m)}, \nu = \tau)} = \frac{1}{p(\boldsymbol{x}_t|\nu = \tau)} \tag{35}$$

almost surely. Therefore, by Slutsky's theorem, we can readily deduce that

$$\lim_{M\to\infty} \frac{\frac{1}{M} \sum_{m=1}^{M} p(\boldsymbol{x}_t|\boldsymbol{z}_t^{(m)}|\nu = \tau)}{\frac{1}{M} \sum_{m=1}^{M} p(\boldsymbol{x}_t|\boldsymbol{z}_t^{(m)}|\nu > \tau)} = \frac{p(\boldsymbol{x}_t|\nu = \tau)}{p(\boldsymbol{x}_t|\nu > \tau)} \tag{36}$$

almost surely. Note that for the binary hypothesis testing problem (in the case of change point detection), the optimal likelihood ratio test is defined as follows

$$\tau^*(X_t) = \begin{cases} 1, & \text{if } \frac{p(\boldsymbol{x}_t|\nu=\tau)}{p(\boldsymbol{x}_t|\nu>\tau)} \geq \gamma \\ 0, & \text{if } \frac{p(\boldsymbol{x}_t|\nu=\tau)}{p(\boldsymbol{x}_t|\nu>\tau)} < \gamma. \end{cases} \tag{37}$$

For the type-I error probability, we have that

$$
\begin{aligned}
&\mathbb{P}\Big( \lim_{M\to\infty} \frac{\frac{1}{M}\sum_{m=1}^{M} p(\boldsymbol{x}_t|\boldsymbol{z}_t^{(m)},\nu=\tau)}{\frac{1}{M}\sum_{m=1}^{M} p(\boldsymbol{x}_t|\boldsymbol{z}_t^{(m)},\nu>\tau)} \geq \gamma \Big) \\
&= \mathbb{P}\Big( \lim_{M\to\infty} \frac{\frac{1}{M}\sum_{m=1}^{M} p(\boldsymbol{x}_t|\boldsymbol{z}_t^{(m)},\nu=\tau)}{\frac{1}{M}\sum_{m=1}^{M} p(\boldsymbol{x}_t|\boldsymbol{z}_t^{(m)},\nu>\tau)} \geq \gamma, \ \lim_{M\to\infty} \frac{\frac{1}{M}\sum_{m=1}^{M} p(\boldsymbol{x}_t|\boldsymbol{z}_t^{(m)},\nu=\tau)}{\frac{1}{M}\sum_{m=1}^{M} p(\boldsymbol{x}_t|\boldsymbol{z}_t^{(m)},\nu>\tau)} = \frac{p(\boldsymbol{x}_t|\nu=\tau)}{p(\boldsymbol{x}_t|\nu>\tau)} \Big) \\
&\quad + \mathbb{P}\Big( \lim_{M\to\infty} \frac{\frac{1}{M}\sum_{m=1}^{M} p(\boldsymbol{x}_t|\boldsymbol{z}_t^{(m)},\nu=\tau)}{\frac{1}{M}\sum_{m=1}^{M} p(\boldsymbol{x}_t|\boldsymbol{z}_t^{(m)},\nu>\tau)} \geq \gamma, \ \lim_{M\to\infty} \frac{\frac{1}{M}\sum_{m=1}^{M} p(\boldsymbol{x}_t|\boldsymbol{z}_t^{(m)},\nu=\tau)}{\frac{1}{M}\sum_{m=1}^{M} p(\boldsymbol{x}_t|\boldsymbol{z}_t^{(m)},\nu>\tau)} \neq \frac{p(\boldsymbol{x}_t|\nu=\tau)}{p(\boldsymbol{x}_t|\nu>\tau)} \Big) \\
&= \mathbb{P}\Big( \lim_{M\to\infty} \frac{\frac{1}{M}\sum_{m=1}^{M} p(\boldsymbol{x}_t|\boldsymbol{z}_t^{(m)},\nu=\tau)}{\frac{1}{M}\sum_{m=1}^{M} p(\boldsymbol{x}_t|\boldsymbol{z}_t^{(m)},\nu>\tau)} \geq \gamma \Big| \ \lim_{M\to\infty} \frac{\frac{1}{M}\sum_{m=1}^{M} p(\boldsymbol{x}_t|\boldsymbol{z}_t^{(m)},\nu=\tau)}{\frac{1}{M}\sum_{m=1}^{M} p(\boldsymbol{x}_t|\boldsymbol{z}_t^{(m)},\nu>\tau)} = \frac{p(\boldsymbol{x}_t|\nu=\tau)}{p(\boldsymbol{x}_t|\nu>\tau)} \Big) \\
&\quad \cdot \mathbb{P}\Big( \lim_{M\to\infty} \frac{\frac{1}{M}\sum_{m=1}^{M} p(\boldsymbol{x}_t|\boldsymbol{z}_t^{(m)},\nu=\tau)}{\frac{1}{M}\sum_{m=1}^{M} p(\boldsymbol{x}_t|\boldsymbol{z}_t^{(m)},\nu>\tau)} = \frac{p(\boldsymbol{x}_t|\nu=\tau)}{p(\boldsymbol{x}_t|\nu>\tau)} \Big) \\
&= \mathbb{P}\Big( \frac{p(\boldsymbol{x}_t|\nu=\tau)}{p(\boldsymbol{x}_t|\nu>\tau)} \geq \gamma \Big),
\end{aligned}
\tag{38}
$$

where the last inequality is from (36). We can also derive the same result for the type-II error probability. $\square$

## B    Datasets and Preprocessing

This work used four datasets to evaluate the proposed and baseline models. We selected these datasets to demonstrate the effectiveness of change point detection in synthetic time-series data generation so that the datasets contain clear value shifts. We preprocessed all datasets by normalizing time-series per sequence with a zero mean and one variance. Let $X$ be a dataset with $N$ data points $x_i$, where $i = 1, 2, \ldots, N$. The normalization of $X$ is performed as follows:

First, calculate the mean $\mu$ and standard deviation $\sigma$ of $X$:

$$
\mu = \frac{1}{N} \sum_{i=1}^{N} x_i
\tag{39}
$$

$$
\sigma = \sqrt{\frac{1}{N} \sum_{i=1}^{N} (x_i - \mu)^2}
\tag{40}
$$

Then, normalize each data point $x_i$ to obtain the normalized value $z_i$ using:

$$
z_i = \frac{x_i - \mu}{\sigma}
\tag{41}
$$

### B.1    S&P 500 Dataset

We collected stock tickers from the S&P 500 and utilized per-sequence normalized price as the first dataset, which contains 504 tickers in total, as shown in Figure 8. We downloaded these original price time-series via the Yahoo Finance Python package [3]. The time range of this dataset starts from January 2020 and ends in June 2020. The length of each sample is 120. This dataset covers the significant stock price drop at the beginning of the COVID-19 pandemic in March 2020.

### B.2    S&P 500 Intraday Dataset

Like the S&P 500 dataset mentioned above, we collected the intraday prices on Feb 5th, 2024, to form the second dataset. The length of this dataset was also set to 120. The intraday prices were originally parsed as

---

[3]https://github.com/ranaroussi/yfinance

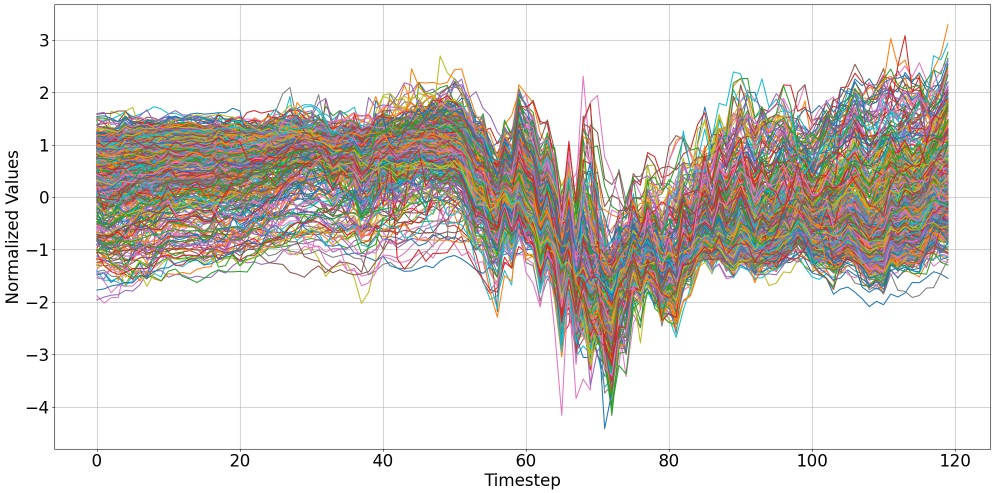

Figure 8: Normalized stock prices in S&P 500.

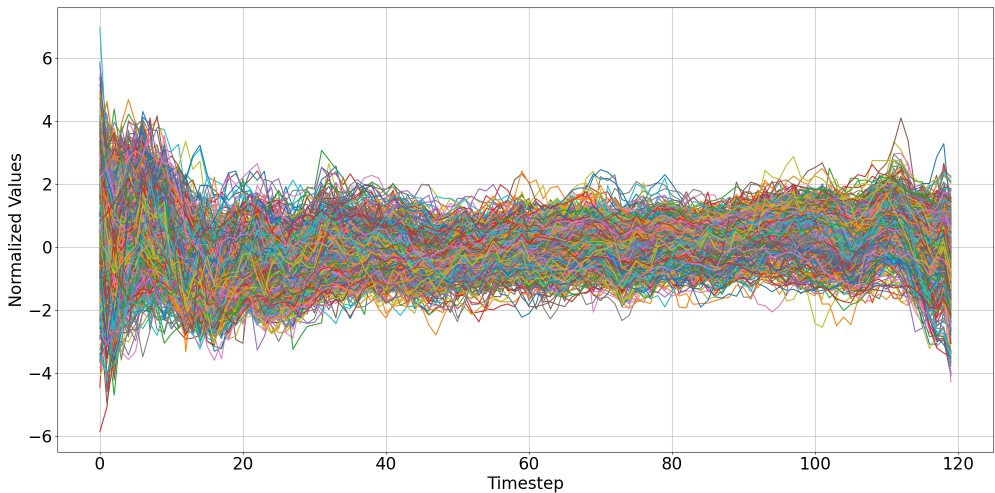

Figure 9: Normalized intraday stock prices in S&P 500.

per-min prices across the training day and then down-sampled to 120 steps for each stock. However, four stocks were removed because they had less than 120 data points because of a limited number of executed trades on the selected day. Thus, the total number of time-series samples is 500.

### B.3 Crypto Dataset

The third dataset is collected from cryptocurrency prices. We downloaded twelve cryptocurrency price time-series via the Yahoo Finance API and normalized them using the method mentioned above. The time range of this dataset starts in early March 2021 and ends in June 2022 with a total length of 120 time steps. This dataset covers when cryptocurrency prices significantly increased and varied in the first half of 2021.

### B.4 Air Quality Dataset

We also used the "Beijing Multi-Site Air-Quality Dataset", which is available on Kaggle [4] (Zhang et al., 2017). This dataset is an extensive collection of air quality measurements from 12 monitoring stations across Beijing.

---

[4]https://www.kaggle.com/datasets/sid321axn/beijing-multisite-airquality-data-set

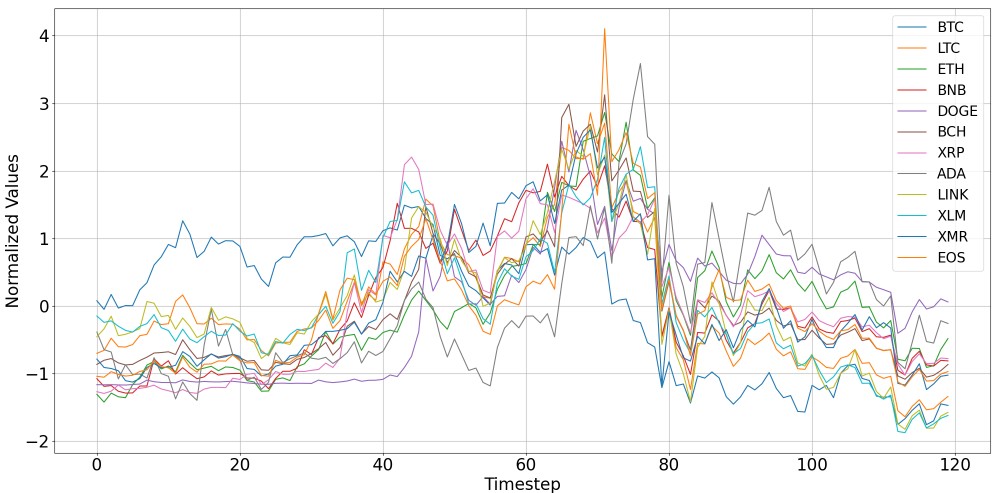

Figure 10: Normalized cryptocurrency prices.

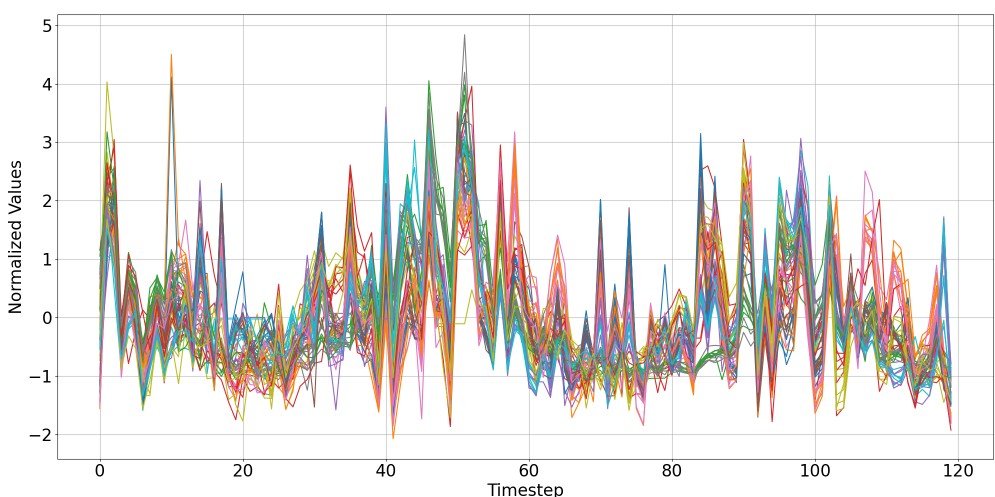

Figure 11: Normalized air quality measures.

It records various pollutants like PM2.5 and PM10 hourly from 2013 to 2018. Here, we aggregated the original hourly-based data into weekly-based data by taking the mean across corresponding values. We constructed the dataset by selecting five pollutants, including PM2.5, PM10, SO2, NO2, and CO. This dataset contains 60 time-series samples with a fixed length 120.

# C Ablation Studies

In this section, we present ablation studies conducted on both the synthetic and real datasets considered in this work.

## C.1 Synthetic OU Dataset

We consider a synthetic OU dataset with a single change point and train our proposed neural SDE model without change points. The goal of this ablation study is to test the various hyperparameters of the base `CP-SDEVAE` model. Table 2 shows the different hyperparameters tested, along with their assumed default value when the hyperparameter is assumed to be held fixed. In the following, we conduct our ablation study

| Parameter | Default | Search Space | Parameter Description |
|---|---|---|---|
| latent_dim | 32 | [4, 8, 16, 32] | latent space dimension |
| hidden_dim_encoder | 128 | [128] | # of encoder hidden units |
| num_layers_encoder | 1 | [1, 2, 3] | # of encoder layers |
| hidden_dim_sde | 64 | [32, 64, 128] | # of drift/diffusion hidden units |
| num_layers_sde | 2 | [1, 2, 3] | # of drift/diffusion layers |
| var_decoder | 1.0 | [0.01, 0.1, 1.0] | decoder variance |
| is_diffusion_homoscedastic | True | [False, True] | latent diffusion type |
| latent_diffusion_val | 1.0 | [0.01, 0.1, 1.0] | latent diffusion value |
| train_diffusion | False | [False, True] | flag for training diffusion |
| decoder_type | 'mlp' | ['linear', 'mlp'] | # of decoder hidden units |
| nll_weight | 1.0 | [0.001, 1.0] | weight of NLL in loss |
| kld_weight | 1.0 | [0.001, 1.0] | weight of KLD in loss |
| pred_nll_weight | 0.05 | [0.0, 0.01, 0.05, 0.1] | weight of predictive NLL |
| num_sde_trajectories | 5 | [1, 5, 10] | # of SDE trajectories |
| euler_step_size | 1.0 | [0.05, 0.1, 1.0] | step size for Euler solver |

Table 2: Hyperparameters: Default values, ablation study configurations, and concise descriptions

by varying the value of two hyperparameters at a time according to their corresponding search space as shown in Table 2. To test performance, we plot the generated trajectories from each trained model after $E = 250$ epochs, along with the corresponding ELBO.

### C.1.1 Latent SDE Size vs. Number of SDE Layers

In this part of the ablation study, we want to understand if varying both the number of layers in the SDE drifts and diffusion and the number of hidden neurons had a big impact on performance. We observe a weak trend that when the number of layers in the neural SDE drift and diffusion networks is small (1-3 layer), increasing the number of neurons achieves higher ELBO. The best parameter configuration here was to utilize 3 hidden layers with 64 neurons per layer, achieving an ELBO value of $-5.40$. The worst performing model was the most complex one (3 hidden layers with 128 neurons per layer), which achieved an ELBO of -7.24.

### C.1.2 Latent Space Size vs. Decoder Type

In this part of the ablation study, our goal is to compare different types of decoder: either a linear decoder (denoted by the configuration 'hidden_dim_decoder=None') or an single-layer MLP with 128 neurons. Along with the type of decoder, we also vary the size of the latent space. In general, we observe the trend that the linear decoder (across all latent dimension sizes), achieves equal or better performance as compared to the MLP decoder. This can be attributed to the fact that the decoder in this example is actually compressing the latent variable (whose size is greater than the time-series dimension). We notice that as the latent dimension gets larger, the performance of the the two decoders becomes more similar.

### C.1.3 Latent Space Size vs. Number of Encoder Layers

In this part of the ablation study, we vary both the number of encoder layers and the dimension of the latent random variable. Here, we observe the trend that utilizing a larger number of layers improves performance in terms of ELBO, across all latent dimension sizes. We note that the best performing model was achieved for the most complex model in our hyperparameter search space, where the size of the latent variable was 32 and the number of encoder layers was 3.

### C.1.4 Latent Space Size vs. Number of SDE Trajectories

In this part of the ablation, we study the impact of sampling additional SDE trajectories in our nested Monte Carlo estimator of the ELBO over a fixed number of iterations. We study this across different sizes of

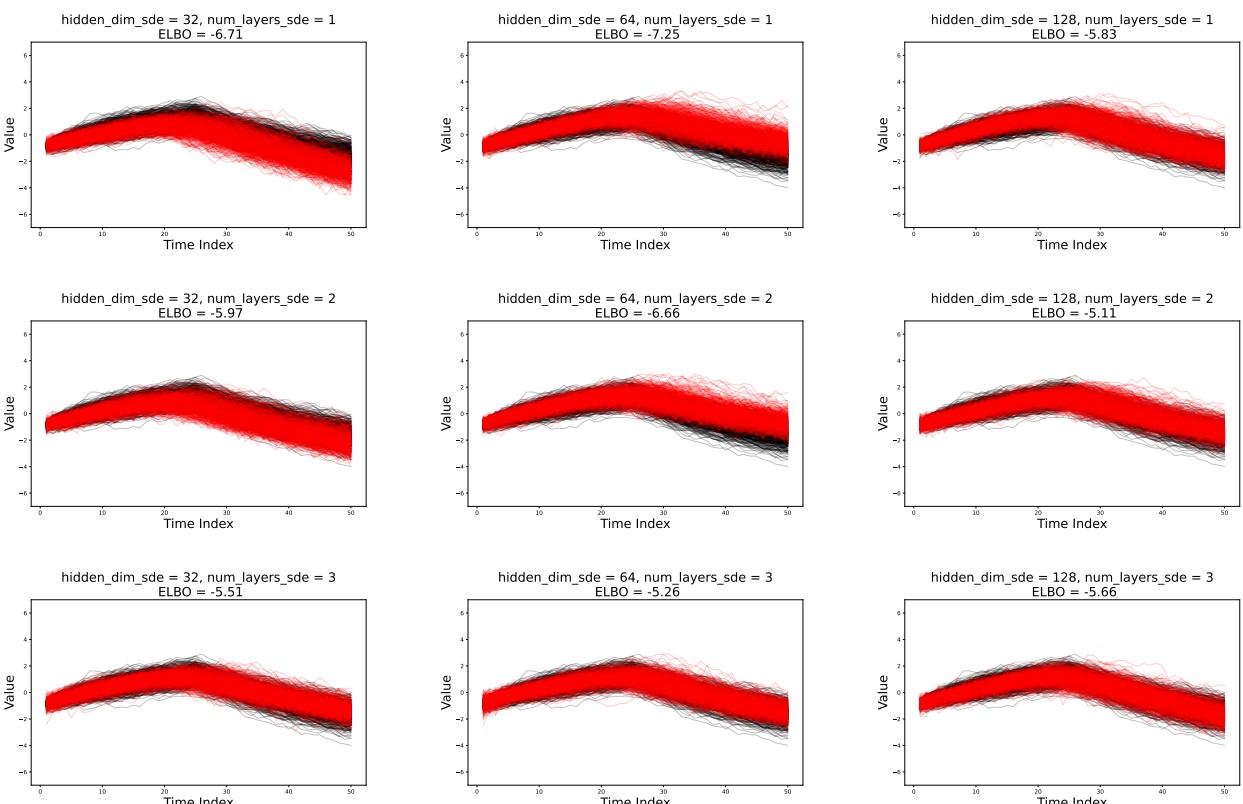

Figure 12: hidden_dim_sde vs. num_layers_sde.

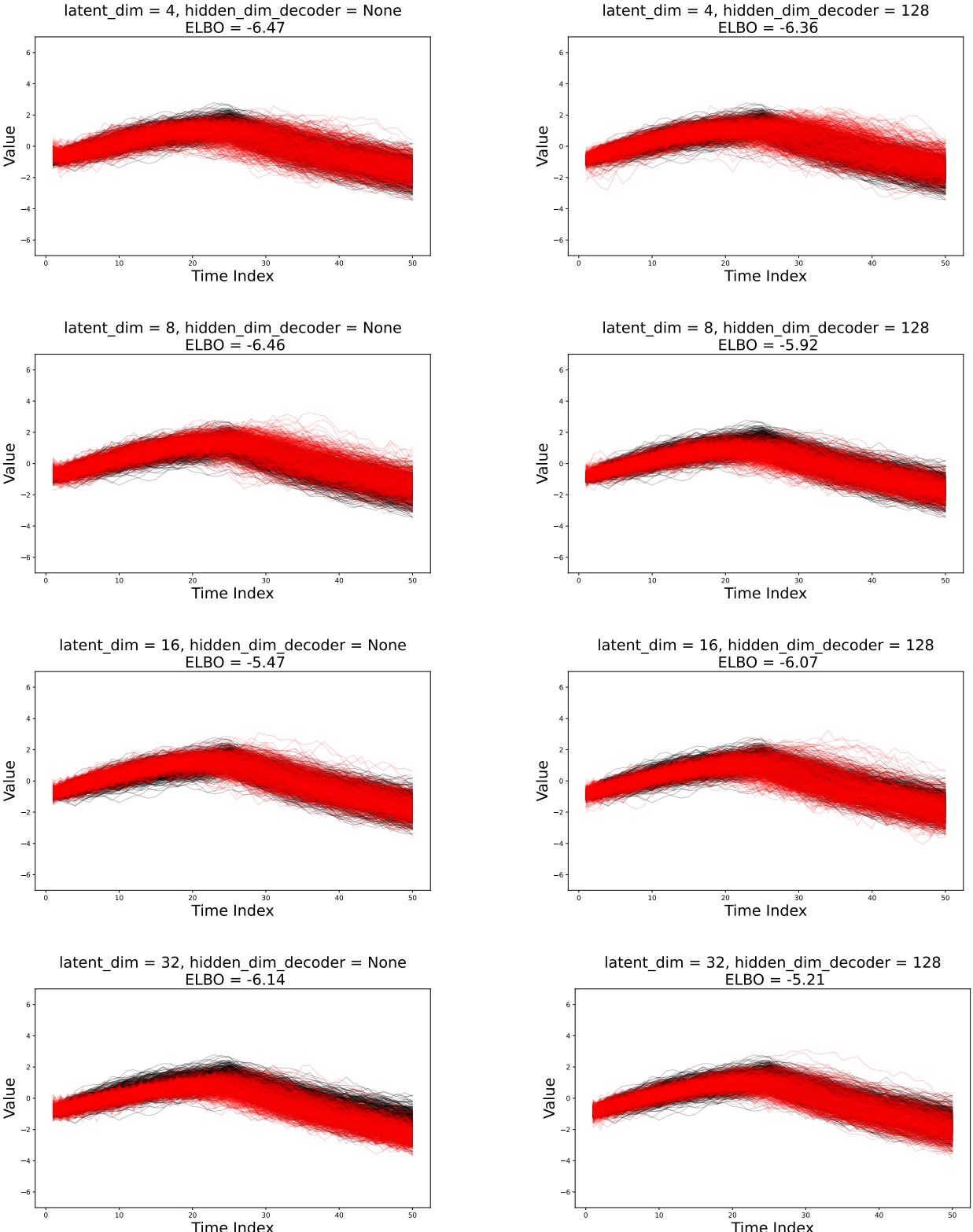

Figure 13: latent_dim vs. decoder_type.

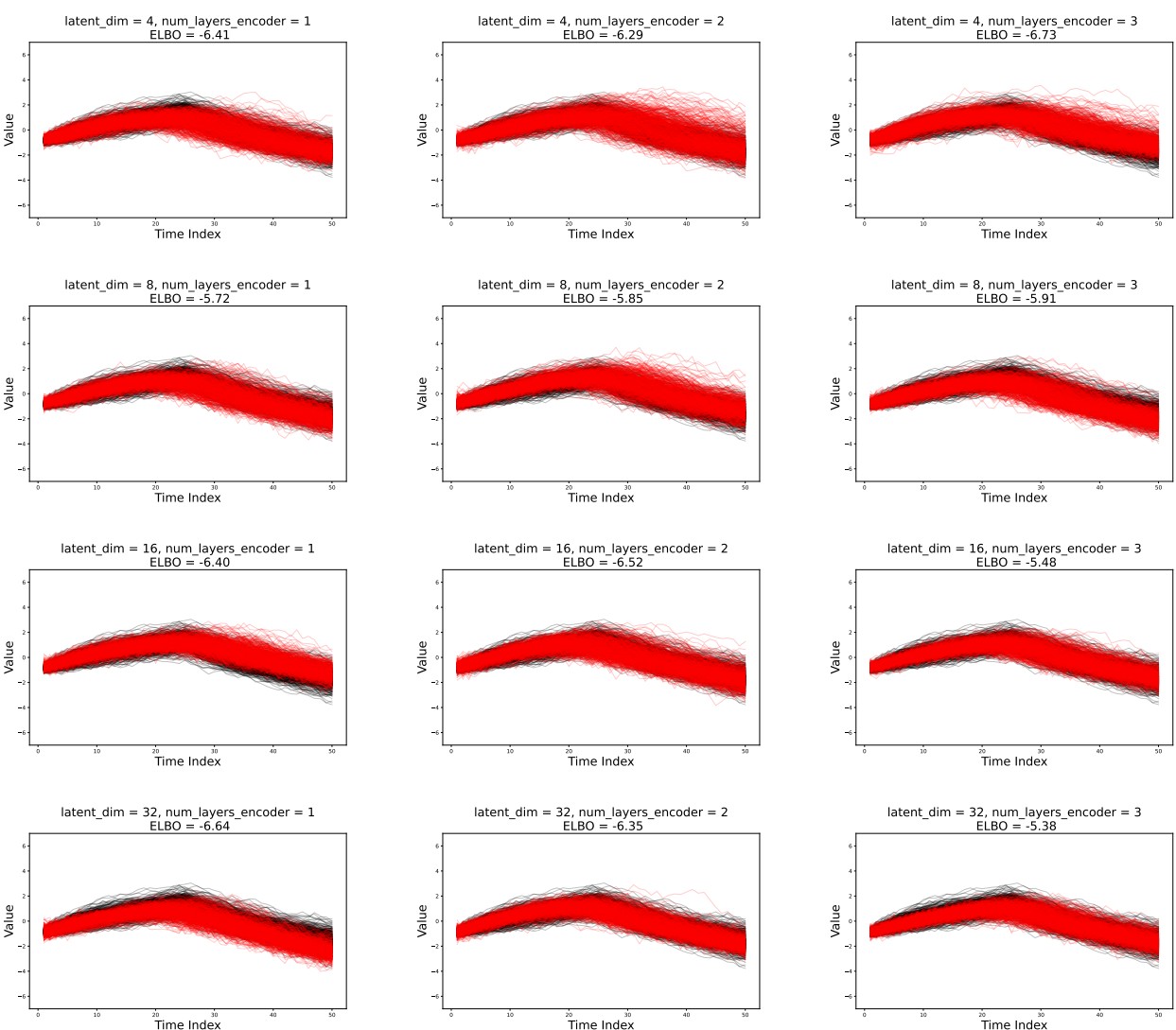

Figure 14: latent_dim vs. num_layers_encoder.

the latent random variable. Across all sizes of the latent variable, we observe the trend that increasing the number of SDE trajectories in the Monte Carlo estimator improves the model performance in terms of ELBO. We add that, based on the results in the figure, the generated samples also appear more realistic and capture the distribution of the underlying data better.

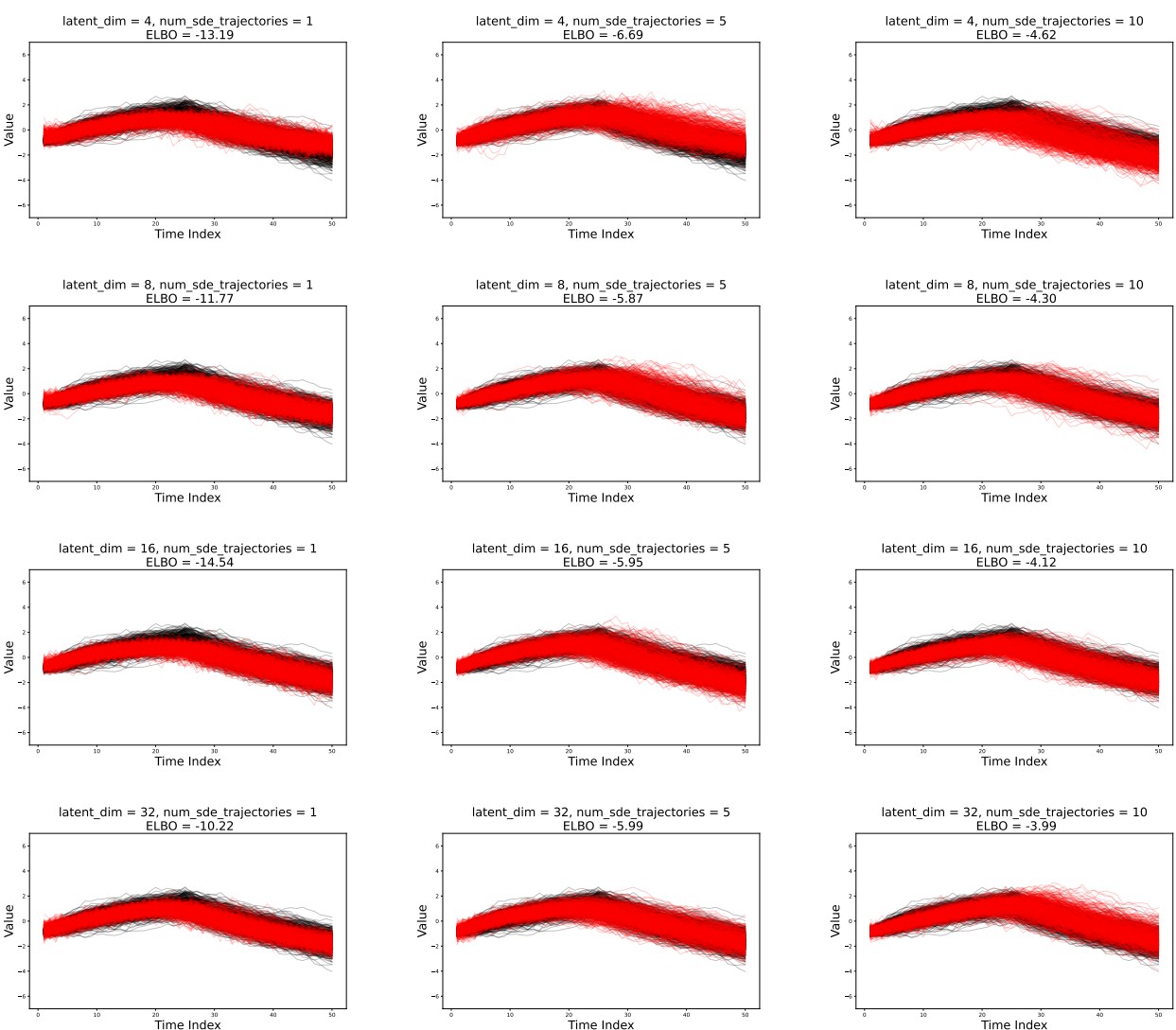

Figure 15: latent_dim vs. num_sde_trajectories.

### C.1.5 NLL Weight vs. Predictive NLL Weight

In this part of the ablation study, we test to see whether or not the predictive negative log-likelihood regularizer is adding any value to the model performance. We vary the value of the regularization parameter of the predictive NLL regularizer under negligible NLL component (NLL weight = 0.001) and under vanilla NLL component (NLL weight = 1). We can see that while removing the predictive NLL results in larger overall ELBO, the generated trajectories under the configuration of (pred NLL weight = 0) are noiseless, implying that the latent diffusion is meaningless (akin to a latent neural ODE). We can see that by incorporating the predictive NLL component, although the ELBO slightly decreases, the trajectories become noisier and thus are more realistic and better suited to modeling stochastic processes.

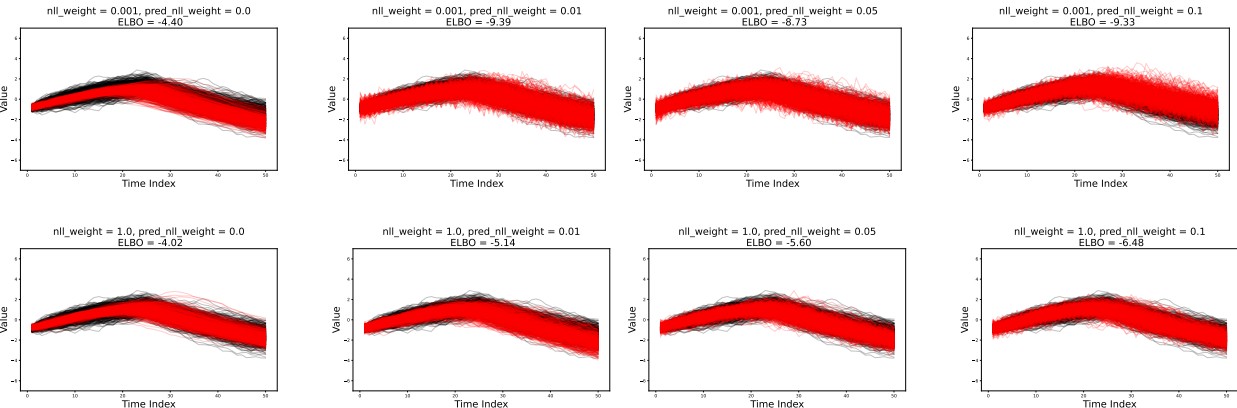

Figure 16: is_diffusion_homoscedastic vs. diffusion_value vs. train_diffusion.

### C.1.6 Homoscedastic vs. Heteroscedastic Diffusion

As part of our ablation study, we investigated three distinct approaches for modeling the latent diffusion in the neural SDE framework: 1.) learnable heteroscedastic diffusion; 2.) learnable homoscedastic diffusion; 3.) fixed homoscedastic diffusion. Our analysis revealed that all three configurations yielded comparable generated time-series, both in terms of trajectory dynamics and corresponding ELBO values. We posit that this similarity in performance can be attributed to the higher dimensionality of the SDE's latent space relative to the time-series data. This dimensional disparity allows even less complex latent dynamics to adequately capture the observed time-series dynamics. It is noteworthy that the heteroscedastic latent diffusion assumption marginally outperformed the other approaches in terms of ELBO. We hypothesize that this superior performance stems from the increased flexibility inherent in the heteroscedastic model.

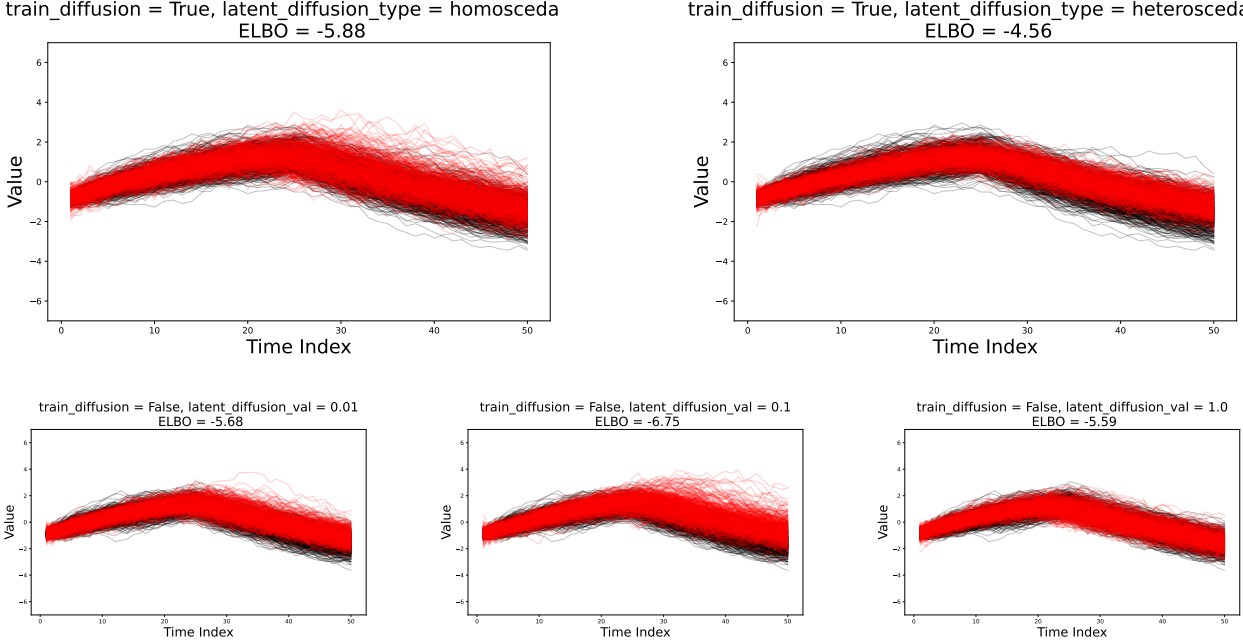

Figure 17: hidden_dim_sde vs. num_layers_sde.

### C.1.7 Decoder Variance vs. Latent Diffusion Value

Figure 18 shows the tradeoff in model performance when varying the variance of the decoder and the variance of the latent diffusion (in the case of a fixed, homoscedastic variance). We can see that decoder variance plays a larger role, as the ELBO is much worse when the decoder variance is too small. The best performance is obtained when the variance of decoder is 0.1 and the latent diffusion is 0.1.

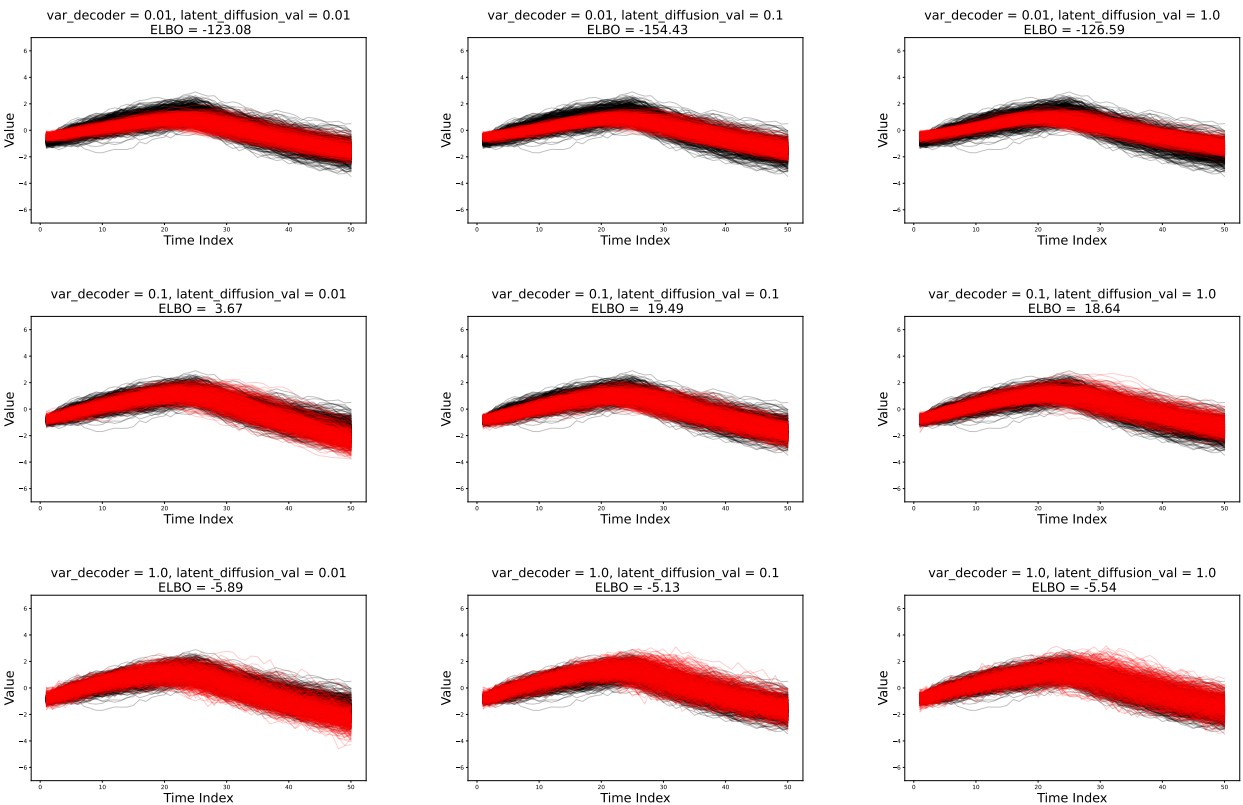

Figure 18: var_decoder vs. diffusion_value.

### C.2 S&P 500 Dataset

In this part of the ablation study, we test the need for change points in the real data experiments, with results for different number of SDE layers and different SDE hidden neuron sizes shown in Table 3. We provide an analysis of the results of this ablation in the main text.

Table 3: Ablation Study on the S&P500 Sectors Dataset

| Hidden Layers | Metric | Latent Size = 16 | | | Latent Size = 32 | | | Latent Size = 64 | | |
|---|---|---|---|---|---|---|---|---|---|---|
| | | SDE (0 CP) | SDE (1 CP) | SDE (2 CPs) | SDE (0 CP) | SDE (1 CP) | SDE (2 CPs) | SDE (0 CP) | SDE (1 CP) | SDE (2 CPs) |
| 1 | Marginal ↓ | $0.036\pm$ 0.004 | $0.036\pm$ 0.009 | $0.031\pm$ 0.009 | $0.037\pm$ 0.005 | $0.031\pm$ 0.002 | $0.032\pm$ 0.004 | $0.041\pm$ 0.006 | $0.033\pm$ 0.002 | $0.034\pm$ 0.003 |
| | Classification ↓ | $0.273\pm$ 0.147 | $0.255\pm$ 0.164 | $0.200\pm$ 0.046 | $0.264\pm$ 0.142 | $0.218\pm$ 0.078 | $0.191\pm$ 0.105 | $0.364\pm$ 0.104 | $0.273\pm$ 0.111 | $0.227\pm$ 0.104 |
| | Prediction ↓ | $0.243\pm$ 0.209 | $0.096\pm$ 0.006 | $0.127\pm$ 0.030 | $0.142\pm$ 0.020 | $0.080\pm$ 0.017 | $0.101\pm$ 0.018 | $0.121\pm$ 0.029 | $0.160\pm$ 0.138 | $0.100\pm$ 0.046 |
| 2 | Marginal ↓ | $0.029\pm$ 0.003 | $0.058\pm$ 0.037 | $0.028\pm$ 0.004 | $0.036\pm$ 0.006 | $0.030\pm$ 0.008 | $0.032\pm$ 0.003 | $0.035\pm$ 0.004 | $0.038\pm$ 0.006 | $0.035\pm$ 0.003 |
| | Classification ↓ | $0.218\pm$ 0.053 | $0.336\pm$ 0.084 | $0.145\pm$ 0.093 | $0.245\pm$ 0.130 | $0.227\pm$ 0.119 | $0.109\pm$ 0.062 | $0.218\pm$ 0.133 | $0.309\pm$ 0.060 | $0.200\pm$ 0.117 |
| | Prediction ↓ | $0.095\pm$ 0.029 | $0.381\pm$ 0.292 | $0.076\pm$ 0.013 | $0.103\pm$ 0.054 | $0.092\pm$ 0.018 | $0.085\pm$ 0.030 | $0.117\pm$ 0.100 | $0.123\pm$ 0.028 | $0.078\pm$ 0.020 |
| 3 | Marginal ↓ | $0.033\pm$ 0.005 | $0.031\pm$ 0.002 | $0.032\pm$ 0.005 | $0.034\pm$ 0.006 | $0.033\pm$ 0.006 | $0.031\pm$ 0.005 | $0.031\pm$ 0.004 | $0.034\pm$ 0.006 | $0.033\pm$ 0.003 |
| | Classification ↓ | $0.109\pm$ 0.089 | $0.173\pm$ 0.145 | $0.236\pm$ 0.034 | $0.109\pm$ 0.084 | $0.209\pm$ 0.084 | $0.227\pm$ 0.076 | $0.100\pm$ 0.067 | $0.200\pm$ 0.124 | $0.200\pm$ 0.084 |
| | Prediction ↓ | $0.072\pm$ 0.014 | $0.100\pm$ 0.079 | $0.059\pm$ 0.008 | $0.074\pm$ 0.018 | $0.074\pm$ 0.018 | $0.071\pm$ 0.028 | $0.057\pm$ 0.006 | $0.055\pm$ 0.008 | $0.063\pm$ 0.014 |

