# OpenReview forum: "Variational Neural Stochastic Differential Equations with Change Points"
_TMLR — Accepted by TMLR_

### Review · Reviewer_EA6f · 2024-11-13

**Summary Of Contributions:**

The submission presents a novel approach to the change point detection problem using neural stochastic differential equations (SDEs) within a variational autoencoder (VAE) framework. The authors propose a new offline neural SDE change point detection algorithm and introduce a modified variational inference method with an updated evidence lower bound (ELBO). This ELBO incorporates a new prior that is based on a more common and relaxed assumption, making the variational inference simpler and more tractable compared to previous approaches. This innovation facilitates more efficient training and improves the model's ability to detect change points in an offline setting.

The iterative algorithm alternates between training the neural SDE model and updating change point estimates through either a maximum likelihood-based method or a sequential likelihood ratio test. The paper includes theoretical guarantees, demonstrating convergence to a stationary point for the training algorithm and proving optimal error probability for the change point detection scheme under specific conditions.

Additionally, use of particle filtering is employed for estimating marginal likelihoods and log-likelihood ratios, enhancing the accuracy of change point detection within the VAE framework. Also, a predictive log-likelihood term is incorporated into the ELBO to improve the generative performance of the model, enabling it to capture complex time-series dynamics more effectively.

The model's effectiveness is validated through comprehensive experiments on synthetic and real-world datasets, such as S&P500 prices and air quality measurements. The results are supported by ablation studies and various evaluation metrics, assessing the model’s performance in terms of marginal distribution similarity, data indistinguishability, and predictive quality.

The authors also provide practical considerations for model training, including initialization strategies, optimizer configurations, and warm-start techniques to enhance stability and reliability.


All these thorough theoretical conclusions, well-described background, and thorough practical considerations demonstrate high expertise of the authors and relative consistency of the provided results. The submission presents a substantial contribution to time-series modeling by combining innovative algorithmic design, simplified variational inference, and robust practical applicability. However, some manuscript aspects need to be addressed before the publication (see below).

**Audience:**

Yes

**Broader Impact Concerns:**

The proposed work on variational neural SDEs for change point detection holds significant promise for advancing the fields of time-series analysis and predictive modeling, with strong potential applications in diverse areas such as finance, healthcare, and environmental monitoring. This contribution is commendable for its focus on detecting distributional shifts, which can enhance decision-making processes, improve system resilience, and foster proactive responses in dynamic environments.

One notable positive aspect is the potential for this technology to empower organizations to identify critical changes in data patterns more effectively, leading to better risk management and adaptive strategies. In finance, this could mean earlier detection of market shifts, while in healthcare, it could enable timely identification of significant changes in patient data trends, contributing to improved outcomes and patient care.

Additionally, the proposed algorithm could democratize access to sophisticated change point detection methods, enabling a wider range of industries to leverage advanced tools that were previously available only to those with substantial technical resources. This is a meaningful step toward making robust analytical capabilities more broadly accessible, which aligns with the overall push for equity in technological advancements.

The authors’ inclusion of practical considerations, such as initialization methods and warm-start strategies, reflects a strong awareness of real-world implementation challenges. This focus on practical application enhances the usability of the model and paves the way for seamless integration into existing analytical workflows, benefiting both research and industry practitioners.

While there are potential concerns, such as data privacy and fairness, the positive impact of this work can be amplified by proactive measures. By embedding best practices in transparency and responsible data use, this model could set a benchmark for ethical AI deployment. The authors should consider adding a **Broader Impact Statement** to highlight how this technology could be responsibly utilized, ensuring it supports fair, transparent, and equitable applications across various domains.

Security and resilience to adversarial scenarios are also areas where this model could shine. Developing robust training practices and emphasizing safeguards would further underscore the reliability and trustworthiness of the technology.

Overall, the manuscript presents a forward-thinking approach that could greatly benefit fields reliant on time-series data. By addressing broader societal and ethical considerations, the authors can reinforce the model's positive contributions and help guide its responsible adoption. A **Broader Impact Statement** would be a valuable addition, showcasing the authors' commitment to maximizing the positive societal impact of their work while acknowledging and mitigating potential risks.

**Claims And Evidence:**

Yes

**Requested Changes:**

# Major Areas for Improvement:

* **Handling Multiple Change Points**: While the paper mentions extending the algorithm to multiple change points, more detail on how this generalization is practically managed would strengthen the discussion on scalability. Now it seems each additional change point in a time series requires an extra SDE model with its own trainable parameters.

* **Types of Anomalies Detected**: Clarifying which types of anomalies (e.g., mean shifts, trends, volatility, seasonality) the algorithm can detect would help readers better understand its capabilities and practical applications.

* **Evaluation on More Datasets**: The use of four real datasets, while valuable, might be insufficient to assert the model’s generalizability. Expanding to more datasets or explaining the rationale behind the choice would strengthen this claim. Also, it is strongly recommended to use common change-point detection benchmarks (datasets corpuses + metrics), or explain the current choice of metrics.

* **Source Code for Reproducibility**: Publishing the source code is recommended for enhancing reproducibility and enabling peer validation.

* **Proof Sketch Enhancement**: The proof sketch on page 13 relies on a strong assumption regarding the impact of change point updates on variational approximation accuracy. Including a note that "The detailed proof can be found in the Appendix" would add more credibility to the "Under the assumption that change point updates do not vastly impact the accuracy of the variational approximation" statement.

* **Explanation of Table Metrics**: Metric names in Table 1 should be explained (Prediction=Synthetic Data Predictive Quality etc.), or have a caption to clarify their meanings. Adding abbreviations or justifying the chosen metrics over other standard benchmarks would improve clarity.

* **Other clarifications required**:
1. In 4.3.2, how the expected log-likelihood differs from (13)? How correct that substitution is? Does this means you decompose the standard log-likelihood L by two terms: \alpha*L+(1-\alpha)*L_pred in order to force the predictive capabilities of the autoecnoder
1. In (22), how to assess p(xt1:k|ν > t) ?

# Minor Areas for Improvement:

* **Discussion**: In the discussion section, the limitations could be expanded to discuss challenges related to scalability, robustness to noise, and highly non-linear data structures. Addressing potential drawbacks gives a balanced view of the model’s applicability. Also, some potential improvements with further research plan can be discussed in the section. Particularly, potential encoder's LSTM substitution with smth more recent and efficient can be discussed here.

* **Not all formulas are enumerated**: ... while some of them should. For instance, in the "Expected Predictive Log-Likelihood" subsection

* **Algorithm Description and Pseudocode**: In Algorithm figures, the current pseudocode format is quite suspicious. For instance, the steps in Algorithm 1, particularly the "Update change point" phase, need more detailed non-textual pseudocode. Also, in Algorithm 2&3 the "PF" stage should be revealed


* **X-Scale Consistency in Figures**: In Figure 6, aligning the X-scale across subplots (especially the Log-Evidence Ratio plot) would aid in direct comparison.


# Additional comment

Basically, the change-point updates iteration looks like an M-step of EM-algorithm. On the other hand, the ELBO estimate and maximization can be considered as E- and M- steps respectively. With this regard, can be embed the change-point updates iteration into the VAE training pipeline to make it end-to-end? You can add this thought to the Discussion section too as a part of further research plan

**Strengths And Weaknesses:**

# Strengths and Weaknesses

## Strengths:
- **Solid Background**: The paper provides a thorough background section, giving readers a solid foundation in related work and existing methodologies.
- **Strong Theoretical Findings**: The theoretical analysis, including convergence proofs and optimality guarantees, adds credibility to the proposed approach.
- **Well-Organized Experiment Design**: The experiments are well-structured, covering both synthetic and real-world datasets, and are supported by ablation studies to assess the impact of various model components.
- **Vast Practical Considerations**: The authors include practical details such as initialization strategies, optimizer configurations, and warm-start training procedures, making the model implementation-friendly.
- **Innovative Methodology**: The combination of change point detection with variational neural SDEs in a VAE framework is a novel approach that adds significant value to time-series modeling and change point analysis.
- **Practical Relevance**: The model’s applications in finance, healthcare, and environmental monitoring underscore its practical importance.
- **Clear Explanation of Training Steps**: The practical training steps provided help bridge the gap between theory and application.
- **Comprehensive Ablation Studies**: The paper’s ablation studies help highlight the significance of different model components.

## Weaknesses:
- **Some Theoretical Statements Are Unclear**: Certain theoretical explanations, such as those related to the proofs, could be more detailed for clarity.
- **Insufficient Number of Experiments on Real Data**: While experiments on synthetic data are strong, the number of real-world datasets used is limited, which raises questions about the model's generalizability.
- **No Limitations Discussion**: The paper does not sufficiently address the potential limitations of the algorithm, such as scenarios where it may not perform well.
- **Algorithm Scalability Issues**: The proposed method is not well-scalable when dealing with an increasing number of change points, potentially limiting its use for longer time-series with frequent changes.

---

> ### Author Response · Authors · 2024-12-31
> **Response to Reviewer EA6f (Number of Change Points & Types of Shifts)**
>
> > While the paper mentions extending the algorithm to multiple change points, more detail on how this generalization is practically managed would strengthen the discussion on scalability. Now it seems each additional change point in a time-series requires an extra SDE model with its own trainable parameters.
>
> Thank you for your comment. Your assumption is correct: for each additional change point considered in the modeling, new latent drift and diffusion functions are needed. We recognize that we only briefly discussed the idea of multiple change points and did not elaborate on it fully.  In the revision of our paper, we have will expand on our section regarding the extension to multiple change points to provide more details. We will also add a discussion on how one would choose between different candidate models (which each assume a different number of change points) using model selection.
>
> > Clarifying which types of anomalies (e.g., mean shifts, trends, volatility, seasonality) the algorithm can detect would help readers better understand its capabilities and practical applications.
>
> Thank you for your comment. We agree that the type of change point that is detected can be more explicitly defined. In our motivation, we explain that neural SDE’s inability to accurately capture distribution shift is due to the fact the existence of a strong solution to the SDE requires Lipschitz drift and diffusion functions. In theory, our model can detect any kind of shift, so long as it is a sharp change (i.e., violates Lipschitz assumption). In practice, our experiments revealed that change points were not needed for more complex datasets if the drift and diffusion networks were expressive enough. The benefit of incorporating the change point modeling in practice is that it may reveal different regimes in a time series, allowing for our model to generate different regimes of time series from our observed dataset in a straightforward way. In the revision of our manuscript, we plan to include both: (1) a discussion on theoretically what kind of changes can be detected with our algorithm; and (2) in practice, what is the utility of incorporating change point modeling for neural SDEs.
>
> > The use of four real datasets, while valuable, might be insufficient to assert the model’s generalizability. Expanding to more datasets or explaining the rationale behind the choice would strengthen this claim. Also, it is strongly recommended to use common change-point detection benchmarks (datasets corpuses + metrics) or explain the current choice of metrics.
>
>
> Thank you for your comment. Indeed, using more datasets would provide more convincing empirical evidence on the generalization capability of our model. During our exploration for datasets, we found it difficult to find open-source data with clear distributional shifts (outside of financial data from Yahoo finance). Our current experiments were focused on financial data due to the fact we have historical accounts for when market shocks occurred and can easily mine financial time series focusing on those shocks. The air quality dataset, while a standard benchmarking dataset for generative modeling of time series, does not have a clear distributional shift and so, it does a poor job showcasing the capability of our model in terms of capturing distributional shift. The air quality dataset is still valuable because it shows the baseline performance of SDEVAE without any change points. In the revision of our paper, we will add a justification for the choice of our datasets and how they were constructed. As of now, we have not expanded the experiments to include more datasets, but we are open to running our models and benchmarks on any suggested time series datasets. We will also explore other potential datasets and hope to find one by the end of the discussion period.
>
> > Publishing the source code is recommended for enhancing reproducibility and enabling peer validation.
>
> Unfortunately, due to restrictions, we are unable to share the source code publicly at this time. Although we cannot guarantee that the source code can be publicly shared in the future, we will make every effort to do so.

---

> ### Author Response · Authors · 2024-12-31
> **Response to Reviewer EA6f (Theoretical Results, Metrics, and Predictive Regularizer)**
>
> > The proof sketch on page 13 relies on a strong assumption regarding the impact of change point updates on variational approximation accuracy. Including a note that "The detailed proof can be found in the Appendix" would add more credibility to the "Under the assumption that change point updates do not vastly impact the accuracy of the variational approximation" statement.
>
> Thank you for your comment. We agree that providing reference to the detailed assumptions and proof would improve the quality of the manuscript. In the revision of our paper, we will explicitly point to our justification of the assumption regarding the variational approximation and the detailed proof of the fixed-point convergence. We also want to point out that Reviewer Pvqi asked for more explicit justification for this assumption as well. We have responded to Reviewer Pvqi in the official comment tilted “Response to Reviewer Pvqi (Validity of Assumption - Inference Gap)”. In this comment, we point out an typo we discovered in the assumption (wrong inequality direction), as well as further elaborate on the assumption relating to the inference gap.
>
> > Metric names in Table 1 should be explained (Prediction=Synthetic Data Predictive Quality etc.), or have a caption to clarify their meanings. Adding abbreviations or justifying the chosen metrics over other standard benchmarks would improve clarity.
>
> In the revision of our manuscript, we will update the results table to clarify each metric in relation to the metrics discussed in the main body of the manuscript. We will also add justification as to why these specific metrics were chosen. In particular, the three metrics: (1) marginal score; (2) classification score; and (3) prediction score, are all measures of the fidelity (realism) of the synthetic data generated from the generative model. It is the best metric to assess which models are best capturing potential distribution shifts in the generated time series datasets, as models that cannot capture distribution shifts are likely to perform poorly across all three of these metrics. Each of these scores have previously been used in other works for time series generative models. For example, SDEGAN, one of the baselines we are comparing to, performs a very similar evaluation [1].
>
> > In 4.3.2, how the expected log-likelihood differs from (13)? How correct that substitution is? Does this means you decompose the standard log-likelihood L by two terms: $\alpha*L+(1-\alpha)*L_{pred}$ in order to force the predictive capabilities of the autoencoder.
>
> Thank you for your comment. If I understand your question correctly, you want to understand the difference between the expected log-likelihood (which is defined in Section 4.3.1) and the expected predictive log-likelihood (which is defined in Section 4.3.2). Firstly, as pointed out by Reviewer Pvqi, there is a typo in the formula for the expected predictive log-likelihood and should instead be:
>
> $$ L_{\theta, \phi}^{pred} (x_{\rm obs}) \triangleq E_{q_{\phi}}\left[\sum_{k=1}^K \log \left(E\left[\prod_{k=1}^K p_{\theta}(x_{t_k}|z_{t_{k-1}}) | z_0\right]\right)\right] $$
>
> With that corrected the main difference between these two log-likelihoods is that the one defined in 4.3.1 is simply the conditioning on $z_{t_{k}}$ (4.3.1) rather that $z_{t_{k-1}}$ (4.3.2). The former is the reconstruction term that comes from our derivation of the ELBO for the SDEVAE, while the latter is a regularizer we intuitively defined in order to help improve the noise estimation properties of our model.  Moreover, the expected predictive log-likelihood requires an additional marginalization to compute $ p_{\theta}(x_{t_k}|z_{t_{k-1}})$, for which we employ an extended Kalman filter (EKF) [2]. We want to clarify that we have not established any connection between the log-likelihood defined in 4.3.1 and the regularizer defined in 4.3.2. As mentioned in our responses to the other reviewers, underestimation of latent diffusion noise is a problem that can occur in neural SDEs [3]. In our revision of the manuscript, we will elaborate on this issue of noise estimation by pointing reference to [3]. We also will elaborate more on how the EKF is used to approximate the predictive log-likelihood.
>
> [1] Kidger, Patrick, et al. "Neural sdes as infinite-dimensional gans." International conference on machine learning. PMLR, 2021.
>
> [2] Smith, Gerald L., Schmidt, Stanley F. and McGee, Leonard A. “Application of statistical filter theory to the optimal estimation of position and velocity on board a circumlunar vehicle”, volume 135. National Aeronautics and Space Administration, 1962.
>
> [3] Heck, Linus, et al. "Improving the Noise Estimation of Latent Neural Stochastic Differential Equations." arXiv preprint arXiv:2412.17499 (2024).

---

> ### Author Response · Authors · 2024-12-31
> **Response to Reviewer EA6f (Minor Comments)**
>
> > In (22), how to assess $p(x_{t_{1:k}}|\nu > t)$?
>
> Thank you for your comment. Since $p(x_{t_{1:k}}|\nu > t)$ is the likelihood that the change happens after time $t$, we can evaluate the likelihood of $x_{t_{1:k}}$ under the pre-change SDE (with parameters $\theta_0$), while for $p(x_{t_{1:k}}|\nu = t)$, we use the post-change SDE (with parameters $\theta_1$).  In the revision of our paper, we will include explicit referencing to the formulas required for computing these likelihoods.
>
> > the discussion section, the limitations could be expanded to discuss challenges related to scalability, robustness to noise, and highly non-linear data structures. Addressing potential drawbacks gives a balanced view of the model’s applicability. Also, some potential improvements with further research plan can be discussed in the section. Particularly, potential encoder's LSTM substitution with smth more recent and efficient can be discussed here.
>
> We will expand the conclusion section of our paper to provide a discussion on the limitations of our algorithm, as long as some potential future directions, including incorporating differentiable change point updates and testing more advanced types of architectures for both the encoder and decoder networks.
>
> > Not all formulas are enumerated: ... while some of them should. For instance, in the "Expected Predictive Log-Likelihood" subsection
>
> In the original version of the manuscript, we tried to limit the equations we numbered to only equations we referenced in the manuscript. We recognize that this is not consistent. In our revision of the paper, we will try to enumerate all relevant formulas.
>
> > In Algorithm figures, the current pseudocode format is quite suspicious. For instance, the steps in Algorithm 1, particularly the "Update change point" phase, need more detailed non-textual pseudocode. Also, in Algorithm 2\&3 the "PF" stage should be revealed
>
> In the revision of the manuscript, we will add additional pseudocode specifically for the particle filtering method implementation we employed in our work, which shows how both the marginal likelihood and the log-evidence ratio are computed. We will also add a function call of this pseudocode in our algorithm table for both of our approaches to updating the change point.
>
> > In Figure 6, aligning the X-scale across subplots (especially the Log-Evidence Ratio plot) would aid in direct comparison.
>
> Thank you for pointing out this inconsistency. In our revision of the paper, we will include newly generated figures to show consistency between the log evidence ratio plots.
>
> > Basically, the change-point updates iteration looks like an M-step of EM-algorithm. On the other hand, the ELBO estimate and maximization can be considered as E- and M- steps respectively. With this regard, can be embed the change-point updates iteration into the VAE training pipeline to make it end-to-end? You can add this thought to the Discussion section too as a part of further research plan
>
> Thank you for your comment. This is an interesting point. At this point, we are not sure exactly how to reconcile simultaneous updates of the change points and the model parameters (via an EM formulation). The main issue with considering this simultaneous update is the fact that the change point updates themselves represent an optimization problem over a discrete number of hypotheses ($K$ possible change points). Since model parameter updates are made by performing a gradient step, it’s not straightforward to merge the two together. One thing that would be an interesting future line of research is the idea of incorporating differentiable change point updates, using ideas like what was proposed in [1].  In the revision of our manuscript, we will add this as part of the direction for future research and include reference to some works on differentiable change point detection.
>
> [1] Koley, Paramita, et al. "Differentiable Change-point Detection With Temporal Point Processes." International Conference on Artificial Intelligence and Statistics. PMLR, 2023.

---

### Review · Reviewer_8zqg · 2024-11-17

**Summary Of Contributions:**

This paper introduces a new framework for training SDEs as VAEs, requiring only a prior on the initial state $\boldsymbol{z}_0$, instead of the entire trajectory. It develops methods for modeling change points in time-series modeled as latent neural SDEs, including an iterative algorithm for joint learning of SDE parameters and change points. The approach is demonstrated to achieve comparable performances to state-of-the-art models on distributional shift generation benchmark datasets, offering improved modeling of complex time-series with abrupt dynamic changes.

**Audience:**

Yes

**Claims And Evidence:**

Yes

**Requested Changes:**

Please see the weeknesses listed above.

Additionally, I recommend that the authors clarify the differences between the proposed method and LS4 in terms of their strategies or underlying ideas. This clarification is particularly important given that LS4, which does not explicitly model change points, achieves competitive or better performance than the proposed method.

**Minor comments**

- $d_z$ is not defined.

- What is $\boldsymbol{\sigma}_{\boldsymbol{\phi}}$ on page 2? I could not find its definition.

- In the first paragraph of Section 4.3.2, $\alpha$ and $\beta$ should be defined.

- The first equation in Section 4.3.2 (i.e., the definition of $\mathcal{L}^{\mathrm{pred}}$) contains $p\_{\boldsymbol{\theta}}(\boldsymbol{z}\_{t\_{k}}|\boldsymbol{x}\_{t\_{k-1}})$, but should this instead be $p\_{\boldsymbol{\theta}}(\boldsymbol{x}\_{t\_{k}}|\boldsymbol{z}\_{t\_{k-1}})$?

- Does Eq. (20) contain typos? Should it be $p(\boldsymbol{x}\_{\mathrm{obs}}|\nu=\tau)=\prod_{k=1}^K p(\boldsymbol{x}\_{t_k}|\boldsymbol{x}\_{t<t_k},\nu=\tau)$?

- Eq. (21) should be consistent with Eq. (16); however, there are several discrepancies. For instance, $\tau$ is not clearly connected with $t\_{k+1}$ in Eq. (16). Additionally, the meaning of $t$ is unclear in Eq. (22).

- The definition of $\hat{p}$ in Eq. (22) is required.

**Strengths And Weaknesses:**

**Strengthes**

1. The proposed strategy is simple, and the algorithm is straightforward.

2. The authors evaluate the proposed method across various settings, ranging from controlled experiments to practical scenarios.

3. The manuscript is easy to follow, even for non-experts in the field.

**Weaknesses**

1. Although the motivation of the paper is clearly stated, it is unclear which empirical results validate the effects of the weaker assumption on the prior. A more in-depth analysis, such as experiments on toy datasets comparing these two priors, would make the paper's motivation more convincing.

2. The number of change points is treated as a hyperparameter, meaning it must be decided before model training. While the authors examined cases with different numbers of change points on toy datasets, further analysis on how the choice of this parameter affects performance (e.g., setting it to more than 2 in Sections 5.1) would enhance the validity of the proposed approach.

3. The motivation for introducing the predictive log-likelihood in Section 4.3.2 is unclear. The explanation provided was not sufficient for me to understand its purpose. Further elaboration would help readers follow this section more effectively.

4. Although Theorem 1 provides convergence analysis for the proposed update scheme, it is not clear how close the stationary point is to the (global) optimal solution. If discussing this gap is difficult, it should at least be mentioned as a limitation of the method.

5. Theorem 2 addresses only the extreme case $J \to \infty$, which is not helpful in understanding how large $J$ should be in practice or how $J$ affects the approximation error.

6. The manuscript contains some errors and undefined quantities in the formulations, which can be easily corrected.

---

> ### Author Response · Authors · 2024-12-31
> **Response to Reviewer 8zqg (Motivating Modeling Choices)**
>
> > Although the motivation of the paper is clearly stated, it is unclear which empirical results validate the effects of the weaker assumption on the prior. A more in-depth analysis, such as experiments on toy datasets comparing these two priors, would make the paper's motivation more convincing.
>
> Thank you for your comments. Reviewer Pvqi has expressed a similar concern, and we agree that more direct comparison to the LatentSDE model (which assumes a prior on the stochastic process) and our model (which assumes a prior only on the initial state) would improve the quality of the paper. In the revision of our paper, we will include a justification for modeling choice; please see the official comment titled “Response to Reviewer Pvqi (Motivation of Paper)”, where we have justified the choice of the prior over the initial state rather than the stochastic process because:
>
> - Empirically, we observed that it produces higher fidelity time series
> - It serves as a VAE analog to the W-GAN formulation of neural SDEs
> - It allows us to easily incorporate change points with changes representing an underlying difference in latent SDE dynamics pre/post change point.
>
> We are in the process of running more elaborate experiments with the LatentSDE model to provide a comparison with our approach using the toy datasets.
>
> > The number of change points is treated as a hyperparameter, meaning it must be decided before model training. While the authors examined cases with different numbers of change points on toy datasets, further analysis on how the choice of this parameter affects performance (e.g., setting it to more than 2 in Sections 5.1) would enhance the validity of the proposed approach.
>
> We agree with the reviewer that it is important to show how the number of assumed change points impacts generative performance. In the revision of our manuscript, we will include an additional part of our toy experiment showcasing the performance of the model when 3-4 change points are assumed (when there is only one true change point). We will also add a discussion in our methodology that explains how one can calibrate the number of change points based on model selection (via cross-validation) using marginal likelihood (please see official comment titled “Response to Reviewer Pvqi (misc. questions)” for more information on what will be included). Our new analysis will show the relationship between the number of change point and performance using both the ELBO objective function and the visual quality of the samples. We hope to have this additional experiment completed before the end of the discussion period.
>
> > The motivation for introducing the predictive log-likelihood in Section 4.3.2 is unclear. The explanation provided was not sufficient for me to understand its purpose. Further elaboration would help readers follow this section more effectively.
>
> Thank you for pointing out the lack of clarity in this section. We agree that further elaboration on the motivation for the predictive log-likelihood regularizer should be included in the revision of the manuscript. To clarify, we introduced the predictive log-likelihood regularizer to improve the quality of the trajectories generated from our model (this was an empirical observation). We found that our model underestimated the noise diffusion when trained with the standard ELBO objective function (a phenomenon that was observed also when training the LatentSDE model - see recent work here [1]). To enhance the quality of our manuscript and to provide adequate context, we will expand the motivation section of the predictive regularizer in the main body of our manuscript. We will also include more explicit reference to our ablation studies relating to this regularizer to explain the benefits to highlight the empirical advantages in terms of model performance.
>
> [1] Heck, Linus, et al. "Improving the Noise Estimation of Latent Neural Stochastic Differential Equations." arXiv preprint arXiv:2412.17499 (2024).

---

> ### Author Response · Authors · 2024-12-31
> **Response to Reviewer 8zqg (Theoretical Results)**
>
> > Although Theorem 1 provides convergence analysis for the proposed update scheme, it is not clear how close the stationary point is to the (global) optimal solution. If discussing this gap is difficult, it should at least be mentioned as a limitation of the method.
>
> We would like to clarify that obtaining the convergence rate of our method is challenging since our method updates the change points and model parameters alternatively and overall, the problem is non-convex. We will discuss it as a limitation of our method in the revision.
>
> > Theorem 2 addresses only the extreme case $J\rightarrow\infty$, which is not helpful in understanding how large $J$ should be in practice or how $J$ affects the approximation error.
>
> Thank you for asking about Theorem 2. While we focus on $J\rightarrow\infty$ in the theorem, this suffices to describe the convergence rate of the estimator, which in turn is related to the approximation error. Note that because the estimator converges almost surely, it implies that it also converges in distribution, meaning we can directly apply the delta method to obtain the convergence rate [1].
>
> With a little abuse of notation and under standard assumptions (e.g., existence of moments), let $\mu(x_{t_{1:k}})$  and $\Sigma(x_{t_{1:k}})$ denote the mean and covariance matrix of the observed time series, respectively. Since we sample $J$ independent trajectories, we have that $\sqrt{J}(x_{t_{1:k}}-\mu(x_{t_{1:k}})) \overset{d}{\rightarrow} \mathcal{N}(0, \Sigma(x_{t_{1:k}}))$. Then, by the delta method,   the estimator of the likelihood ratio converges in distribution to the true likelihood ratio at the same rate (up to a multiplicative constant) of $\frac{1}{\sqrt{J}}$.  This creates a clear trade-off: larger $J$ values improve accuracy but increase computational cost. In practice, $J$ can be tuned to balance performance and computational efficiency.
>
> In the revision of our paper, we will include the convergence rate of the estimator and provide a discussion on how the number of paths sampled in the BPF impacts the approximation error, and hence the accuracy of the change point updates.
>
> [1] Fisher, Ronald Aylmer. "Theory of statistical estimation." Mathematical proceedings of the Cambridge philosophical society. Vol. 22. No. 5. Cambridge University Press, 1925.

---

> ### Author Response · Authors · 2024-12-31
> **Response to Reviewer 8zqg (Regarding comparison with LS4)**
>
> > Additionally, I recommend that the authors clarify the differences between the proposed method and LS4 in terms of their strategies or underlying ideas. This clarification is particularly important given that LS4, which does not explicitly model change points, achieves competitive or better performance than the proposed method.
>
> Thank you for your comments. We agree that since the LS4 method achieves competitive (or better) performance in comparison to our model, more elaboration as to how the two models are different would improve the quality of the manuscript. To that end, we will expand our description of the baseline methods and add more details to our description of the LS4 method in our experiments section of the paper. We will also add a discussion on how that model differs from neural SDE models in general (regardless of the inclusion of change points).

---

### Review · Reviewer_Pvqi · 2024-11-25

**Summary Of Contributions:**

The present paper introduces a novel framework for modeling time-series data with distributional shifts using neural stochastic differential equations (SDEs). Leveraging a VAE framework, the proposed method relies on a variational prior for the initial state of a latent neural SDE instead of priors on the full process as used in previous work. Moreover, the method integrates change points into the models by representing the dynamics with different neural SDEs between different change points. The model alternates between training neural SDE parameters and refining change points either using greedy maximum likelihood-based updates or sequential change point detection schemes. The paper provides theoretical insights regarding convergence guarantees for the training algorithm and optimality for the detection scheme under specific conditions. Finally, the method is empirically validated on synthetic data, such as Ornstein-Uhlenbeck processes with distributional shifts, and real-world datasets, including financial and air quality datasets.

**Audience:**

Yes

**Claims And Evidence:**

No

**Requested Changes:**

**Motivation:**
- It remains unclear why a prior over the entire latent stochastic process might be too strong an assumption. The reasoning "as the training data may not always conform to this prior" should be further explained. Also, one could argue that regularization can be helpful in penalizing large deviations from the prior drift (see (8)). It also remains questionable whether summarizing all the observations into a single initial condition for the SDE is a good modeling paradigm.

**Minor issues:**
- Neural SDE also fall under the "traditional SDE modeling" as formulated in the introduction, i.e., they have "parametric models for the drift and diffusion functions" and "model parameters are then learned using [e.g.] maximum likelihood estimation."
- $p_\alpha$ is defined as initial distribution; however, it seems to be used differently in (8). Generally, this section is hard to follow.
- The variational distribution and its parameters $\phi$ appear in the Algorithm Summary in Section 4.2; however, they are not properly defined before.
- The Lipschitz continuity of the drift and diffusion terms seems to be required in the spatial dimension. In the temporal dimension, Hölder continuity (with $\alpha=0.5$) seems to be sufficient for, e.g., Euler-Maruyama.

**Questions:**
- One version ("sequential likelihood ratio test") of the second step of the proposed algorithm ("Update the change points") seems similar to the work by Ryzhikov et al. (2022). In that sense, the distinction between *offline* and *online* algorithms in Section 3.2 seems somewhat artificial.
- Is "backpropagating through the SDE solver" not implicitely using "the reparametrization trick". It would be good to clarify the discussion before Section 4.3.2.
- It would enhance accessibility to add further explanations to Section 4.3.2, both on the motivation/derivation and the estimator. Is it correct that this regularization is not considered in the theoretical results?
- The "assumption that the inference gap as a result of the variational approximation does not widen after change points are updated" seems to be quite strong. Can the authors comment on its validity?
- Why are the MLE-based change point updates not evaluated for the experiment in Section 5.1.2?
- Can the authors evaluate and discuss the performance vs. cost tradeoff (e.g., in terms of runtime/flops) of the considered methods in Table 1? In particular, it would be interesting to observe the empirical scaling of the costs of the proposed method with different numbers of change points (including > 2). Specifically, the observation that the "complexity of the underlying neural SDE model increases, the necessity for change points to enhance performance diminishes" raises the question of whether the additional cost for estimating the change points is justified.
- Why does the experiment in Section 5.2 not consider the method by Sun et al. (2024)?
- Can the authors comment on the shortcoming that the number of change points needs to be known in advance?

**Typos:**
- "a prior" -> a priori
- In several places, `\citet` instead of `\citep` should be used.
- "identity distribution shifts" -> identify

**Strengths And Weaknesses:**

**Strengths:**

- The paper is generally well-written and well-structured.
- Focusing on change points in time-series data is a promising and impactful research direction given a range of applications with abrupt distributional shifts.
- The overall VAE-based framework and alternative training methodology (between neural SDE parameter updates and change point detection) seem to be novel.
- Theoretical guarantees are provided, and the resulting method achieves robust and competitive performance in the presented experiments.

**Weaknesses:**
- It remains unclear how strong the method really is because of the following considerations (which are not sufficiently evaluated in real-world scenarios):
	- the number of change points needs to be known in advance.
	- the computational complexity seems to increase significantly with the number of change points.
	- increasing the network complexity can diminish the importance of change point detection.
	- other methods such as LS4 achieve comparable performance.
- The method could be better motivated and explained (see the comments below).

---

> ### Author Response · Authors · 2024-12-30
> **Response to Reviewer Pvqi (Motivation of Paper)**
>
> > It remains unclear why a prior over the entire latent stochastic process might be too strong an assumption. The reasoning "as the training data may not always conform to this prior" should be further explained. Also, one could argue that regularization can be helpful in penalizing large deviations from the prior drift (see (8)). It also remains questionable whether summarizing all the observations into a single initial condition for the SDE is a good modeling paradigm.
>
> Thank you for these important points. Our choice to use a Gaussian prior on the initial latent state rather than a Wiener process prior was motivated by several empirical and theoretical advantages:
> - While developing our method, we found that when we compared the quality of the generation time series with the implementation in [1], our approach achieved better fidelity in generated time-series.
> - Our formulation enables direct comparison with the W-GAN formulation proposed in [2] (SDEGAN), which similarly uses a Gaussian initial state distribution and allows the learned latent SDE dictate the dynamics of the generated time series.
> - Most importantly, our approach straightforwardly and elegantly accommodates change point modeling. The LatentSDE model from [1] would require either (a) using a more complex prior (e.g., jump process) that may make the KLD intractable, or (b) implementing different decoders before and after the change point, which forces the shift to be captured solely in the decoder component. Our method avoids these limitations by maintaining analytical KLD expressions (KLD between two Gaussians has a simple closed form expression) while explicitly incorporating change points in the learned latent neural SDE (we can distinguish between pre/post change SDEs).
>
> We acknowledge these advantages weren't sufficiently explained in the manuscript. We will add clarification comparing the three models and demonstrating why SDEVAE (our approach) is better suited for change point modeling. The revised version will also include additional toy experiments showing the benefits of SDEVAE versus the latent stochastic process prior proposed in [1] (LatentSDE).
>
> [1] Li, Xuechen, et al. "Scalable gradients and variational inference for stochastic differential equations." Symposium on Advances in Approximate Bayesian Inference. PMLR, 2020.
>
> [2] Kidger, Patrick, et al. "Neural sdes as infinite-dimensional gans." International conference on machine learning. PMLR, 2021.

---

> ### Author Response · Authors · 2024-12-30
> **Response to Reviewer Pvqi (Minor Issues)**
>
> > Neural SDE also fall under the "traditional SDE modeling" as formulated in the introduction, i.e., they have "parametric models for the drift and diffusion functions" and "model parameters are then learned using [e.g.] maximum likelihood estimation."
>
> Thank you for your comment. While neural networks are technically parametric models, there is an important distinction between them and the parametric functions typically used for modeling drifts and diffusions in SDE models. We propose revising our wording to "simple parametric models" to better differentiate between simple functions and the more flexible neural network parameterizations. We have updated the manuscript accordingly.
>
> > $p_\alpha$ is defined as initial distribution; however, it seems to be used differently in (8). Generally, this section is hard to follow.
>
> Thank you for pointing out this ambiguity. We have made several corrections to improve clarity:
>
> - We now specify that the prior SDE's initial state distribution as $\mu_0$ (and make it independent of $\alpha$), which aligns better with our problem formulation.
> - We corrected equation (8) to show decoder parameters $\theta$ rather than $\alpha$ in the likelihood function. This was a typo in the original version of the manuscript.
> - We changed the notation for the variational approximation of the initial state $z_0$ from $p_\phi(z_0|x_{\textrm{obs}})$ to $q_\phi(z_0|x_{\textrm{obs}})$ to make the rest of the manuscript more consistent.
>
> These revisions should make the variational neural SDE formulation clearer to understand.
>
> > The variational distribution and its parameters $\phi$ appear in the Algorithm Summary in Section 4.2; however, they are not properly defined before.
>
> Thank you for noticing this oversight. We have updated the Algorithm Summary section to properly define the variational approximation and its parameters $\phi$. We also fixed the model diagram to correctly show $q_\phi(z_0|x_{\rm obs})$ instead of $p_\phi(z_0|x_{\rm obs})$. These changes will be reflected in the final version of the manuscript.
>
> > The Lipschitz continuity of the drift and diffusion terms seems to be required in the spatial dimension. In the temporal dimension, Holder continuity (with $\alpha=0.5$) seems to be sufficient for, e.g., Euler-Maruyama.
>
> Thank you for this comment, as it helps to improve the technical rigor of our manuscript. Based on Theorem 5.2.1 in [1], the existence of a strong solution to an SDE (assumed to take both time and state as input) requires only:
>
> - *Linear growth bound*: $\|f(t, x)\|+\|g(t, x)\| \leq \gamma_1 (1+\|x\|)$, where $ x\in\mathbb{R}^{d_x}$ and $t\in[0, T]$.
> - *Lipschitz continuity in spatial dimension*: $\|f(t, x_1)-f(t, x_2)\|+\|g(t, x_1) - g(t, x_2)\| \leq \gamma_2 \|x_1-x_2\|$, where $x_1, x_2 \in\mathbb{R}^{d_x}$ and  $t\in[0, T]$ for some constants $\gamma_1$ and $\gamma_2$.
>
> To the best of our knowledge, temporal smoothness is not required for strong solution existence. This is also not used in the Euler-Maruyama convergence proofs in [2]. In our revision of the manuscript, we will revise our assumptions on the nature of the SDE to be consistent with both [1] and [2] and cite the references to the appropriate assumptions/theorems directly. We also will remove any assumption regarding smoothness in the temporal dimension of the SDE.
>
> [1] Oksendal, Bernt. Stochastic differential equations: an introduction with applications. Springer Science & Business Media, 2013.
>
> [2] Kloeden, Peter E., et al. Stochastic differential equations. Springer Berlin Heidelberg, 1992.

---

> ### Author Response · Authors · 2024-12-30
> **Response to Reviewer Pvqi (comparison to Ryzhikov et al. + SDE backpropagation)**
>
> > One version ("sequential likelihood ratio test") of the second step of the proposed algorithm ("Update the change points") seems similar to the work by Ryzhikov et al. (2022). In that sense, the distinction between offline and online algorithms in Section 3.2 seems somewhat artificial.
>
> Thank you for your comment. The work proposed by [1] proposes to use a trained neural SDE model (LatentSDE) for change point detection. According to Algorithm 1 in their paper, time-series are used to train a single LatentSDE model (*no change points assumed*). After the model is trained, trajectories are sampled from the posterior of the LatentSDE, which are then used to compute the change point detection score (a lagged sum of log-likelihood ratios with hyperparameter $L$ denoting the lag).  In the approach proposed in [1], change points are not incorporated in the modeling of the LatentSDE. Rather, a single expressive LatentSDE model is used to compute likelihood ratios for change point detection. To elaborate, a change point detection score, which they define as:
>
> $$\textrm{CCPD}(x_t) = \sum_{l=1}^L \log\left( \frac{p(x_t|t)}{ p(x_t|t-l)}\right)$$
>
> Assuming a lag of $L=1$, this reduces to a single likelihood ratio comparing the likelihood of $x_t$ being explained by $f(z_{t-1})$ (null hypothesis) and $f(z_{t})$ (alternative hypothesis), where $f$ denotes the decoder function in their model. Importantly, to compute this statistic, change points do not need to be incorporated in the LatentSDE model. In our approach, our likelihood ratio explicitly tests whether a new observation $x_t$ is better explained by propagating the LatentSDE with drifts/diffusion with parameters $\theta_0$ (null hypothesis) and $\theta_1$ (alternative hypothesis), respectively. In turn, this requires that we explicitly incorporate change points in our model. Therefore, the nature of the sequential likelihood ratio tests utilized in [1] and our paper are completely different. We recognize that this distinction was not made clear in our initial version of the manuscript. We will update the manuscript accordingly to make the distinction clearer.
>
> > Is "backpropagating through the SDE solver" not implicitely using "the reparametrization trick". It would be good to clarify the discussion before Section 4.3.2.
>
> Thank you for your comment. After carefully thinking about it, we think it is important to highlight the distinction between the two. The reparameterization trick is a general technique that allows us to reduce noise in gradient computation when backpropagating through stochastic nodes belonging to certain families of parametric distributions (like Gaussians). Generally, the reparameterization trick can exist in contexts outside of SDEs but can be applied to them. For example, the reparameterization trick can be easily applied when the Euler-Maruyama numerical employed is used for sampling from an SDE, since the noise is driven by Brownian motion. There are other ways to "backpropagate" through an SDE; one method typically used is the adjoint sensitivity method, which computes gradients by solving a backward equation and does not use the reparameterization trick. Another technique is the pathwise method [3], which is the akin to being the ``continuous-time analog" of the reparameterization trick. [2] provides a more thorough discussion on the difference between the solvers, as well as their memory/time complexities. In our revision of the manuscript, we will clarify the distinction between backpropagating through an SDE and the reparameterization trick. We will also include reference to the pathwise method for obtaining stochastic gradients with appropriate references. We will also include a reference to [2] for more information on comparing different solvers.
>
> [1] Ryzhikov, Artem, Mikhail Hushchyn, and Denis Derkach. "Latent Stochastic Differential Equations for Change Point Detection." IEEE Access (2023).
>
> [2] Li, Xuechen, et al. "Scalable gradients for stochastic differential equations." International Conference on Artificial Intelligence and Statistics. PMLR, 2020.
>
> [3] Yang, Jichuan, and Harold J. Kushner. "A Monte Carlo method for sensitivity analysis and parametric optimization of nonlinear stochastic systems." SIAM journal on control and optimization 29.5 (1991): 1216-1249.

---

> ### Author Response · Authors · 2024-12-30
> **Response to Reviewer Pvqi (misc. questions)**
>
> > It would enhance accessibility to add further explanations to Section 4.3.2, both on the motivation/derivation and the estimator. Is it correct that this regularization is not considered in the theoretical results?
>
> That is correct, the regularization is not considered in the theoretical results and the choice of this regularizer was chosen based on intuition rather than concrete theoretical evidence. Understanding theoretically why the regularizer works is open research that we will consider for future work, although we believe it is linked to noise estimation issues in neural SDEs model, which was recently discussed in [4]. In our revision of the manuscripts, we will expand the discussion and motivation of the predictive regularizer. We will also include a reference to [1] and include the theoretical understanding of the regularizer as a line of future work.
>
> > Why are the MLE-based change point updates not evaluated for the experiment in Section 5.1.2?
>
> The MLE-based change point update has a theoretical guarantee but is computationally expensive in practice, requiring ${\cal O}(NK|\mathcal{T}|)$ runs of a particle filter (where $N$ denotes the number of time-series samples, $K$ denotes the number of potential change points, and $|\mathcal{T}|$ represents the number of propagation steps. We demonstrated in the toy experiments that the updates based on the SLRT achieves similar performance and can be implemented efficiently, as all change points scores can be computed with a single pass of the particle filter. Therefore, in the real-data experiments, we opted to only evaluate the SLRT-based updates instead of the MLE-based change point update.
>
> > Why does the experiment in Section 5.2 not consider the method by Sun et al. (2024)?
>
> In our real data experiments, we compared SDEGAN and our approach, but did not show a comparison with [2]. We made this choice because we observed that out of the box, our model (without any change points considered) outperforms SDEGAN by large margins (across all metrics) on most datasets. One would not expect that SDEGAN with change points would improve this margin. We will try adding CP-SDEGAN [2] in our real data experiments to further substantiate our claim. In our revision of the manuscript, we will also point out this difference in performance between our model (without change points) and SDEGAN.
>
> > Can the authors comment on the shortcoming that the number of change points needs to be known in advance?
>
> In the revision of the manuscript, we will comment on this specific limitation. We will also add a discussion on how one can choose the number of changes using model selection. In practice, we can find the best estimate of the number of change points by cross validation. Since our model allows for straightforward computation of the marginal likelihood, we can use this as a metric to compare models assuming different numbers of change points. We will also mention that a future direction of research would be tackling the problem of unknown number of change points and heterogeneous change points (different time series having different change points occurring at different times).
>
> [1] Heck, Linus, et al. "Improving the Noise Estimation of Latent Neural Stochastic Differential Equations." arXiv preprint arXiv:2412.17499 (2024).
>
> [2] Sun, Zhongchang, Yousef El-Laham, and Svitlana Vyetrenko. "Neural Stochastic Differential Equations with Change Points: A Generative Adversarial Approach." ICASSP 2024-2024 IEEE International Conference on Acoustics, Speech and Signal Processing (ICASSP). IEEE, 2024.

---

> ### Author Response · Authors · 2024-12-30
> **Response to Reviewer Pvqi (Validity of Assumption - Inference Gap)**
>
> > The "assumption that the inference gap as a result of the variational approximation does not widen after change points are updated" seems to be quite strong. Can the authors comment on its validity?
>
> Our justification for Assumption 3 can be found in our appendix. Confusion regarding this assumption may be coming from a typo we found in our justification. In our manuscript, we claimed that:
>
> $$ \Delta{\cal L}(\nu', \nu) \geq \delta_{\rm cp}, \quad \delta_{\rm cp}\geq 0 \qquad \mathrm{and} \qquad \Delta D_{\rm KL}(\nu', \nu) \geq \delta_{\rm KL}, \quad \delta_{\rm cp} \geq \delta_{\rm KL}$$
>
> implies that:
>
> $$  \Delta {\cal L}(\nu', \nu) \geq \Delta D_{\rm KL}(\nu', \nu) \implies \Delta{\cal E}(\nu', \nu)= \Delta {\cal L}(\nu', \nu)- \Delta D_{\rm KL}(\nu', \nu)\geq 0,$$
>
> which implies fixed point convergence in terms of the evidence lower-bound (ELBO). The logic above is incorrect. It can be corrected by modifying the assumption about the change in the KLDs to be as follows:
>
> $$ \Delta D_{\rm KL}(\nu', \nu) \leq \delta_{\rm KL}, \quad \delta_{\rm cp}\geq \delta_{\rm KL} $$
>
> We emphasize this error was a typo and the logic from our justification for Assumption 3 should still follow. To elaborate more, the justification for Assumption 3 boils down to showing that:
>
> $$ E_{q_{\phi}}\left[\log\left(\frac{p_{\theta,\nu}(x_{\rm obs}|z_0)}{p_{\theta,\nu'}(x_{\rm obs}|z_0)}\right)\right] \leq 0$$
>
> If the above statement is true, then indeed $\Delta D_{\rm KL}(\nu', \nu) \leq \delta_{\rm KL}\leq \delta_{\rm cp}$. In our manuscript, we justified the above inequality by saying that the change point $\nu'$ should lead to better conditional likelihood than the old change point $\nu$, since change point updates are made to improve the marginal likelihood. To elaborate we have that for a change point $\tau$:
>
> $$ p_{\theta,\tau}(x_{\rm obs}|z_0) = \int \left(\prod_{k=1}^K p_{\theta}(x_{t_k}|z_{t_k}) p_{\theta}(z_{t_k}|z_{t_{k-1}}, \nu=\tau) \right)dz_{t_1}\cdots dz_{t_K}$$
>
>  where we define $ {z}_{t_0}= {z}_0$. The only difference between this expression and the marginal likelihood (which improves with every change point update) is the conditioning on $ {z}_0$. We can see that as the time horizon increases, the dependence on the initial state will get smaller and smaller. Intuitively, this would imply that for longer time series, the updated change point should have larger conditional likelihood regardless of the ${z_0}$,:
>
> $$ \log\left(\frac{p_{ \theta,\nu}( x_{\rm obs}| z_0)}{p_{ \theta,\nu'}( x_{\rm obs}| z_0)}\right) \leq 0, \quad  z_0 \in \mathbb{R}^{d_z}$$
>
> In our revision of the manuscript, we will correct the direction of the inequality in Assumption 3 and also add an extended discussion on the dependence of the conditional likelihood function on the length of the time-series in order to provide further justification for the validity of Assumption 3.

---

### Decision · Action_Editor_Xv7v · 2025-02-07

**Recommendation:** Accept with minor revision

**Comment:**

The paper is based on a principled framework and is accessible. The reviewers all favor publication.

The authors have largely taken the comments of the reviewers into account, with some exceptions that I find important to include in the final version (see comments below). I also have the following comments that I would like the authors to address before publication:

- Please clarify what distribution p(zt) in Eq (11) is.
- What is the approximation that is done from Eq (10) to (11)?  Please add an explanation to the paper.
- From the notation, I guess the samples in Eq (12) are the samples from the variational posterior? Please clarify in the paper. (If is the variational posterior, using q_phi(z_t|xobs)) as before Eq (8) would be more suitable notation than the p(zt) in (11)
- Sec 4.3.2, first paragraph. Note the typo "is the it will" in the sentence ", a weakness of the training loss for the model parameters of the SDEVAE model is the it will also lead to an underestimation of noise variance"

- Results: You point out that LS4 is the "closest rival". Please add a discussion about the pros and cons of LS4 and your method, and when to use which. Please note that Reviewer 8zqg asked a similar question, but I couldn't spot a discussion in the paper. Please add a sufficiently detailed one, or let me know where you have included it if done already.

- I cannot see the "broader impact statement" requested by Reviewer EA6f. Can you please add a brief paragraph?

**Audience:**

The paper is concerned with modelling time-series data with distributional shifts, and detecting them. The model is based on latent stochastic differential equations, and for estimation, a VAE framework is adopted.  These are all topics that some readers of TMLR will find interesting.

**Claims And Evidence:**

Theoretical analysis and simulations, both with synthetic and real data, provide evidence for the proposed method.